# ACTIONABLE NEURAL REPRESENTATIONS:
# GRID CELLS FROM MINIMAL CONSTRAINTS

**William Dorrell, Peter Latham**
Gatsby Unit, UCL
dorrellwec@gmail.com

**Timothy E.J. Behrens**
UCL & Oxford

**James C.R. Whittington**
Oxford & Stanford
jcrwhittington@gmail.com

## ABSTRACT

To afford flexible behaviour, the brain must build internal representations that mirror the structure of variables in the external world. For example, 2D space obeys rules: the same set of actions combine in the same way everywhere (step north, then south, and you won't have moved, wherever you start). We suggest the brain must represent this consistent meaning of actions across space, as it allows you to find new short-cuts and navigate in unfamiliar settings. We term this representation an 'actionable representation'. We formulate actionable representations using group and representation theory, and show that, when combined with biological and functional constraints - non-negative firing, bounded neural activity, and precise coding - multiple modules of hexagonal grid cells are the optimal representation of 2D space. We support this claim with intuition, analytic justification, and simulations. Our analytic results normatively explain a set of surprising grid cell phenomena, and make testable predictions for future experiments. Lastly, we highlight the generality of our approach beyond just understanding 2D space. Our work characterises a new principle for understanding and designing flexible internal representations: they should be actionable, allowing animals and machines to predict the consequences of their actions, rather than just encode.

## 1 INTRODUCTION

Animals should build representations that afford flexible behaviours. However, different representation make some tasks easy and others hard; representing red versus white is good for understanding wines but less good for opening screw-top versus corked bottles. A central mystery in neuroscience is the relationship between tasks and their optimal representations. Resolving this requires understanding the representational principles that permit flexible behaviours such as zero-shot inference.

Here, we introduce **actionable representations**, a representation that permits flexible behaviours. Being actionable means encoding not only variables of interest, but also how the variable transforms. Actions cause many variables to transform in predictable ways. For example, actions in 2D space obey rules; north, east, south, and west, have a universal meaning, and combine in the same way everywhere. Embedding these rules into a representation of self-position permits deep inferences: having stepped north, then east, then south, an agent can infer that stepping west will lead home, having never taken that path - a zero-shot inference (Figure 1A).

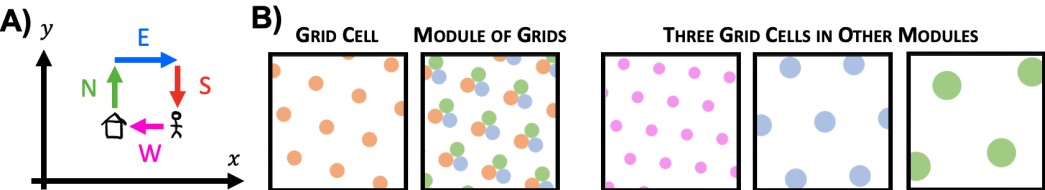

Figure 1: **A** 2D space is defined by rules, e.g. at all positions north $= -$south. **B** Left: Entorhinal grid cells are hexagonally tuned cells (orange). Different cells within a module are translated copies (orange/blue/green). Right: Different modules have different lattice scale (pink/blue/green).

Indeed biology represents 2D space in a structured manner. Grid cells in medial entorhinal cortex represent an abstracted 'cognitive map' of 2D space (Tolman, 1948). These cells fire in a hexagonal lattice of positions (Hafting et al., 2005), (Figure 1B), and are organised in modules; cells within one module have receptive fields that are translated versions of one another, and different modules have firing lattices of different scales and orientations (Figure 1B), (Stensola et al., 2012).

Biological representations must be more than just actionable - they must be **functional**, encoding the world efficiently, and obey **biological** constraints. We formalise these three ideas - actionable, functional, and biological - and analyse the resulting optimal representations. We define *actionability* using group and representation theory, as the requirement that each action has a corresponding matrix that linearly updates the representation; for example, the 'step north' matrix updates the representation to its value one step north. *Functionally*, we want different points in space to be represented maximally differently, allowing inputs to be distinguished from one another. *Biologically*, we ensure all neurons have non-negative and bounded activity. From this constrained optimisation problem we derive optimal representations that resemble multiple modules of grid cells.

Our problem formulation allows analytic explanations for grid cell phenomena, matches experimental findings, such as the alignment of grids cells to room geometry (Stensola et al., 2015), and predicts some underappreciated aspects, such as the relative angle between modules. In sum, we 1) propose actionable neural representations to support flexible behaviours; 2) formalise the actionable constraint with group and representation theory; 3) mix actionability with biological and functional constraints to create a constrained optimisation problem; 4) analyse this problem and show that in 2D the optimal representation is a good model of grid cells, thus offering a mathematical understanding of why grid cells look the way they do; 5) provide several neural predictions; 6) highlight the generality of this normative method beyond 2D space.

## 1.1 RELATED WORK

Neuroscientists have long explained representations with normative principles like information maximisation (Attneave, 1954; Barlow et al., 1961), sparse (Olshausen & Field, 1996) or independent (Hyvärinen, 2010) latent encodings, often mixed with biological constraints such as non-negativity (Sengupta et al., 2018), energy budgets (Niven et al., 2007), or wiring minimisation (Hyvärinen et al., 2001). On the other hand, deep learning learns task optimised representations. A host of representation-learning principles have been considered (Bengio et al., 2013); but our work is most related to geometric deep learning (Bronstein et al., 2021) which emphasises input transformations, and building neural networks which respect (equivariant) or ignore (invariant) them. This is similar in spirit but not in detail to our approach, since equivariant networks do not build representations in which all transformations of the input are implementable through matrices. Most related are Paccanaro & Hinton (2001), who built representations in which relations (e.g. $x$ is the father of $y$) are enacted by a corresponding linear transform, exactly like our notion of actionable!

There is much previous theory on grid cells, which can be categorised as relating to our actionable, functional, and biological constraints. **Functional:** Many works argue that grid cells provide an efficient representation of position, that hexagons are optimal (Mathis et al., 2012a;b; Sreenivasan & Fiete, 2011; Wei et al., 2015) and make predictions for relative module lengthscales (Wei et al., 2015). Since we use similar functional principles, we suspect that some of our novel results, such as grid-to-room alignment, could have been derived by these authors. However, in contrast to our work, these authors assume a grid-like tuning curve. Instead we give a normative explanation of why be grid-like at all, explaining features like the alignment of grid axes within a module, which are detrimental from a pure decoding view (Stemmler et al., 2015). **Actionability:** Grid cells are thought to a basis for predicting future outcomes (Stachenfeld et al., 2017; Yu et al., 2020), and have been classically understood as affording path-integration (integrating velocity to predict position) with units from both hand-tuned Burak & Fiete (2009) and trained recurrent neural network resembling grid cells (Cueva & Wei, 2018; Banino et al., 2018; Sorscher et al., 2019). Recently, these recurrent network approaches have been questioned for their parameter dependence (Schaeffer et al., 2022), or relying on decoding place cells with bespoke shapes that are not observed experimentally (Sorscher et al., 2019; Dordek et al., 2016). Our mathematical formalisation of path-integration, combined with biological and functional constraints, provides clarity on this issue. Our approach is linear, in that actions update the representation linearly, which has previously been explored theoretically (Issa & Zhang, 2012), and numerically, in two works that learnt grid cells (Whittington et al., 2020; Gao

et al., 2021). Our work could be seen as extracting and simplifying the key ideas from these papers that make hexagonal grids optimal (see Appendix H), and extending them to multiple modules, something both papers had to hard code. **Biological:** Lastly, both theoretically (Sorscher et al., 2019) and computationally (Dordek et al., 2016; Whittington et al., 2021), non-negativity has played a key role in normative derivations of hexagonal grid cells, as it will here.

## 2 ACTIONABLE NEURAL REPRESENTATIONS: AN OBJECTIVE

We seek a representation $\boldsymbol{g}(\boldsymbol{x}) \in \mathbb{R}^N$ of 2D position $\boldsymbol{x} \in \mathbb{T}^2$, where $N$ is the number of neurons. Our representation is built using three ideas: functional, biological, and actionable; whose combination will lead to multiple modules of grid cells, and which we'll now formalise.

**Functional:** To be useful, the representation must encode different positions differently. However, it is more important to distinguish positions 1km apart than 1mm, and frequently visited positions should be separated the most. To account for these, we ask our representation to minimise

$$\mathcal{L} = \iint e^{-\frac{\|\boldsymbol{g}(\boldsymbol{x}) - \boldsymbol{g}(\boldsymbol{x}')\|^2}{2\sigma^2}} \chi(\boldsymbol{x}, \boldsymbol{x}') p(\boldsymbol{x}) p(\boldsymbol{x}') d\boldsymbol{x} d\boldsymbol{x}' \tag{1}$$

The red term measures the representational similarity of $\boldsymbol{x}$ and $\boldsymbol{x}'$; it is large if their representations are nearer than some distance $\sigma$ in neural space and small otherwise. By integrating over all pairs $\boldsymbol{x}$ and $\boldsymbol{x}'$, $\mathcal{L}$ measures the total representational similarity, which we seek to minimise. The green term is the agent's position occupancy distribution, which ensures only visited points contribute to the loss, for now simply a Gaussian of lengthscale $L$. Finally, the blue term weights the importance of separating each pair, encouraging separation of points more distant than a lengthscale, $l$.

$$\chi(\boldsymbol{x}, \boldsymbol{x}') = 1 - e^{-\frac{\|\boldsymbol{x} - \boldsymbol{x}'\|^2}{2l^2}} \qquad p(\boldsymbol{x}) = e^{-\frac{\|\boldsymbol{x}\|^2}{2L^2}} \tag{2}$$

**Biological:** Neurons have non-negative firing rates, so we constrain $\boldsymbol{g}_i(\boldsymbol{x}) \geq 0$. Further, neurons can't fire arbitrarily fast, and firing is energetically costly, so we constrain each neuron's response $g_n(\boldsymbol{x})$ via $\int g_n^2(\boldsymbol{x}) p(\boldsymbol{x}) d\boldsymbol{x} = 1$

**Actionable:** Our final constraint requires that the representation is actionable. This means each transformations of the input must have its own transformation in neural space, independent of position. For mathematical convenience we enact the neural transformation using a matrix. Labelling this matrix $\boldsymbol{T}(\Delta\boldsymbol{x}) \in \mathbb{R}^{N \times N}$, for transformation $\Delta\boldsymbol{x}$, this means that for all positions $\boldsymbol{x}$,

$$\boldsymbol{g}(\boldsymbol{x} + \Delta\boldsymbol{x}) = \boldsymbol{T}(\Delta\boldsymbol{x})\boldsymbol{g}(\boldsymbol{x}) \tag{3}$$

For intuition into how this constrains the neural code $\boldsymbol{g}(\boldsymbol{x})$, we consider a simple example of two neurons representing an angle $\theta \in [0, 2\pi)$. Replacing $\boldsymbol{x}$ with $\theta$ in equation 3 we get the equivalent constraint: $\boldsymbol{g}(\theta + \Delta\theta) = \boldsymbol{T}(\Delta\theta)\boldsymbol{g}(\theta)$. Here the matrix $\boldsymbol{T}$ performs a rotation, and the solution (up to a linear transform) is for $\boldsymbol{T}$ to be the standard $2 \times 2$ rotation matrix, with frequency $n$.

$$\boldsymbol{g}(\theta + \Delta\theta) = \begin{pmatrix} \cos(n[\theta + \Delta\theta]) \\ \sin(n[\theta + \Delta\theta]) \end{pmatrix} = \begin{pmatrix} \cos(n\Delta\theta) & -\sin(n\Delta\theta) \\ \sin(n\Delta\theta) & \cos(n\Delta\theta) \end{pmatrix} \begin{pmatrix} \cos(n\theta) \\ \sin(n\theta) \end{pmatrix} = \boldsymbol{T}(\Delta\theta)\boldsymbol{g}(\theta) \tag{4}$$

Thus as $\theta$ rotates by $2\pi$ the neural representation traces out a circle an integer number, $n$, times. Thanks to the problem's linearity, extending to more neurons is easy. Adding two more neurons lets the population contain another sine and cosine at some frequency, just like the two neurons in equation 4. Extrapolating this we get our actionability constraint: the neural response must be constructed from some invertible linear mixing of the sines and cosines of $D < \frac{N}{2}$ frequencies,

$$\boldsymbol{g}(\theta) = \boldsymbol{a}_0 + \sum_{d=1}^{D} \boldsymbol{a}_d \sin(n_d\theta) + \boldsymbol{b}_d \cos(n_d\theta) \qquad \text{for integer } n_d \tag{5}$$

The vectors $\{\boldsymbol{a}_d, \boldsymbol{b}_d\}_{d=1}^{D} \in \mathbb{R}^N$ are coefficient vectors that mix together the sines and cosines, of which there are $D$. $\boldsymbol{a}_0$ is the coefficient vector for a frequency that cycles 0 times.

This argument comes from an area of maths called Representation Theory (a different meaning of representation!) that places constraints on the matrices $\boldsymbol{T}$ for variables whose transformations form a mathematical object called a group. This includes many of interest, such as position on a circle,

torus, or sphere. These constraints on matrices can be translated into constraints on an actionable neural code just like we did for $g(\theta)$ (see Appendix A). When generalising the above example to 2D space (a torus), we must consider a few things: First, the space is two-dimensional, so compared to our previous equation 5, the frequencies, denoted $k_d$, are now two dimensional. Second, to approximate a finite region of flat 2D space, we consider a similarly sized region of a torus. As the radius of the torus grows this approximation becomes arbitrarily good (see Appendix A.4 for discussion). Periodicity constrains the frequencies in equation 5 to be $\frac{n}{R}$ for integer $n$ and ring radius $R$. As the loop (torus in 2D) becomes very large these permitted frequencies become arbitrarily close, so we drop the integer constraint,

$$g(\boldsymbol{x}) = \boldsymbol{a}_0 + \sum_{d=1}^{D} \boldsymbol{a}_d \sin(\boldsymbol{k}_d \cdot \boldsymbol{x}) + \boldsymbol{b}_d \cos(\boldsymbol{k}_d \cdot \boldsymbol{x}) \tag{6}$$

Our constrained optimisation problem is complete. Equation 6 specifies the set of actionable representations. Without additional constraints these codes are meaningless: random combinations of sines and cosines produce random neural responses (Figure 2A). We will choose from amongst the set of actionable codes by optimising the parameters $\boldsymbol{a}_0, \{\boldsymbol{a}_d, \boldsymbol{b}_d, \boldsymbol{k}_d\}_{d=1}^{D}$ to minimise $\mathcal{L}$, subject to biological (non-negative and bounded firing rates) constraints.

## 3 OPTIMAL REPRESENTATIONS

Optimising over the set of actionable codes to minimise $\mathcal{L}$ with biological constraints gives multiple modules of grid cells (Figure 2B). This section will, using intuition and analytics, explain why.

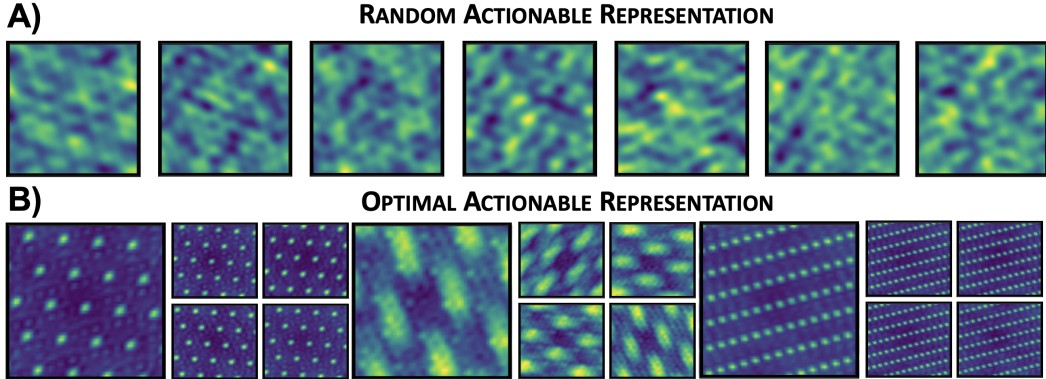

Figure 2: **A** Random actionable representations (equation 6) are meaningless combinations of sines and cosines. ($g_n(\boldsymbol{x})$ plotted for different neurons, $n$) **B** Optimising among actionable codes to achieve functional and biological constraints produces multiple modules of $\sim$hexagonal grid cells. (Figure 6 shows that all the neurons in the population belong to one of these three modules)

### 3.1 NON-NEGATIVITY LEADS TO A MODULE OF LATTICE CELLS

To understand how non-negativity produces modules of lattice responses we will study the following simplified loss, which maximises the *Euclidean distance* between representations of angle, $g(\theta)$,

$$\mathcal{L}_0 = -\frac{1}{4\pi^2} \iint_{-\pi}^{\pi} \|g(\theta) - g(\theta')\|^2 d\theta d\theta' \tag{7}$$

This is equivalent to the full loss (equation 1) for uniform $p(\theta)$, $\chi(\theta, \theta') = 1$, and $\sigma$ very large. Make no mistake, this is a bad loss. For contrast, the full loss encouraged the representations of different positions to be separated by more than $\sigma$, enabling discrimination[1]. Therefore, sensibly, the representation is most rewarded for separating nearby (closer than $\sigma$) points. $\mathcal{L}_0$ does the opposite! It grows quadratically with separation, so $g(\theta)$ is most rewarded for pushing apart already well-separated points, a terrible representational principle! Nonetheless, $\mathcal{L}_0$ will give us key insights.

---

[1]$\sigma$ could be interpreted as a noise level, or a minimum discriminable distance, then points should be far enough away for a downstream decoder to distinguish them.

Since actionability gives us a parameterised form of the representations (equation 5), we can compute the integrals to obtain the following constrained optimisation problem (details: Appendix C)

$$\min_{\substack{\boldsymbol{a}_0, \\ \{\boldsymbol{a}_d, \boldsymbol{b}_d, n_d\}_{d=1}^D}} \mathcal{L}_0 = -\sum_{d=1}^D \|\boldsymbol{a}_d\|^2 + \|\boldsymbol{b}_d\|^2 \quad \text{with} \quad \overbrace{\boldsymbol{g}(\theta) > 0}^{\text{Non-negativity}}, \quad \overbrace{\|\boldsymbol{a}_0\|^2 + \frac{1}{2}\sum_{d=1}^D \|\boldsymbol{a}_d\|^2 + \|\boldsymbol{b}_d\|^2 = N}^{\text{Bounded firing rates}}$$

(8)

Where $N$ is the number of neurons. This is now something we can understand. First, reminding ourselves that the neural code, $\boldsymbol{g}(\theta)$, is made from a constant vector, $\boldsymbol{a}_0$, and $\theta-$dependent parts (equation 5; Figure 3A), we can see that $\mathcal{L}_0$ separates representations by encouraging the size of each varying part, $\|\boldsymbol{a}_d\|^2 + \|\boldsymbol{b}_d\|^2$, to be maximised. This effect is limited by the firing rate bound, $\|\boldsymbol{a}_0\|^2 - \frac{1}{2}\mathcal{L}_0 = N$. Thus, to minimise $\mathcal{L}_0$ we must minimise the constant vector, $\boldsymbol{a}_0$. This would be easy without non-negativity (when any code with $\|\boldsymbol{a}_0\| = 0$ is optimal), but no sum of sines and cosines can be non-negative for all $\theta$ without an offset. Thus the game is simple; choose frequencies and coefficients so the firing rates are non-negative, but using the smallest possible constant vector.

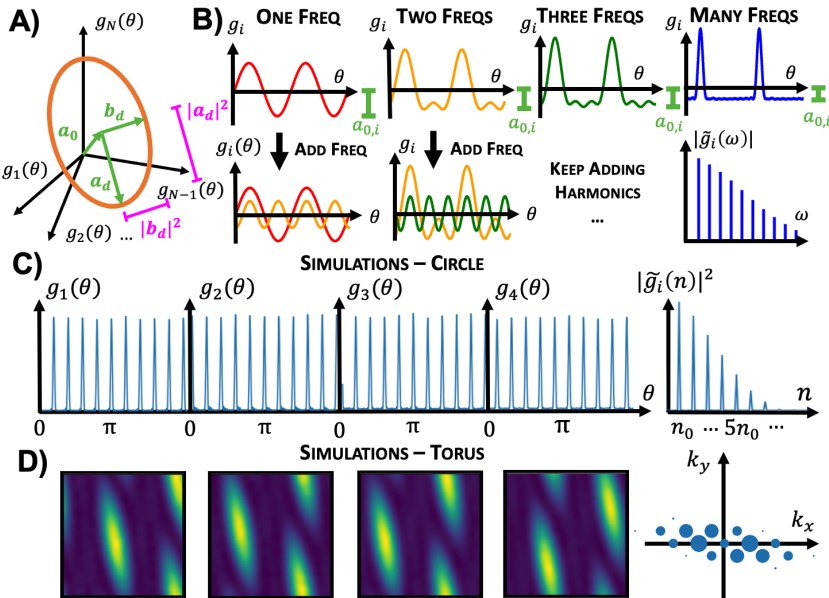

Figure 3: **A** The neural activity consists of a constant vector, $\boldsymbol{a}_0$, and $\theta-$dependent loops. **B** Progressively adding harmonic frequencies increases the code's minima, allowing the code to be made non-negative using the smallest possible $\boldsymbol{a}_0$. This give a grid-like tuning curve. Simulations results confirm this heuristic in **C** 1D and **D** 2D, right: frequency spectrum, 2D dot size is frequency power.

**One lattice cell**. We now heuristically argue, and confirm in simulation, that the optimal solution for a single neuron is a lattice tuning curve (see Appendix G for why this problem is non-convex). Starting with a single frequency component, e.g. $\sin(\theta)$, achieving non-negativity requires adding a constant offset, $\sin(\theta) + 1$ (Figure 3B). However, we could also have just added another frequency. In particular adding harmonics of the base frequency (with appropriate phase shifts) pushes up the minima (Figure 3B). Extending this argument, we suggest non-negativity, for a single cell, can be achieved by including a grid of frequencies. This gives a lattice tuning curve (Figure 3B right).

**Module of lattice cells**. Achieving non-negativity for this cell used up many frequencies. But as discussed (Section 2), actionability only allows a limited number frequencies in the population ($< \frac{N}{2}$ since each frequency uses 2 neurons (sine and cosine)), thus how can we make lots of neurons non-negative with limited frequencies? Fortunately, we can do so by making all neuron's tuning curves translated versions of each other, as translated curves contain the same frequencies but with different phases. This is a module of lattice cells. We validate our arguments by numerically optimising the coefficients $\boldsymbol{a}_0, \{\boldsymbol{a}_d, \boldsymbol{b}_d\}_{d=1}^D$ and frequencies $\{n_d\}_{d=1}^D$ to minimise $\mathcal{L}_0$ subject to constraints, producing a module of lattices (Figure 3C; details in Appendix B). These arguments equally apply to representations of a periodic 2D space (a torus; Figure 3D).

Studying $\mathcal{L}_0$ has told us why lattice response curves are good. But surprisingly, all lattices are equally good, even at infinitely high frequency. Returning to the full loss will break this degeneracy.

## 3.2 Prioritising Important Pairs of Positions Produces Hexagonal Grid Cells

Now we return to the full loss and understand its impact in two steps, beginning with the reintroduction of $\chi$ and $p$, which break the lattice degeneracy, forming hexagonal grid cells.

$$\mathcal{L} = \iint_{-\infty}^{\infty} e^{-\frac{\|g(\boldsymbol{x}) - g(\boldsymbol{x}')\|^2}{2\sigma^2}} \chi(\boldsymbol{x}, \boldsymbol{x}') p(\boldsymbol{x}) p(\boldsymbol{x}') d\boldsymbol{x} d\boldsymbol{x}' \tag{9}$$

$\chi$ **prefers low frequencies:** recall that $\chi = 1 - e^{-\frac{\|\boldsymbol{x} - \boldsymbol{x}'\|^2}{2l^2}}$ ensures very distant inputs have different representations, while allowing similar inputs to have similar representations, up to a resolution, $l$. This encourages low frequencies ($\|\boldsymbol{k}_d\| < \frac{1}{l}$), which separate distant points but produce similar representations for pairs closer than $l$ (Analytics: Appendix D.1). At this stage, for periodic 2D space, the lowest frequency lattices, place cells, are optimal (see Appendix F; Sengupta et al. (2018)).

$p(\boldsymbol{x})$ **prefers high frequencies:** However, the occupancy distribution of the animal, $p(\boldsymbol{x})$, counters $\chi$. On an infinite 2D plane animals must focus on representing a limited area, of lengthscale $L$. This encourages high frequencies ($\|\boldsymbol{k}_d\| > \frac{1}{L}$), whose response varies among the visited points (Analytics: Appendix D.2). More complex $p(x)$ induce more complex frequency biases, but, to first order, the effect is always a high frequency bias (Figure 5F-G, Appendix L).

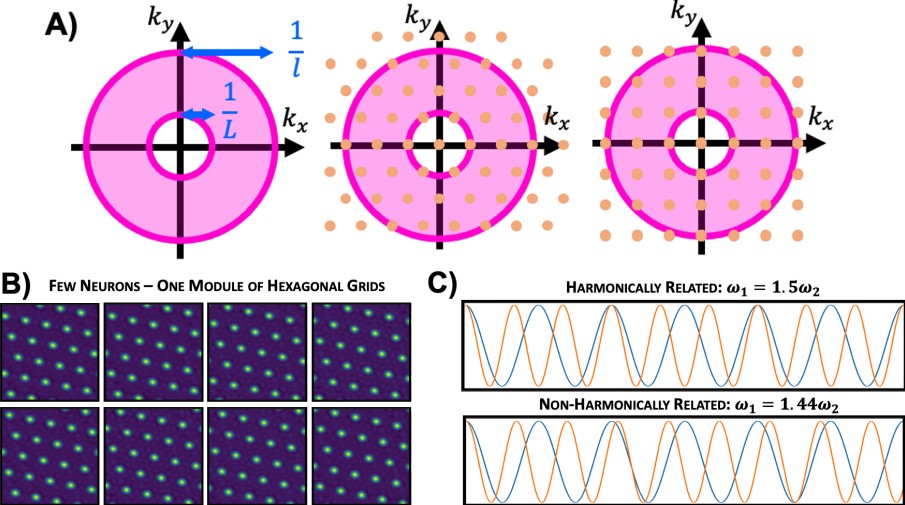

Figure 4: **A** $\chi$ and $p(\boldsymbol{x})$ induce a bias towards frequencies with magnitudes between $\frac{1}{L}$ and $\frac{1}{l}$. Since, of all lattices, hexagons fit the most frequencies within the annulus, they are preferred, and hexagonal frequency lattices lead to hexagonal grid cells. **B** Simulations confirm. **C** Harmonically related frequencies co-repeat more often than non-harmonic, meaning that, as a pair, harmonically related frequencies are worse at encoding, since they encode many points in the same way.

**Combination $\rightarrow$ Hexagons:** Satisfying non-negativity and functionality required a lattice of many frequencies, but now $p$ and $\chi$ bias our frequency choice, preferring those beyond $\frac{1}{L}$ (to separate points the animal visits) but smaller than $\frac{1}{l}$ (to separate distant visited points). Thus to get as many of these preferred frequencies as possible, we want the lattice with the densest packing within a Goldilocks annulus in frequency space (Figure 4A). This is a hexagonal lattice in frequency space which leads to a hexagonal grid cell. Simulations with few neurons agree, giving a module of hexagonal grid cells (Figure 4B).

## 3.3 A Harmonic Tussle Produces Multiple Modules

Finally, we will study the neural lengthscale $\sigma$, and understand how it produces multiple modules.

$$\mathcal{L} = \iint_{-\infty}^{\infty} e^{-\frac{\|g(\boldsymbol{x}) - g(\boldsymbol{x}')\|^2}{2\sigma^2}} \chi(\boldsymbol{x}, \boldsymbol{x}') p(\boldsymbol{x}) p(\boldsymbol{x}') d\boldsymbol{x} d\boldsymbol{x}' \tag{10}$$

As discussed, $\mathcal{L}$ prioritises the separation of poorly distinguished points, those whose representations are closer than $\sigma$. This causes certain frequencies to be desired *in the overall population*, in particular those unrelated to existing frequencies by simple harmonic ratios, i.e. not $\omega_1 = \frac{3}{2}\omega_2$ (Figure 4C; see Appendix E for a perturbative derivation of this effect). This is because pairs of harmonically related frequencies represent more positions identically than a non-harmonically related pair, so are worse for separation (similar to arguments made in Wei et al. (2015)).

This, however, sets up a 'harmonic tussle' between what the population wants - non-harmonically related frequencies for $\mathcal{L}$ - and what single neurons want - harmonically related frequency lattices for non-negativity (Section 3.1). Modules of grid cells resolve this tension: harmonic frequencies exist within modules to give non-negativity, and non-harmonically related modules allow for separation, explaining the earlier simulation results (Figure 2B; further details in Appendix E.3).

This concludes our main result. We have shown three constraints on neural populations - actionable, functional, and biological - lead to multiple modules of hexagonal grid cells, and we have understood why. We posit this is the minimal set of requirements for grid cells (see Appendix I for ablations simulations and discussion).

## 4 PREDICTIONS

Our theory makes testable predictions about the structure of optimal actionable codes for 2D space. We describe three here: tuning curve sharpness scales with the number of neurons in a module; the optimal angle between modules; and the optimal grid alignment to room geometry.

### 4.1 LATTICE SIZE:FIELD WIDTH RATIO SCALES WITH NUMBER OF NEURONS IN MODULE

In our framework the number of neurons controls the number of frequencies in the representation (equation 6). A neuron within a module only contains frequencies from that module's frequency lattice, since other modules have non-harmonically related frequencies. More neurons in a module, means more and higher frequencies in the lattice, which sharpen grid peaks (Figure 5A). We formalise this (Appendix J) and predict that the number of neurons within a module scales with the square of the lattice lengthscale, $\nu$, to field width, $\mu$, ratio, $N \propto \left(\frac{\nu}{\mu}\right)^2$. This matches the intuition that the sharper a module's peak, the more neurons you need to tile the entire space. In a rudimentary analysis, our predictions compare favourably to data from Stensola et al. (2012) assuming uniform sampling of grid cells across modules (Figure 5B). We are eager to test these claims quantitatively.

### 4.2 MODULES ARE OPTIMALLY ORIENTED AT SMALL OFFSETS ($\sim 4°$)

In section 3.3 we saw how frequencies of different modules are maximally non-harmonically related in order to separate the representation of as many points as possible. To maximise non-harmonicity between two modules, the second module's frequency lattice can be both stretched *and* rotated relative to the first. 0 or $30°$ relative orientations are particularly bad coding choices as they align the high density axes of the two lattices (Figure 5C). The optimal angular offset of two modules, calculated via a frequency overlap metric (Appendix K), is small (Figure 5D); the value depends on the grid peak and lattice lengthscales, $\mu$ and $\nu$, but varies between $3°$ and $8°$ degrees. Multiple modules should orient at a sequence of small angles (Appendix K). In a rudimentary analysis, our predictions compare favourably to the observations of Stensola et al. (2012) (Figure 5E).

### 4.3 OPTIMAL GRIDS MORPH TO ROOM GEOMETRY

In Section 3.2 (and Appendix D) we showed that $p(x)$, the animal's occupancy distribution, introduced a high frequency bias - grid cells must peak often enough to encode visited points. However, changing $p(x)$ changes the shape of this high frequency bias (Appendix L). In particular, we examine an animal's encoding of square, circular, or rectangular environments, Appendix L, with the assumption that $p(x)$ is uniform over that space. In each case the bias is coarsely towards high frequencies, but has additional intricacies: in square and rectangular rooms optimal frequencies lie on a lattice, with peaks at integer multiples of $\frac{2\pi}{L}$ along one of the cardinal axes, for room width/height $L$ (Figure 5F); whereas in circular rooms optima are at the zeros of a Bessel function (Figure 5G).

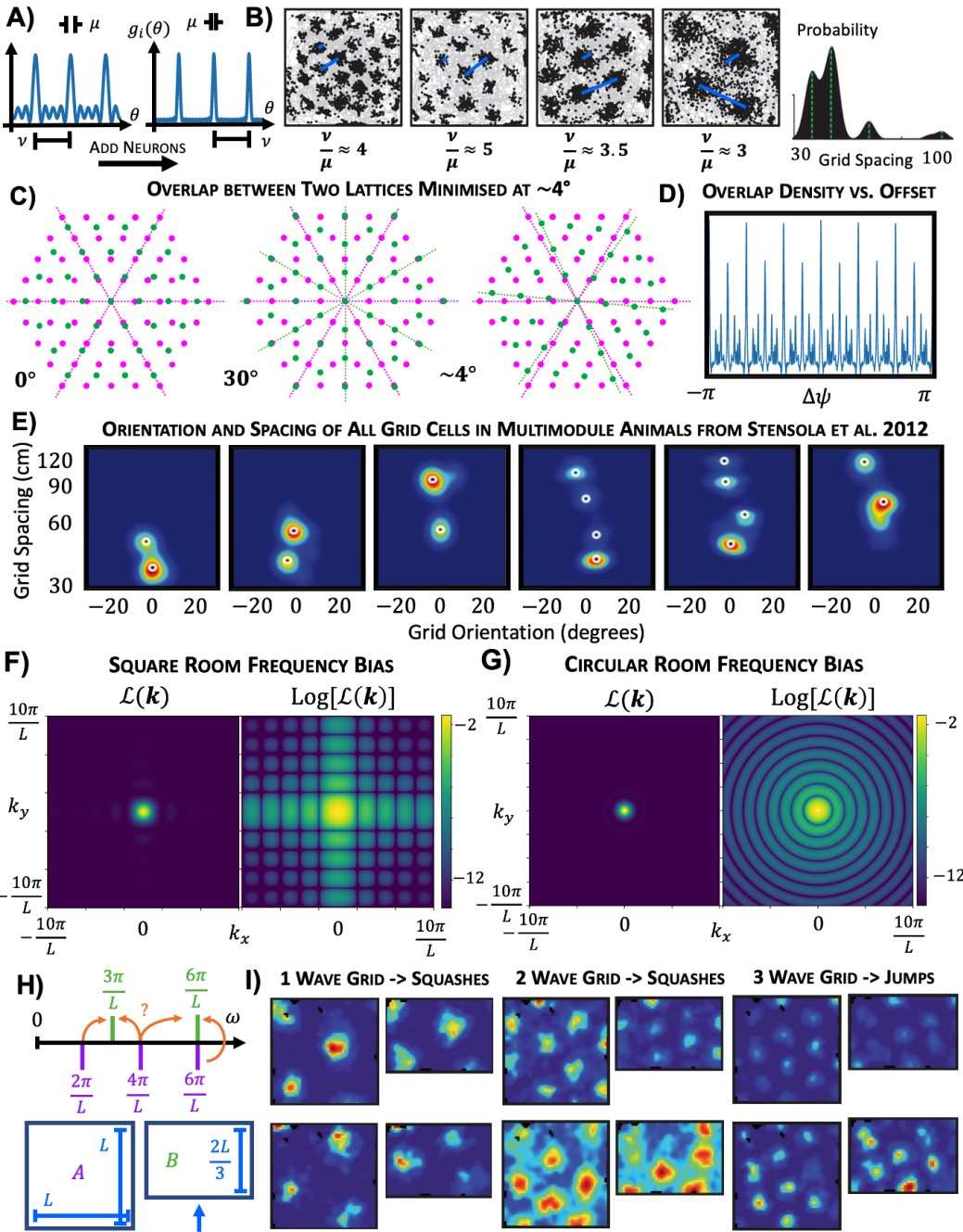

Figure 5: **A** Adding neurons to a module sharpens the grid peaks. **B** In data from Stensola et al. (2012) the most sharply peaked grids were recorded most often (2nd from left), and the broadest the least (rightmost). **C** Aligning the high-density axes of two lattices creates high overlap, while small offsets minimise it. **D** We quantified the overlap as a function of offset angle (Appendix K) and found the minima occured at $\sim 4°$ from aligned (aligned = 6 maxima, $4°$ offsets are the 12 minima). **E** The orientation and spacing of all grids in animals with multiple modules recorded by Stensola et al. (2012). Many modules are misaligned by small offsets, matching our prediction. **F** Square or **G** circular rooms create complex frequency biases, $\mathcal{L}(k)$ is the loss for a one frequency code of fixed amplitude. **H** The optimal frequencies along one axis of a box occur at $\frac{2\pi n}{L}$ for integer $n$: Squishing a room makes the optimal frequencies expand. Grids should change to fit the optimal patterns in the recorded environment, unless they happen to be optimal for both, as $\frac{6\pi}{L}$ is. **I** Most grid cells scale with the room, but, when one side is squashed by a factor of 2/3, those at $\frac{6\pi}{L}$ are stable.

These ideas make several predictions. For example, grid modules in circular rooms should have lengthscales set by the optimal radii in Figure 5G, but they should still remain hexagonal since the Bessel function is circularly symmetric. However, the optimal representation in square rooms should not be perfectly hexagonal since $p(x)$ induces a bias inconsistent with a hexagonal lattice (this effect is negligible for high frequency grids). Intriguingly, shearing towards squarer lattices is observed in square rooms (Stensola et al., 2015), and it would be interesting to test its grid-size dependence.

Lastly, these effects make predictions about how grid cells morph when the environment geometry changes. A grid cell that is optimal in both geometries can remain the same, however sub-optimal grid cells should change. For example turning a square into a squashed square (i.e. a rectangle), stretches the optimal frequencies along the squashed dimension. Thus, some cells are optimal in both rooms and should stay stable, will others will change, presumably to nearby optimal frequencies (Figure 5H). Indeed Stensola et al. (2012) recorded the same grid cells in a square and rectangular environment (Figure 5I), and observed exactly these phenomena.

## 5 Discussion & Conclusions

We have proposed actionability as a fundamental representational principle to afford flexible behaviours. We have shown in simulation and with analytic justification that the optimal actionable representations of 2D space are, when constrained to be both biological and functional, multiple modules of hexagonal grid cells, thus offering a mathematical understanding of grid cells. We then used this theory to make three novel grid cell predictions that match data on early inspection.

While this is promising for our theory, there remain some grid cell phenomena that, as it stands, it will never predict. For example, grid cell peaks vary in intensity (Dunn et al., 2017), and grid lattices bend in trapezoidal environments (Krupic et al., 2015). These effects may be due to incorporation of sensory information or uncertainty - things we have not included - to better infer position. Including these may recapitulate these findings, similar to Ocko et al. (2018) and Kang et al. (2023).

Our theory is normative and abstracted from implementation. However, both modelling (Burak & Fiete, 2009) and experimental (Gardner et al., 2022; Kim et al., 2017) work suggests that continuous attractor networks (CANs) implement path integrating circuits. Actionability and CANs imply seemingly different representation update equations; future work could usefully compare the two.

While we focused on understanding the optimal representations of 2D space and their relationship to grid cells, our theory is more general. Most simply, it can be applied to behaviour in other, non 2D, spaces. In fact many variables whose transformations form a group are relatively easily analysed. The brain represents many such variables, e.g. heading directions, (Finkelstein et al., 2015), object orientations, (Logothetis et al., 1995), the '4-loop task' of Sun et al. (2020) or 3-dimensional space (Grieves et al., 2021; Ginosar et al., 2021). Interestingly, our theory predicts 3D representations with regular order (Figure 19E in Appendix M), unlike those found in the brain (Grieves et al., 2021; Ginosar et al., 2021) suggesting the animal's 3D navigation is sub-optimal.

Further, the brain represents these differently-structured variables not one at a time, but simultaneously; at times mixing these variables into a common representation (Hardcastle et al., 2017), at others giving each variable its own set of neurons (e.g. grid cells, object-vector cells Hydal et al. (2019)). Thus, one potential concern about our work is that it assumes a separate neural population represents each variable. However, in a companion paper, we show that our same biological and functional constraints encourage any neural representation to encode independent variables in separate sub-populations (Whittington et al., 2023), to which our theory can then be cleanly applied.

But, most expansively, these principles express a view that representations must be more than just passive encodings of the world; they must embed the consequences of predictable actions, allowing planning and inferences in never-before-seen situations. We codified these ideas using Group and Representation theory, and demonstrated their utility in understanding grid cells. However, the underlying principle is broader than the crystalline nature of group structures: the world and your actions within it have endlessly repeating structures whose understanding permits creative analogising and flexible behaviours. A well-designed representation should reflect this.

ACKNOWLEDGEMENTS

We thank Cengiz Pehlevan for insightful discussions, the Parietal Team at INRIA for helpful comments on an earlier version of this work, Changmin Yu for reading a draft of this work, the Mathematics Stack Exchange user Roger Bernstein for pointing us to equation 69, and Pierre Glaser for teaching the ways of numpy array broadcasting. We thank the following funding sources: the Gatsby Charitable Foundation to W.D.; the Gatsby Charitable Foundation and Wellcome Trust (110114/Z/15/Z) to P.E.L.; Wellcome Principal Research Fellowship (219525/Z/19/Z), Wellcome Collaborator award (214314/Z/18/Z), and Jean-Franois and Marie-Laure de Clermont-Tonnerre Foundation award (JSMF220020372) to T.E.J.B.; Sir Henry Wellcome Post-doctoral Fellowship (222817/Z/21/Z) to J.C.R.W.; the Wellcome Centre for Integrative Neuroimaging and Wellcome Centre for Human Neuroimaging are each supported by core funding from the Wellcome Trust (203139/Z/16/Z, 203147/Z/16/Z).

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

# A CONSTRAINING REPRESENTATIONS WITH REPRESENTATION THEORY

Having an actionable code means the representational effect of every transformation of the variable, $\Delta \boldsymbol{x}$, can be implemented by a matrix:

$$\boldsymbol{g}(\boldsymbol{x} + \Delta \boldsymbol{x}) = \boldsymbol{T}(\Delta \boldsymbol{x})\boldsymbol{g}(\boldsymbol{x}) \tag{11}$$

In Section 2 we outlined a rough argument for how this equation leads to the following constraint on actionable representations of 2D position, that was vital for our work:

$$\boldsymbol{g}(\boldsymbol{x}) = \boldsymbol{a}_0 + \sum_{d=1}^{D} \boldsymbol{a}_d \sin(\boldsymbol{k}_d \cdot \boldsymbol{x}) + \boldsymbol{b}_d \cos(\boldsymbol{k}_d \cdot \boldsymbol{x}) \tag{12}$$

In this section, we'll repeat this argument more robustly using Group and Representation Theory. In doing so, it'll become clear how broadly our version of actionability can be used to derive clean analytic representational constraints; namely, the arguments presented here can be applied to the representation of any variable whose transformations form a group whose representations are well understood.

Sections A.1 and A.2 contain a review of the Group and Representation theory used. Section A.3 applies it to our problem.

## A.1 GROUP THEORY

A mathematical group is a collection of things (like the set of integers), and a way to combine two members of the group that makes a third (like addition of two integers, that always creates another integer), in a way that satisfies a few rules:

1. There is an identity element, which is a member of the group that when combined with any other element doesn't change it. For adding integers the identity element is $0$, since $a + 0 = a$ for all $a$.
2. Every member of the group has an inverse, defined by its property that combining an element with its inverse produces the identity element. In our integer-addition example the inverse of any integer $a$ is $-a$, since $-a + a = 0$, and $0$ is the identity element.
3. Associativity applies, which just means the order in which you perform operations doesn't matter: $(a + b) + c = a + (b + c)$

Groups are ubiquitous. We mention them here because they will be our route to understanding actionable representations - representations in which transformations are also encoded consistently. The set of transformations of many variables of interest are groups. For example, the set of transformations of 2D position, i.e. 2D translations - $\Delta \boldsymbol{x}$, is a group if you define the combination of two translations via simple vector addition, $\Delta \boldsymbol{x}_{1+2} = \Delta \boldsymbol{x}_1 + \Delta \boldsymbol{x}_2$. We can easily check that they satisfy all three of the group rules:

1. There is an identity translation: simply add $\mathbf{0}$.
2. Each 2D translation has its inverse: $-\Delta \boldsymbol{x} + \Delta \boldsymbol{x} = \mathbf{0}$
3. Associativity: $(\Delta \boldsymbol{a} + \Delta \boldsymbol{b}) + \Delta \boldsymbol{c} = \Delta \boldsymbol{a} + (\Delta \boldsymbol{b} + \Delta \boldsymbol{c})$

The same applies for the transformations of an angle, $\theta$, or of positions on a torus or sphere, and much else besides. This is a nice observation, but in order to put it to work we need a second area of mathematics, Representation Theory.

## A.2 REPRESENTATION THEORY

Groups are abstract, in the sense that the behaviour of many different mathematical objects can embody the same group. For example, the integers modulo $P$ with an addition combination rule

form a group, called $C_P$. But equally, the $P$ roots of unity ($\{e^{\frac{2\pi n}{P}}\}_{n=0}^{P-1}$) with a multiplication combination rule obey all the same rules: you can create a 1-1 correspondence between the integers 0 through $P-1$ and the $P$ roots of unity by labelling all the roots with the integer $n$ that appears in the exponent $e^{\frac{2\pi n}{P}}$, and, under their respective combination rules, all additions of integers will exactly match onto multiplication of complex roots (i.e. $e^{\frac{2\pi n_1}{P}} * e^{\frac{2\pi n_2}{P}} = e^{\frac{2\pi \bmod_P (n_1+n_2)}{P}}$)

In this work we'll be interested in a particular instantiation of our groups, defined by sets of matrices, $\boldsymbol{T}(\Delta \boldsymbol{x})$, combined using matrix multiplication. For example, a matrix version of the group $C_P$ would be the set of 2-by-2 rotation matrices that rotate by $\frac{2\pi}{P}$ increments:

$$\boldsymbol{T}(n) = \begin{pmatrix} \cos(\frac{2\pi n}{P}) & -\sin(\frac{2\pi n}{P}) \\ \sin(\frac{2\pi n}{P}) & \cos(\frac{2\pi n}{P}) \end{pmatrix} \quad \text{such that} \quad \boldsymbol{T}(n_1)\boldsymbol{T}(n_2) = \boldsymbol{T}(\bmod_P(n_1+n_2)) \quad (13)$$

Combining these matrices using matrix multiplication follows all the same patterns that adding the integers modulo P, or multiplying the P roots of unity, followed; they all embody the same group.

Representation theory is the branch of mathematics that specifies the structure of sets of matrices that, when combined using matrix multiplication, follow the rules of a particular group. Such sets of matrices are called a representation of the group; to avoid confusion arising from this unfortunate though understandable convergent terminology, we distinguish group representations from neural representations by henceforth denoting *representations* of a group in italics. We will make use of one big result from *Representation* Theory, hinted at in Section 2: the Peter-Weyl theorem (Knapp, 2002). For compact topological groups (which include transformations of a point on a circle, torus, and sphere) any matrix that is a *representation* of a group can be composed from the direct product of a set of blocks, called irreducible *representations*, or *irreps* for short, up to a linear transformation. i.e. if $\boldsymbol{T}(\Delta \boldsymbol{a})$ is a *representation* of the group of transformations of some variable $\boldsymbol{a}$:

$$\boldsymbol{T}(\Delta \boldsymbol{a}) = \boldsymbol{S} \begin{bmatrix} I_1(\Delta \boldsymbol{a}) & 0 & 0 & \dots & 0 \\ 0 & I_2(\Delta \boldsymbol{a}) & 0 & \dots & 0 \\ 0 & 0 & I_3(\Delta \boldsymbol{a}) & \dots & 0 \\ \vdots & \vdots & \vdots & \ddots & \vdots \\ 0 & 0 & 0 & \dots & I_D(\Delta \boldsymbol{a}) \end{bmatrix} \boldsymbol{S}^{-1} \quad (14)$$

where $I_d(\Delta \boldsymbol{a})$ are the *irreps* of the group in question, which are square matrices not necessarily of the same size as $d$ varies, and $\boldsymbol{S}$ is some invertible square matrix.

To motivate our current rabbit hole further, this is exactly the result we hinted towards in Section 2. We discussed $\boldsymbol{T}(\Delta \theta)$, and, in 2-dimensions, argued it was, up to a linear transform, the 2-dimensional rotation matrix. Further, we discussed how every extra two neurons allowed you to add another frequency, i.e. another rotation matrix. This was an attempt to motivate the plausibility of the Peter-Weyl theorem: the rotation matrices are the *irreps* of the group of transformations of an angle, and adding two neurons allows you to create a larger $\boldsymbol{T}(\Delta \theta)$ by stacking rotation matrices on top of one another. Now including the invertible linear transform, $\boldsymbol{S}$, we can state the 4-dimensional version of equation 4:

$$\boldsymbol{T}(\Delta \theta) = \boldsymbol{S} \begin{pmatrix} \cos(n_1 \Delta \theta) & -\sin(n_1 \Delta \theta) & 0 & 0 \\ \sin(n_1 \Delta \theta) & \cos(n_1 \Delta \theta) & 0 & 0 \\ 0 & 0 & \cos(n_2 \Delta \theta) & -\sin(n_2 \Delta \theta) \\ 0 & 0 & \sin(n_2 \Delta \theta) & \cos(n_2 \Delta \theta) \end{pmatrix} \boldsymbol{S}^{-1} \quad (15)$$

In performing this decomposition the role of the rotation matrices as *irreps* is clear. There are infinitely many types, each specified by an integer frequency, $n_d$, and their direct product produces any *representation* of the rotation group.

We can now return to actionable neural representations of periodic 2D space, a torus, where this theorem comes in very useful. We sought codes, $\boldsymbol{g}(\boldsymbol{x})$, that could be manipulated via matrices:

$$\boldsymbol{g}(\boldsymbol{x} + \Delta \boldsymbol{x}) = \boldsymbol{T}(\Delta \boldsymbol{x})\boldsymbol{g}(\boldsymbol{x}) \quad (16)$$

Now we recognise that we're asking $\boldsymbol{T}(\Delta\boldsymbol{x})$ to be a *representation* of the transformation group of $\boldsymbol{x}$, and we know the shape $\boldsymbol{T}(\Delta\boldsymbol{x})$ must take to fit this criteria. To specify $\boldsymbol{T}(\Delta\boldsymbol{x})$ from this constrained class we must simply choose a set of *irreps* that fill up the dimensionality of the neural space, stack them one on top of each other, and rotate and scale them using the matrix $\boldsymbol{S}$.

A final detail is that there is a trivial way to make a *representation* of any group: choose $\boldsymbol{T}(\theta) = \mathbb{I}_N$, the N-by-N identity matrix. This also fits our previous discussion, but corresponds to a representation made from the direct product of $N$ copies of something called the trivial *irrep*, a 1-dimensional *irrep*, $I_{\text{trivial}}(\Delta\boldsymbol{a}) = 1$.

Armed with this knowledge, the following subsection will show how this theory can be used to constrain the neural code, $\boldsymbol{g}(\boldsymbol{x})$. To finish this review subsection, we list the non-trivial *irreps* of the groups used in this work. To be specific, in all our work we use only the real-valued *irreps*, i.e all elements of $\boldsymbol{T}$ are real numbers, since firing rates are real numbers. These are less common than complex *irreps*, which is often what is implied by the simple name *irreps*.

| Transformation group of which variable | Non-trivial Real *Irreps* |
|---|---|
| An angle on a unit circle, $\theta$ | $\begin{pmatrix} \cos(n\Delta\theta) & -\sin(n\Delta\theta) \\ \sin(n\Delta\theta) & \cos(n\Delta\theta) \end{pmatrix}$ for $n \in \mathbb{Z}$ |
| Position on a very big circle $\approx$ a line, $x$ | $\begin{pmatrix} \cos(\omega\Delta\theta) & -\sin(\omega\Delta\theta) \\ \sin(\omega\Delta\theta) & \cos(\omega\Delta\theta) \end{pmatrix}$ for $\omega \in \mathbb{R}$ |
| Angles on a unit torus, $\boldsymbol{\theta}$ | $\begin{pmatrix} \cos(\boldsymbol{a} \cdot \Delta\boldsymbol{\theta}) & -\sin(\boldsymbol{a} \cdot \Delta\boldsymbol{\theta}) \\ \sin(\boldsymbol{a} \cdot \Delta\boldsymbol{\theta}) & \cos(\boldsymbol{a} \cdot \Delta\boldsymbol{\theta}) \end{pmatrix}$ for $\boldsymbol{a} \in \mathbb{Z}^2$ |
| Position on a very big torus $\approx$ plane, $\boldsymbol{x}$ | $\begin{pmatrix} \cos(\boldsymbol{k} \cdot \Delta\boldsymbol{x}) & -\sin(\boldsymbol{k} \cdot \Delta\boldsymbol{x}) \\ \sin(\boldsymbol{k} \cdot \Delta\boldsymbol{x}) & \cos(\boldsymbol{k} \cdot \Delta\boldsymbol{x}) \end{pmatrix}$ for $\boldsymbol{k} \in \mathbb{R}^2$ |
| Position on a unit sphere, $\phi$ | Real Wigner-D Matrices |

Proofs and discussions for deriving these *irreps* of the circle, torus, and sphere transformation groups can be found in any textbook on the *representation* theory of compact Lie Groups (e.g. Knapp (2002)). The step from complex to real *irreps* can be done using the Frobenius-Schur indicator, which gives a recipe for mapping from the complex *irreps* of a compact group to the real *irreps* (Fulton & Harris, 1991). The real Wigner D-Matrices were calculated recursively as detailed in Ivanic & Ruedenberg (1996; 1998), by translating matlab code from Politis et al. (2016) into python.

Our derivations of the *irreps* on the very large circle and torus are simple. In 1D the constraint that the frequencies $n$ be integers is only because $\theta$ lives on the unit circle. Then the frequencies are constrained such that after rotating by $2\pi$ the function is identical. If you change the radius of the circle to $R$ this constraint becomes $\omega_d = \frac{n_d}{R}$ where $n_d$ is any integer. As you take the circle's radius to infinity, in the process making finite sections of the circle a better and better approximation of finite patches of flat 1D space, the lattice of permitted frequencies $\omega_d$ becomes arbitrarily close together. Eventually they are separated by machine precision, and so we can simply implement the code as if they were continuous, hence $\omega_d \in \mathbb{R}$. The same argument applies analogously for 2D frequencies on a very very large torus.

## A.3 REPRESENTATIONAL CONSTRAINTS

Finally, we will translate these constraints on transformation matrices into constraints on the neural code. This is, thankfully, not too difficult. Consider the representation of an angle for simplicity, and take some arbitrary origin in the input space, $\theta_0 = 0$. The representation of all other angles can be derived via:

$$\boldsymbol{g}(\theta) = \boldsymbol{T}(\theta)\boldsymbol{g}(0) = \boldsymbol{S} \begin{bmatrix} I_1(\theta) & 0 & 0 & \dots & 0 \\ 0 & I_2(\theta) & 0 & \dots & 0 \\ 0 & 0 & I_3(\theta) & \dots & 0 \\ \vdots & \vdots & \vdots & \ddots & \vdots \\ 0 & 0 & 0 & \dots & I_D(\theta) \end{bmatrix} \boldsymbol{S}^{-1}\boldsymbol{g}(0) \tag{17}$$

In the case of an angle, each *irrep*, $I_d(\theta)$, is just a 2-by-2 rotation matrix at frequency $d$, table A.2, and for an $N$-dimensional $\boldsymbol{T}$ we can fit maximally $\frac{N}{2}$ (for $N$ even) different frequencies, where $N$ is the number of neurons. Hence the representation, $\boldsymbol{g}(\theta)$, is just a linear combination of the sine and cosine of these different frequencies, exactly as quoted previously:

$$\boldsymbol{g}(\theta) = \boldsymbol{a}_0 + \sum_{d=1}^{D} \boldsymbol{a}_d \sin(n_d\theta) + \boldsymbol{b}_d \cos(n_d\theta) \qquad \text{for integer } n_d, \tag{18}$$

$\boldsymbol{a}_0$ corresponds to the trivial *irrep*, and we include it since we know it must be in the code to make the firing rates non-negative for all $\theta$. It also cleans up the constraint on the number of allowable frequencies, for even or odd $N$ we require $D < \frac{N}{2}$. This is because if $N$ is odd one dimension is used for the trivial *irrep*, the rest can be shared up amongst $\frac{N-1}{2}$ frequencies, so $D$ must be an integer smaller than $\frac{N}{2}$. If $N$ is even, one dimension must still be used by the trivial *irrep*, so $D$ can still maximally be only the largest integer smaller than $\frac{N}{2}$.

Extending this to other groups is relatively simple, the representation is just a linear combination of the functions that appear in the *irrep* of the group in question, see table A.2. For position on a circle, line (very large circle), torus, or plane (very large torus), they all take the relatively simple form as in equation 18, but requiring appropriate changes from 1 to 2 dimensions, or integer to real frequencies. Representations of position on a sphere are slightly more complex, instead being constructed from linear combinations of sets of spherical harmonics.

### A.4 PERIODIC VS INFINITE SPACES

We finally give further details about a key step in our argument for representations of flat 2D space. We approximate a finite region of the infinite 2D plane by a finite region of a very large periodic 2D space, the torus. We do this because the *representation* theory of the group of 2D translations is not as well characterised (to the best our knowledge there is no equivalent of the Peter-Weyl theorem that could be used to provide a target for optimisation). Fortunately, any animal only cares about a finite region of 2D space (encoded in equation 2 via the lengthscale $L$). We therefore approximate this finite region of flat 2D space with an equivalently sized region of a torus. This enables us to use the fully characterised *representation* theory of the set of translations of periodic space.

Now, there are legitimate concerns over whether this is a reasonable approximation. Fortunately, we are free to choose the radii of the torus as we wish, and we make use of this freedom to make the approximation of the finite flat 2D space arbitrarily good. As the radii increase to infinity the small region of torus approximates the flat 2D space arbitrarily well.

This use of periodic space does exclude some representations of the full set of 2D translations, for example:

$$\boldsymbol{T}(\Delta\boldsymbol{x}) = \boldsymbol{S} \begin{pmatrix} 1 & 0 & \Delta x \\ 0 & 1 & \Delta y \\ 0 & 0 & 1 \end{pmatrix} \boldsymbol{S}^{-1} \tag{19}$$

There are two reasons not to be concerned by this. First, this solution would never have been allowed, as including it ensures that your representation, $\boldsymbol{g}(\boldsymbol{x})$, cannot have non-negative or bounded firing rates (we suspect this would be the case for all such additional representations of flat 2D translations). Second, any result which was dependent on these kind of boundary effects at $\infty$ would be highly suspect. After all, animals are not truly representing flat 2D space, rather they are representing an approximately flat section of the surface of a sphere (the Earth).

# B NUMERICAL OPTIMISATION DETAILS

In order to test our claims, we numerically optimise the parameters that define the representation to minimise the loss, subject to constraints. Despite sharing many commonalities, our optimisation procedures are different depending on whether the variable being represented lives in a very large periodic space (approximations to the line, plane, or volume) or finite (circle, torus, sphere). We describe each of these schemes in turn, beginning with the very large spaces. All code is available on github: `https://github.com/WilburDoz/ICLR_Actionable_Reps`.

## B.1 NUMERICAL OPTIMISATION FOR VERY LARGE SPACES

We will use the representation of a point on a line for explanation, planes or volumes are a simple extension. $\boldsymbol{g}(x)$ is parameterised as follows:

$$\boldsymbol{g}(x) = \boldsymbol{a}_0 + \sum_{d=1}^{D} \boldsymbol{a}_d \cos(\omega_d x) + \boldsymbol{b}_d \sin(\omega_d x) \qquad \boldsymbol{a}_0, \{\boldsymbol{a}_d, \boldsymbol{b}_d\}_{d=1}^{D} \in \mathbb{R}^N, \omega_d \in \mathbb{R} \qquad (20)$$

Our loss is made from three terms: the first is the functional objective, the last two enforce the non-negativity and boundedness constraint respectively:

$$\mathcal{L} = \mathcal{L}_{\text{functional}} + \lambda_{\text{p}} \mathcal{L}_{\text{non-negativity}} + \lambda_{\text{b}} \mathcal{L}_{\text{bounded}} \qquad (21)$$

To evaluate the functional component of our loss we sample a set of points $\{x_i\}_{i=1}^{M}$ from $p(x)$ and calculate their representations $\{\boldsymbol{g}(x_i)\}_{i=1}^{M}$. To make the bounded constraint approximately satisfied (there's still some work to be done, taken care of by $\mathcal{L}_{\text{bounded}}$) we calculate the following neuron-wise norm and use it to normalise each neuron's firing:

$$\|g_n\|^2 = \sum_{m=1}^{M} g_n(x_m)^2 \qquad \tilde{g}_n(\theta) = \frac{g_n(\theta)}{\|g_n\|} \qquad (22)$$

The functional form of $\mathcal{L}_{\text{functional}}$ varies, as discussed in the main paper. For example, for the full loss we compute the following using the normalised firing, a discrete approximation to equation 1:

$$\mathcal{L}_{\text{functional}} = \frac{1}{M^2} \sum_{m=1}^{M} \sum_{m'=1}^{M} e^{-\frac{\|\tilde{\boldsymbol{g}}(x_m) - \tilde{\boldsymbol{g}}(x_{m'})\|^2}{2\sigma^2}} \chi(x_m, x_{m'}) \qquad (23)$$

Now we come to our two constraints, which enforce that the representation is non-negative and bounded. We would like our representation to be reasonable (i.e. non-negative and bounded) for all values of $x$. If we do not enforce this then the optimiser finds various trick solutions in which the representation is non-negative and bounded only in small region, but negative and growing explosively outside of this, which is completely unbiological, and uninteresting. Of course, $x$ is infinite, so we cannot numerically ensure these constraints are satisfied for all $x$. However, ensuring they are true in a region local to the animal's explorations (which are represented by $p(x)$) suffices to remove most of these trick solutions. As such, we sample a second set of $M_S$ 'shift positions', $\{x_m\}_{m=1}^{M_S}$, from a scaled up version of $p(x)$, using a scale factor $S$. We then create a much larger set of positions, by shifting the original set by each of the 'shift positions', creating a dataset of size $M * M_S$, and use these to calculate our two constraint losses.

Our non-negativity loss penalises negative firing rates:

$$\mathcal{L}_{\text{non-negativity}} = \frac{1}{M M_S N} \sum_{m_s=1}^{M_S} \sum_{m=1}^{M} \sum_{n=1}^{N} |\tilde{g}_n(x_{m_s} + x_m)| \mathbb{I}(\tilde{g}_n(x_{m_s} + x_m) < 0) \qquad (24)$$

Where $\mathbb{I}$ is the indicator function, 1 if the firing rate is negative, 0 else, i.e. we just average the magnitude of the negative portion of the firing rates.

The bounded loss penalises the deviation of each neuron's norm from 1, in each of the shifted rooms:

$$\mathcal{L}_{\text{bounded}} = \frac{1}{NM_S} \sum_{n=1}^{N} \sum_{m_s=1}^{M_S} \left( \sum_{m=1}^{M} \tilde{g}_n^2(x_{m_s} + x_m) - 1 \right)^2 \tag{25}$$

That completes our specification of the losses. We minimise the full loss over the parameters $(\boldsymbol{a}_0, \{\boldsymbol{a}_d, \boldsymbol{b}_d, \omega_d\}_{d=1}^{D})$ using a gradient-based algorithm, ADAM (Kingma & Ba, 2014). We initialise these parameters by sampling from independent zero-mean gaussians, with variances as in table B.3.

The final detail of our approach is the setting of $\lambda_p$ and $\lambda_b$, which control the relative weight of the constraints vs. the functional objective. We don't want the optimiser to strike a tradeoff between the objective and constraints, since the constraints must actually be satisfied. But we also don't want to make $\lambda_p$ so large that the objective is badly minimised in the pursuit of only constraint satisfaction. To balance these demands we use GECO (Rezende & Viola, 2018), which specifies a set of stepwise updates for $\lambda_p$ and $\lambda_b$. These ensure that if the constraint is not satisfied their coefficients increase, else they decrease allowing the optimiser to focus on the loss. The dynamics that implement this are as follows, for a timestep, $t$.

GECO defines a log-space measure of how well the constraint is being satisfied:

$$L_t = \log(\mathcal{L}_t) - k \tag{26}$$

(or small variations on this), where $k$ is a log-threshold that sets the target log-size of the constraint loss in question, and $\mathcal{L}_t$ is the value of the loss at timestep $t$. $L_t$ is then smoothed:

$$\hat{L}_t = \alpha \hat{L}_t + (1 - \alpha) L_t \tag{27}$$

And this smoothed measure of how well the constraint is being satisfied controls the behaviour of the coefficient:

$$\lambda_{t+1} = \lambda_t e^{\gamma \hat{L}_t} \tag{28}$$

This specifies the full optimisation, parameters are listed in table B.3.

### B.2 Numerical Optimisation for Finite Spaces

Optimising the representation of finite variables has one added difficulty, and one simplification. Again, we have a parameterised form, e.g. for an angle $\theta$:

$$\boldsymbol{g}(\theta) = \boldsymbol{a}_0 + \sum_{d=1}^{D} \boldsymbol{a}_d \cos(n_d \theta) + \boldsymbol{b}_d \sin(n_d \theta) \qquad \boldsymbol{a}_0, \{\boldsymbol{a}_d, \boldsymbol{b}_d\}_{d=1}^{D} \in \mathbb{R}^N, n_d \in \mathbb{Z} \tag{29}$$

The key difficulty is that each frequency, $n_d$, has to be an integer, so we cannot optimise it by gradient descent. Similar problems arise for positions on a torus, or sphere. We shall spend the rest of this section outlining how we circumvent this problem. However, we do also have one major simplification. Because the variable is finite, we can easily ensure the representation is non-negative and bounded across the whole space, avoiding the need for an additional normalisation constraint. Further, for the uniform occupancy distributions we consider in finite spaces, we can analytically calculate the neuron norm, and scale the parameters appropriately to ensure it is always 1:

$$\|g_n(\theta)\|^2 = \frac{1}{2\pi} \int_{-\pi}^{\pi} g_n^2(\theta) d\theta = \|\boldsymbol{a}_0\|^2 + \frac{1}{2} \sum_{d=1}^{D} \|\boldsymbol{a}_d\|^2 + \|\boldsymbol{b}_d\|^2 \tag{30}$$

$$\tilde{g}_n(\theta) = \frac{g_n(\theta)}{\|g_n(\theta)\|} \tag{31}$$

As such, we simply sample a set of angles $\{\theta_m\}_{m=1}^M$, and their normalised representations $\{\tilde{\boldsymbol{g}}(\theta_m)\}_{m=1}^M$, and compute the appropriate functional and non-negativity loss, as in equations 23 & 24.

The final thing we must clear up is how to learn which frequencies, $n_d$, to include in the code. To do this, we make a code that contains many many frequencies, up to some cutoff $D_{\max}$, where $D_{\max}$ is bigger than $N$:

$$\boldsymbol{g}(\theta) = \boldsymbol{a}_0 + \sum_{d=1}^{D_{\max}} \boldsymbol{a}_d \cos(d\theta) + \boldsymbol{b}_d \sin(d\theta) \qquad \boldsymbol{a}_0, \{\boldsymbol{a}_d, \boldsymbol{b}_d\}_{d=1}^{D_{\max}} \in \mathbb{R}^N \tag{32}$$

We then add a term to the loss that forces the code to choose only $D$ of these $D_{\max}$ frequencies, setting all other coefficient vectors to zero, and hence making the code actionable again but allowing the optimiser to do so in a way that minimises the functional loss.

The term that we add to achieve this is inspired by the *representation* theory we used to write these constraints, Section A.2. We create first a $2D_{\max} + 1$-dimesional vector, $\mathbf{I}(\theta)$, that contains all the *irreps* i.e. each of the $D_{\max}$ sines and cosines and a constant. We then create a $(2D_{\max} + 1) \times (2D_{\max} + 1)$-dimensional transition matrix, $\boldsymbol{G}_I(\Delta\theta)$ that is a *representation* of the rotation group in this space: $\boldsymbol{G}_I(\Delta\theta)\mathbf{I}(\theta) = \mathbf{I}(\theta + \Delta\theta)$. $\boldsymbol{G}_I$ can be simply made by stacking 2-by-2 rotation matrices. Then we create the neural code by projecting the frequency basis through a rectangular weight matrix, $\boldsymbol{W}$: $\boldsymbol{g}(\theta) = \boldsymbol{W}\mathbf{I}(\theta)$. Finally, we create the best possible *representation* of the group in the neural space:

$$\boldsymbol{G}(\Delta\theta) = \boldsymbol{W}\boldsymbol{G}_I(\Delta\theta)\boldsymbol{W}^* \tag{33}$$

Where $\boldsymbol{W}*$ denotes the Moore-Penrose pseudoinverse. We then learn $\boldsymbol{W}$, which is equivalent to learning $\boldsymbol{a}_0$ and $\{\boldsymbol{a}_d, \boldsymbol{b}_d\}_{d=1}^{D_{\max}}$.

*Representation* theory tells us this will not do a perfect job at rotating the neural representation, unless the optimiser chooses $\boldsymbol{W}$ to cut out all but $D$ of the frequencies. As such, we sample a set of shifts $\{\theta_{m_s}\}_{m_s=1}^M$, and measure the following discrepancy:

$$\mathcal{L}_{\text{Transformation}} = \frac{1}{M_S M} \sum_{m_S=1}^{M_S} \sum_{m=1}^{M} \|\boldsymbol{G}(\theta_{m_s})\boldsymbol{g}(\theta_m) - \boldsymbol{g}(\theta_m + \theta_{m_s})\|^2 \tag{34}$$

Minimising it as we minimised the other constraint terms will force the code to choose $D$ frequencies and hence make the code actionable.

Since calculating the pseudoinverse is expensive, we replace $\boldsymbol{W}^*$ with a matrix $\boldsymbol{B}$ that we also learn by gradient descent. $\boldsymbol{B}$ quickly learns to approximate the pseudoinverse, speeding our computations.

That wraps up our numerical description. We are again left with three loss terms, two of which enforce constraints. This can be applied to any group, even if you can't do gradient descent through their *irreps*, at the cost of limiting the optimisation to a subset of *irreps* below a certain frequency, $D_{\max}$.

$$\mathcal{L} = \mathcal{L}_{\text{functional}} + \lambda_{\text{p}}\mathcal{L}_{\text{non-negativity}} + \lambda_{\text{T}}\mathcal{L}_{\text{Transformation}} \tag{35}$$

## B.3 PARAMETERS VALUES

We list the parameter values used to generate the grid cells in Figure 2B, and show the full population of neurons in figure 6.

| Parameter | Meaning | Value |
|---|---|---|
| $\sigma$ | neural lengthscale | 0.2 |
| $l$ | $\chi$ lengthscale | 0.5 |
| $T$ | number of gradient steps | 150000 |
| $N$ | number of neurons | 64 |
| $M$ | number of sampled points every $n_{\text{resample}}$ steps | 150 |
| $M_S$ | number of room shifts sampled every $n_{\text{resample}}$ steps | 15 |
| $S$ | standard deviation of normal for shift sampling | 3 |
| $n_{\text{resample}}$ | number of steps per resample of points | 5 |
| $\lambda_{p0}$ | initial positivity weighting coefficient | 0.1 |
| $k_p$ | log positivity target | -9 |
| $\alpha_p$ | positivity target smoothing | 0.9 |
| $\gamma_p$ | positivity coefficient dynamics coefficient | 0.0001 |
| $\lambda_{n0}$ | same set of GECO parameters for norm constraint | 0.005 |
| $k_n$ | ditto | 4 |
| $\alpha_n$ | ditto | 0.9 |
| $\gamma_n$ | ditto | 0.0001 |
| $\epsilon_w,$ | coefficient gradient step size | 0.1 |
| $\epsilon_{\text{om}}$ | frequency gradient step size | 0.1 |
| $\beta_1$ | exponential moving average of first gradient moment | 0.9 |
| $\beta_2$ | exponential moving average of second moment | 0.9 |
| $\eta$ | ADAM non-exploding term | $1 \times 10^{-8}$ |

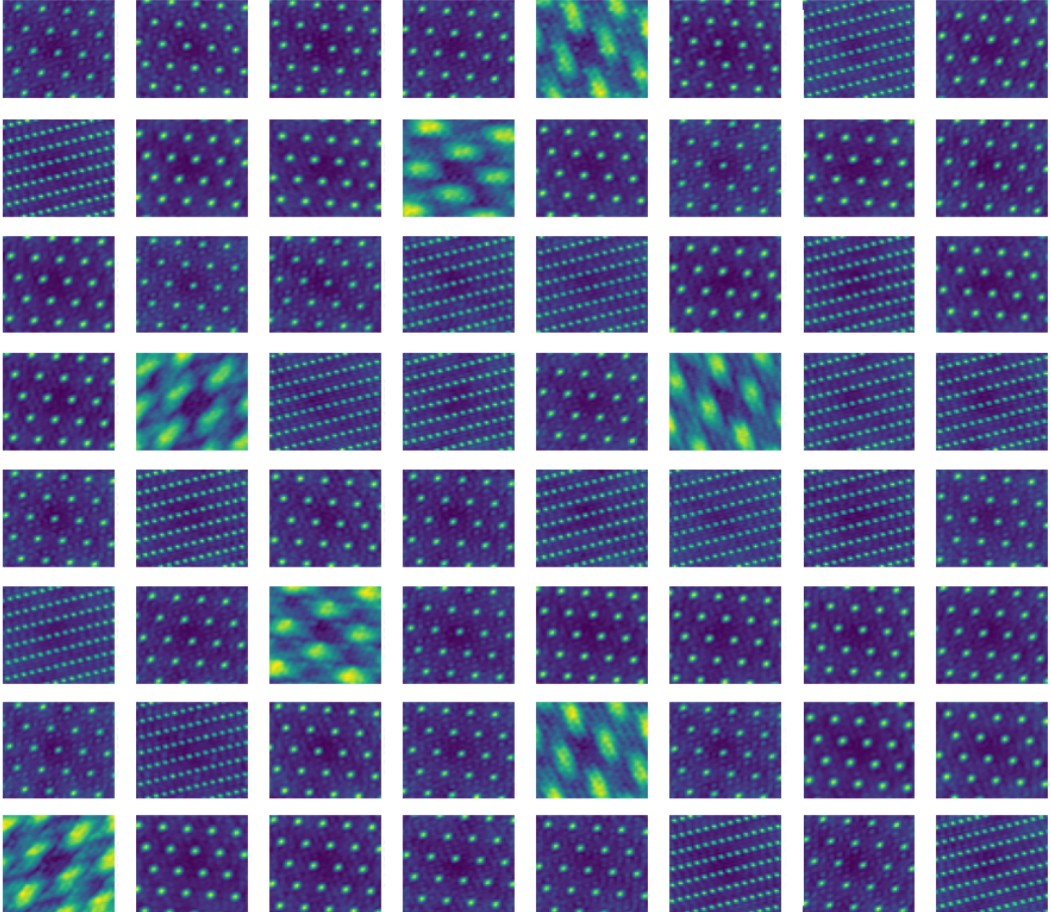

Figure 6: All the neurons for the population in figure 2: all neurons fall into one of three modules.

### B.4 ROBUSTNESS TO PARAMETER VALUES

In this section we explore the parameter-dependence of our numerical solutions. Most of the parameters in table B.3 control the behaviour of the optimisation scheme (for example controlling the behaviour of ADAM (Kingma & Ba, 2014) or GECO (Rezende & Viola, 2018)). These were tuned so that the optimiser found the best solutions possible; any parameter dependence in these does not seem deeply worrying, reflecting the optimisers used rather than the core problem we're solving.

We therefore focus our explorations on the parameters that define key parts of the loss function: the neural lengthscale, $\sigma$, the two spatial lengthscales, $L$ and $l$, and the number of neurons, $N$. We choose the units of the input space such that $L = 1$, leaving us with three parameters to explore. (The neural space units are not so arbitrary, due to the firing rate constraint)

Our theory actually makes predictions for how these parameters should change the representation. As discussed in appendix E.3, if the ratio of $N$ and $\sigma$ is small there should be one module of neurons, and increasing it should increase the number of modules in the optimal solutions. $l$ enforces the push to hexagons, so, assuming there's only one module for simplicity, if $l$ is sufficiently large the optimal solution should be hexagonal grid cells. If $l$ is very small then it will have no effect, so all sufficiently high frequency grids should be equal and their shape should matter less.

We verify these claims in a series of numerical experiments, and we show that there are reasonably large parameter regimes in which our suggested qualitative solutions emerge (one module for high $\sigma$, as in figure 4, multiple modules for low $\sigma$, as in figure 2). In these experiments most other parameters are kept at the values in table B.3, barring the number of steps which was varied with the number of neurons, and $\lambda_{p0}$ and $\lambda_{n0}$ which should be scaled with the approximate size of the loss, that varies as $\sigma$ and $l$ vary.

### B.4.1 EFFECT OF VARYING $l$

The top 6 panels of figure 8 show that for a fixed number of neurons one module of hexagonal grids is optimal for a range of $l$. We start at $l = 1$, as that corresponds to the rough lengthscale of the environment. Eventually for small $l$ the optimiser chooses other, non-hexagonal grids. While hexagons are optimal for a reasonable range of $l$, we might wonder why our arguments do not hold for even smaller $l$?

We believe this is due to the small number of neurons we are using. We argued hexagons were optimal because they packed the most frequencies into a Goldilocks annulus in frequency space, figure 4. As $l$ decreases the outer ring of this annulus moves further away, providing a larger Goldilocks region. When the number of neurons in a module, i.e. the number of frequencies, is small this permits many different lattices to pack frequencies within the Goldilocks annulus equally well, figure 7. So, for a given number of neurons, there is a $l$-threshold, beyond which there is no longer a push towards hexagons, and this threshold can be decreased by increasing the number of neurons.

We verify this in the last panel of figure 8, where we show that at $l = 0.1$, where 64 neuron modules were not hexagonal, 100 neuron modules were. Therefore, we expect for modules containing many neurons (like those in the brain) hexagons are optimal for a much larger range of $l$.

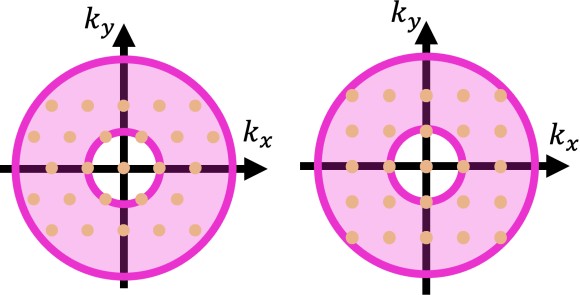

Figure 7: For small $l$, relative to the number of frequencies, many different lattices pack frequencies into the Goldilocks annulus equally well, e.g. these square and hexagonal lattice.

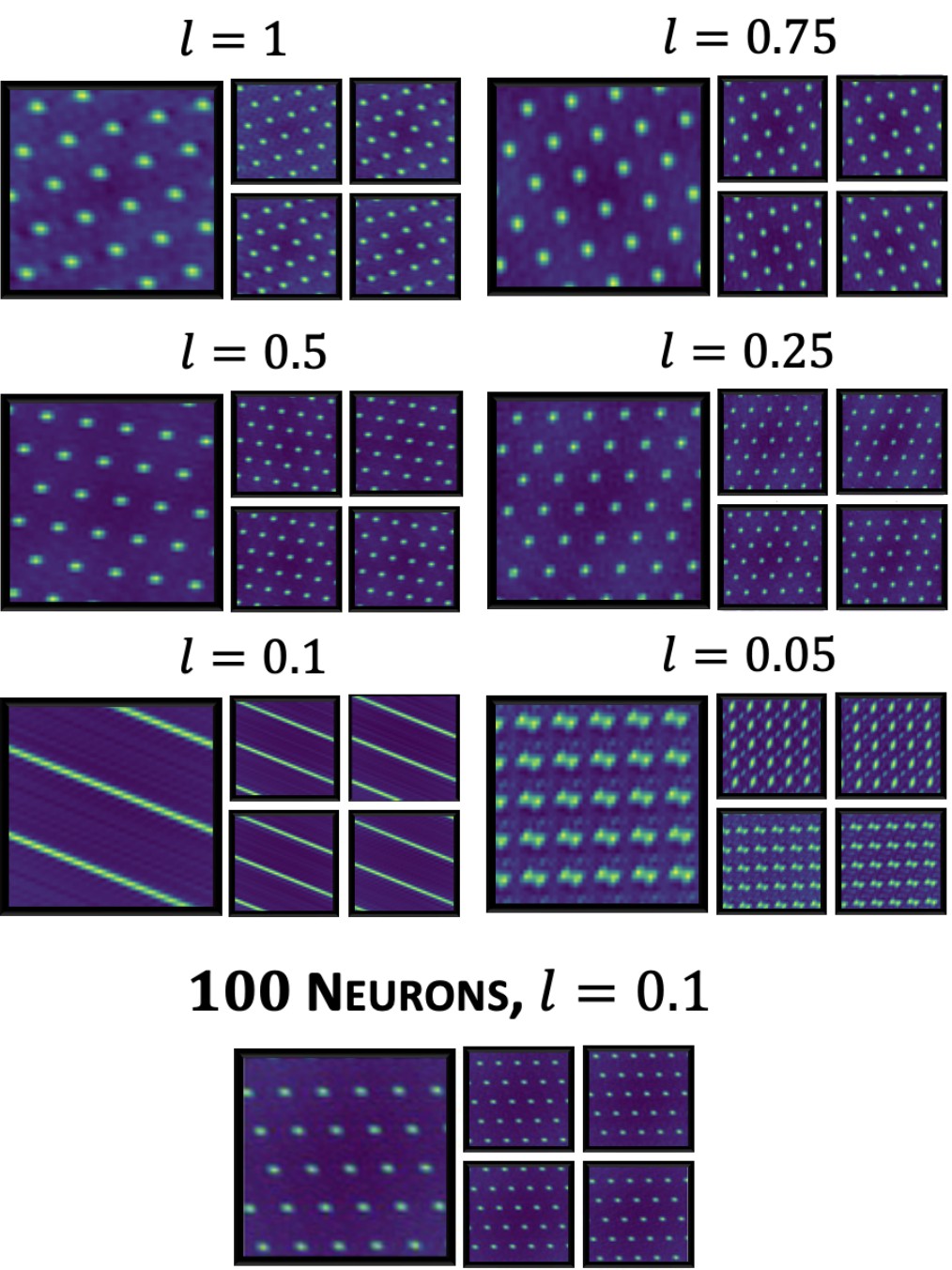

Figure 8: For 64 neurons with $\sigma = 0.4$ one module of grids is consistently optimal, all representations contained one module with occasionally one stray neuron with a garbled response. At high $l$ the modules are all hexagonal, for low $l$ other grids start appearing, like band cells or square grids. This is likely due to the small number of neurons used, because when we increase the number of neurons to 100 as in the bottom plot the module remains hexagonal at lower values of $l$.

### B.4.2   EFFECT OF VARYING $\sigma$

As discussed in Appendix E.3, varying $\sigma$ varies the push towards non-harmonicity, and hence how many modules we would expect in the optimal solution. We confirm this in Figure 9. For large $\sigma$ (up to $\sigma = 1$) we get one hexagonal module, decreasing it leads to solutions with more modules.

The more modules the less hexagonal they are. We suspect this is again partly a finite neuron effect, as in Appendix B.4.1. Future work could usefully explore whether more constraints are needed to robustly generate many hexagonal modules, or whether more neurons is enough as it was in figure 8.

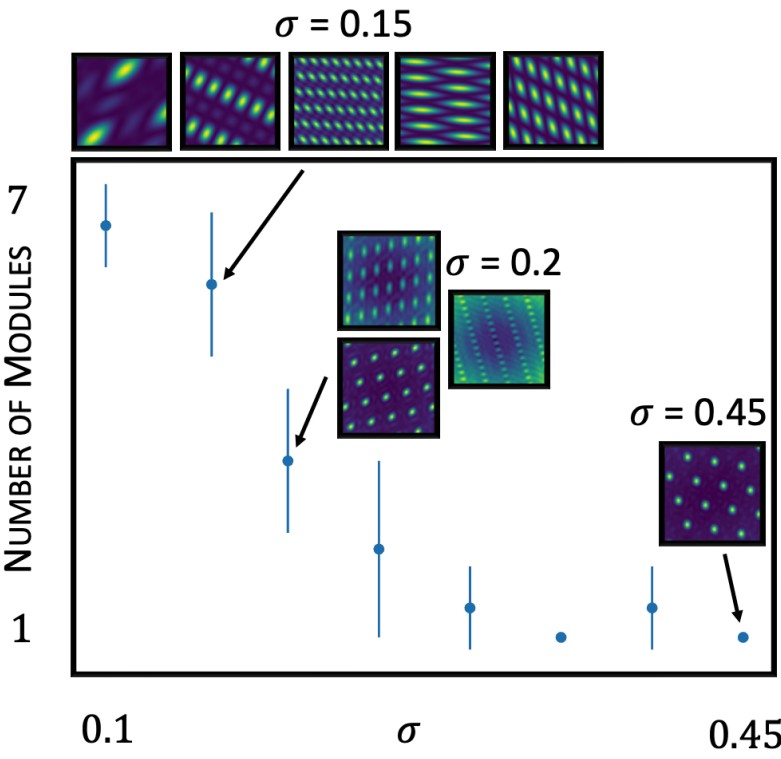

Figure 9: For fixed $l = 0.2$ and $N = 64$ we vary $\sigma$ and count the number of modules. For large $\sigma$ (including up to $\sigma = 1$) there is consistently one hexagonal module. As $\sigma$ decreases you get more modules. Error bars are standard deviation over multiple simulations. Inset are the module shapes in one simulation for each of three $\sigma$ values.

### B.4.3   VARYING NUMBER OF NEURONS

Figures 10 and  11 show that as we vary the number of neurons (with fixed $l$), we can find values of $\sigma$ for which the population produces either one module of hexagonal grids (Figure 10) or multiple modules (Figure 11). Hence, our qualitative solution types are found across a range of population sizes.

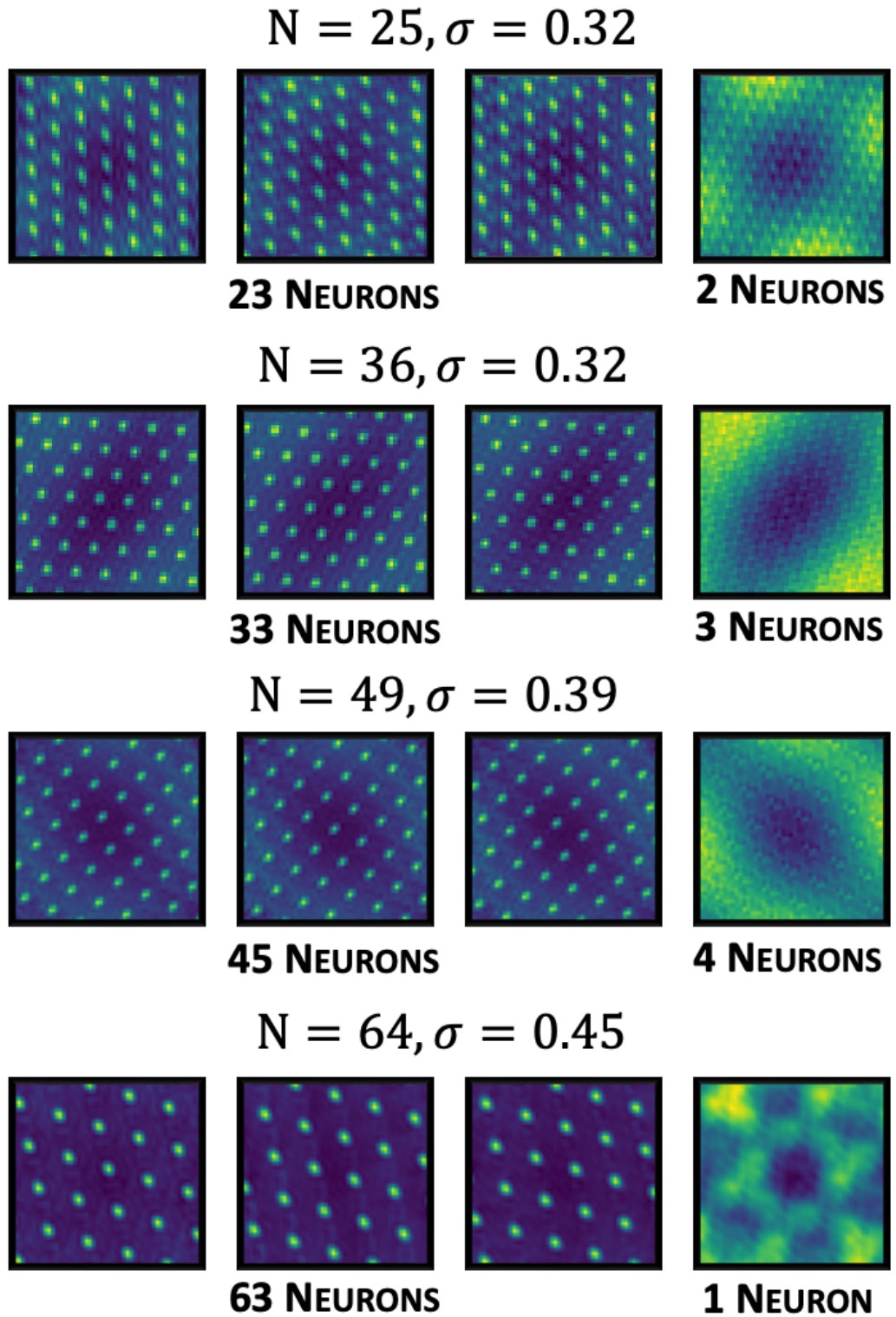

Figure 10: Each row of this figure summarises one simulation, and each simulation has a different population size, $N$. We fix $l = 0.5$ and show a $\sigma$ value at which almost all neurons form one hexagonal module.

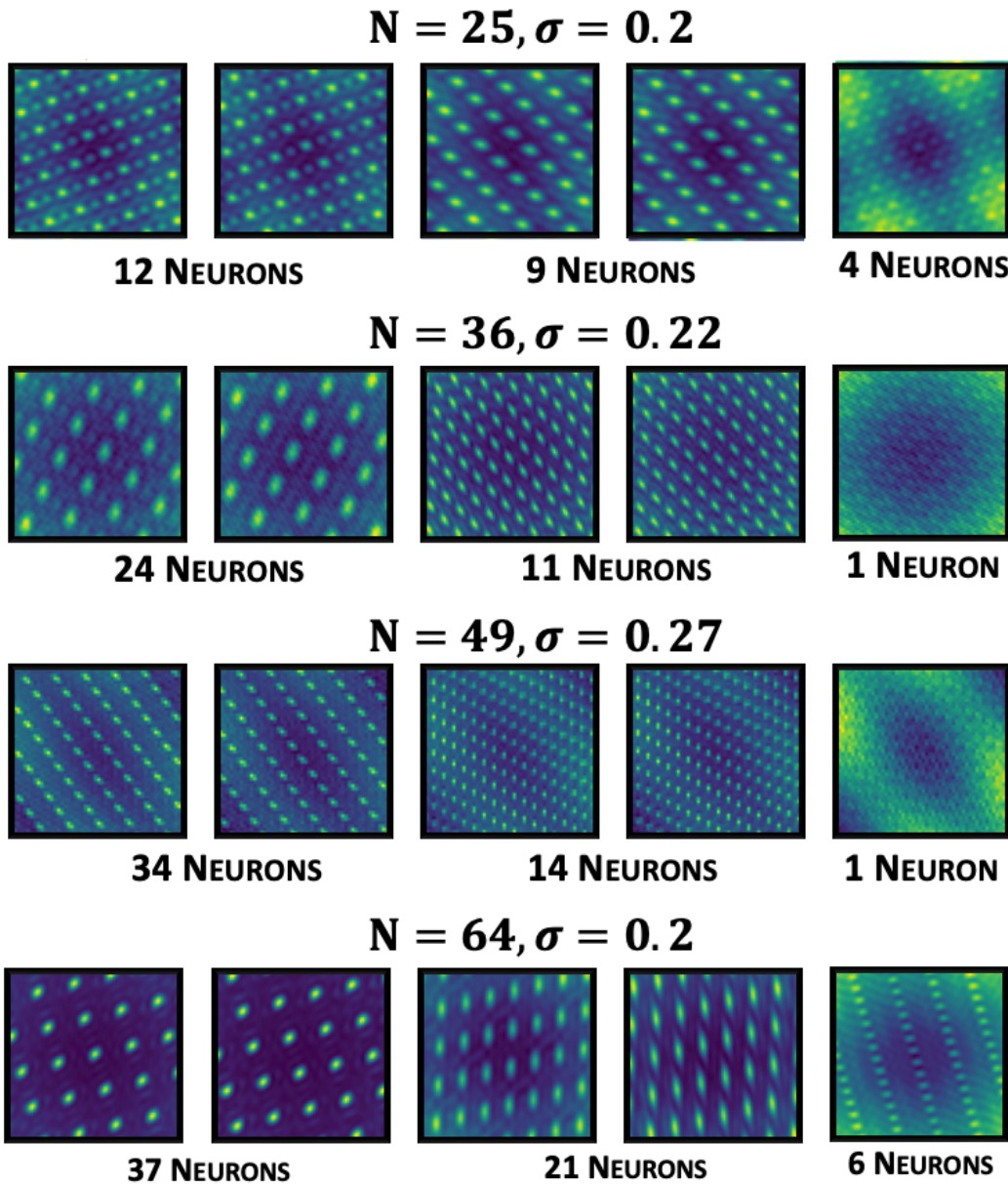

Figure 11: Each row of this figure summarises one simulation, and each simulation has a different population size, $N$. We fix $l = 0.5$ and show a $\sigma$ value at which the population falls into multiple modules.

## C  ANALYSIS OF SIMPLEST LOSS THAT LEADS TO ONE LATTICE MODULE

In this section we analytically study the simplified loss suggested in section 3.1 and derive equation 8. The actionability constraint tells us that:

$$\boldsymbol{g}(\theta) = \boldsymbol{a}_0 + \sum_{d=1}^{D} \boldsymbol{a}_d \sin(n_d\theta) + \boldsymbol{b}_d \cos(n_d\theta) \qquad n_d \in \mathbb{Z} \tag{36}$$

Where D is smaller than half the number of neurons (Appendix A). We ensure $n_d \neq n_{d'}$ if $d \neq d'$ by combining like terms. Our loss is:

$$\mathcal{L}_0 = -\frac{1}{4\pi^2} \iint_{-\pi}^{\pi} \|\boldsymbol{g}(\theta) - \boldsymbol{g}(\theta')\|^2 d\theta d\theta' \tag{37}$$

Developing this and using trig identities for the sum of sines and cosines:

$$\mathcal{L}_0 = -\frac{1}{4\pi^2} \iint_{-\pi}^{\pi} \|\boldsymbol{g}(\theta) - \boldsymbol{g}(\theta')\|^2 d\theta d\theta' = -\frac{1}{4\pi^2} \sum_n \iint_{-\pi}^{\pi} (g_n(\theta) - g_n(\theta'))^2 d\theta d\theta' \tag{38}$$

$$= -\sum_n \iint_{-\pi}^{\pi} \left(\sum_d a_{n,d} \cos\left[\frac{n_d(\theta+\theta')}{2}\right] \sin\left[\frac{n_d(\theta-\theta')}{2}\right] - b_{n,d} \sin\left[\frac{n_d(\theta-\theta')}{2}\right] \sin\left[\frac{n_d(\theta-\theta')}{2}\right]\right)^2 \frac{d\theta d\theta'}{\pi^2} \tag{39}$$

We now change variables to $\alpha = \frac{\theta-\theta'}{2}$ and $\beta = \frac{\theta+\theta'}{2}$. This introduces a factor of two from the Jacobian ($\frac{1}{2}$), which we cancel by doubling the range of the integral while keeping the same limits, despite the change of variable, Figure 12. This gives us:

$$= -\frac{1}{\pi^2} \sum_n \iint_{-\pi}^{\pi} \left(\sum_d a_{n,d} \cos n_d\beta \sin n_d\alpha - b_{n,d} \sin n_d\beta \sin n_d\alpha\right)^2 d\alpha d\beta \tag{40}$$

$$= -\frac{1}{\pi^2} \sum_{n,d,d'} \iint_{-\pi}^{\pi} \Big[ a_{n,d}a_{n,d'} \sin(n_d\alpha) \sin(n_{d'}\alpha) \cos(n_d\beta) \cos(n_{d'}\beta)$$
$$- a_{n,d}b_{n,d'} \sin(n_d\alpha) \sin(n_{d'}\alpha) \cos(n_d\beta) \sin(n_{d'}\beta)$$
$$- b_{n,d}a_{n,d'} \sin(n_d\alpha) \sin(n_{d'}\alpha) \sin(n_d\beta) \cos(n_{d'}\beta)$$
$$+ b_{n,d}b_{n,d'} \sin(n_d\alpha) \sin(n_{d'}\alpha) \sin(n_d\beta) \sin(n_{d'}\beta) \Big] d\alpha d\beta \tag{41}$$

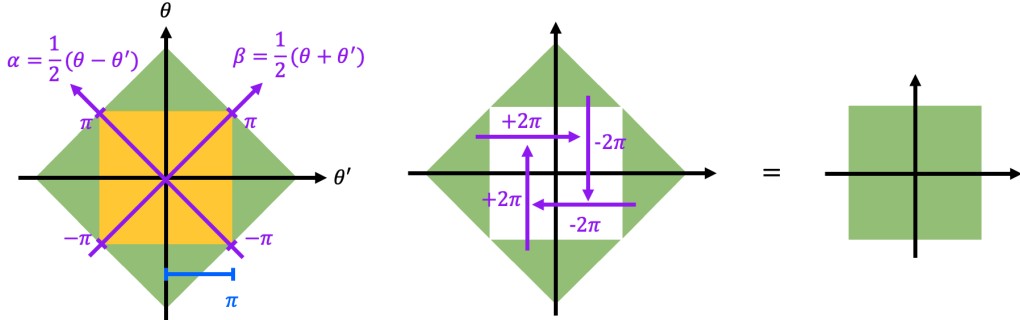

Figure 12: The original integral is shown in orange. We rotate to the purple axes and, for convenience, use the same limits (i.e. $-\pi$ to $\pi$). This corresponds to adding the green area to the integral but, using the fact that these functions are unchanged by shifts of $2\pi$ in either $\theta$ or $\theta'$ we can see that the green region is equal to the original orange region. Therefore, performing the full purple integral just gets us twice our desired integral.

Performing these integrals is easy using the fourier orthogonality relations; the two cross terms with mixed $\sin$ and $\cos$ of $\beta$ are instantly zero, the other two terms are non-zero only if $n_d = n_{d'}$, i.e. $d = d'$. When $d = d'$ these integrals evaluate to $\pi^2$, giving:

$$\mathcal{L}_0 = -\sum_{n,d}(a_{n,d}^2 + b_{n,d'}^2) \tag{42}$$

Exactly as quoted in equation 7. We can do the same development for the firing rate constraint:

$$\frac{1}{2\pi}\int_{-\pi}^{\pi} g_i^2(\theta)d\theta = a_{0,i}^2 + \frac{1}{2}\sum_{d=1}^{D} a_{d,i}^2 + b_{d,i}^2 = 1, \qquad \forall i = 1,...,N \tag{43}$$

Again using the orthogonality relation. These are the constraints that must be enforced. For illustration it is useful to see one slice of the constraints made by summing over the neuron index:

$$N = \|\boldsymbol{a}_0\|^2 + \frac{1}{2}\sum_{d=1}^{D}\|\boldsymbol{a}_d\|^2 + \|\boldsymbol{b}_d\|^2 \tag{44}$$

This is the constraint shown in equation 8, and is strongly intuition guiding. But remember it is really a stand-in for the $N$ constraints, one per neuron, that must each be satisfied!

This analysis can be generalised to derive exactly the same result in 2D by integrating over pairs of points on the torus. It yields no extra insight, and is no different, so we just quote the result:

$$\mathcal{L}_0 = -\frac{1}{16\pi^4}\int\iint\int_{-\pi}^{\pi}\|\boldsymbol{g}(\boldsymbol{x}) - \boldsymbol{g}(\boldsymbol{x}')\|^2 d\boldsymbol{x}d\boldsymbol{x}' = -\sum_{d}\|\boldsymbol{a}_d\|^2 + \|\boldsymbol{b}_d\|^2 \tag{45}$$

# D  ANALYSIS OF PARTIALLY SIMPLIFIED LOSS THAT LEADS TO A MODULE OF HEXAGONAL GRIDS

In this section we will study an intermediate loss: it will compute the euclidean distance, like in Appendix C, but we'll include $\chi$ and $p$. We'll show that they induce a frequency bias, $\chi$ for low frequencies, and $p$ for high, which together lead to hexagonal lattices. We'll study $\chi$ first for representations of a circular variable, then we'll study $p$ for representations of a line, then we'll combine them. At each stage generalising to 2-dimensions is a conceptually trivial but algebraically nasty operation with no utility, so we do not detail here. In Appendix L we will return to the differences between 1D and 2D, and show instances where they become important.

## D.1  LOW FREQUENCY BIAS: PLACE CELLS ON THE CIRCLE

We begin with the representation of an angle on a circle, introduce $\chi$, and show the analytic form of the low frequency bias it produces. Since we're on a circle $\chi$ must depend on distance on the circle, rather than on a line (i.e. not $\chi = 1 - e^{-\frac{\|x-x'\|^2}{2l^2}}$). We define the obvious generalisation of this to periodic spaces, Figure 13A, which is normalised to integrate to one under a uniform $p(\theta)$:

$$\chi(\theta,\theta') = \frac{e^k - e^{k\cos(\theta-\theta')}}{e^k - I_0(k)} \tag{46}$$

Where $I_0$ is the zeroth-order modified Bessel function. Hence the loss is:

$$\mathcal{L}_1 = -\frac{1}{4\pi^2}\iint_{-\pi}^{\pi}\|\boldsymbol{g}(\theta) - \boldsymbol{g}(\theta')\|^2 \frac{e^k - e^{k\cos(\theta-\theta')}}{e^k - I_0(k)}d\theta d\theta' \tag{47}$$

In this expression $k$ is a parameter that controls the generalisation width, playing the inverse role of $l$ in the main paper. Taking the same steps as in Appendix C we arrive at:

$$= -\frac{1}{\pi^2}\sum_{n}\iint_{-\pi}^{\pi}(\sum_{d} a_{n,d}\cos n_d\beta\sin n_d\alpha - b_{n,d}\sin n_d\beta\sin n_d\alpha)^2\frac{e^k - e^{k\cos(2\alpha)}}{e^k - I_0(k)}d\alpha d\beta \tag{48}$$

The $\beta$ integral again kills the cross terms:

$$= -\frac{1}{\pi} \sum_{n,d} (a_{n,d}^2 + b_{n,d}^2) \int_{-\pi}^{\pi} \sin^2(n_d\alpha) \frac{e^k - e^{k\cos(2\alpha)}}{e^k - I_0(k)} d\alpha \tag{49}$$

These integrals can be computed by relating them to modified Bessel functions of the first kind, which can be defined for integer $n$ as:

$$I_n(k) = \frac{1}{2\pi} \int_{-\pi}^{\pi} e^{k\cos(\theta)} \cos(n\theta) d\theta \tag{50}$$

Hence the loss is:

$$\mathcal{L}_1 = -\sum_{n,d} (a_{n,d}^2 + b_{n,d}^2) \frac{e^k + I_{n_d}(k)}{e^k - I_0(k)} \tag{51}$$

This is basically the same result as before, equation 42, all the frequencies decouple and decrease the loss in proportion to their magnitude. However, we have added a weighting factor, that decreases with frequency, Figure 13B, i.e. a simple low frequency bias!

This analysis agrees with that of Sengupta et al. (2018), who argued that the best representation of angles on a ring for a loss related to the one studied here should be place cells. We disentangle the relationship between our two works in Appendix F. Regardless, our optimisation problem now asks us to build a module of lattice neurons with as many low frequencies as possible. The optimal solution to this is the lowest frequency lattice, i.e. place cells! (we validate this numerically in Appendix M). Therefore, it is only additional developments of the loss, namely the introduction of an infinite space for which you prioritise the coding of visited regions, and the introduction of the neural lengthscale, $\sigma$, that lead us to conclude that modules of grid cells are better than place cells.

### D.2   HIGH FREQUENCY BIAS: OCCUPANCY DISTRIBUTION

Now we'll examine how a Gaussian $p(x)$ affects the result, using a representation of $x$, a 1-dimensional position:

$$\mathcal{L}_1 = -\iint_{-\infty}^{\infty} \|\boldsymbol{g}(x) - \boldsymbol{g}(x')\|^2 p(x)p(x')dxdx' = -\frac{1}{2\pi L^2} \iint_{-\infty}^{\infty} \|\boldsymbol{g}(x) - \boldsymbol{g}(x')\|^2 e^{-\frac{x^2 + x'^2}{2L^2}} dxdx' \tag{52}$$

Developing the argument as before:

$$= -\frac{4}{\pi L^2} \sum_n \iint_{-\infty}^{\infty} \left(\sum_d a_{n,d}\cos\omega_d\beta\sin\omega_d\alpha - b_{n,d}\sin\omega_d\beta\sin\omega_d\alpha\right)^2 e^{-\frac{\alpha^2+\beta^2}{L^2}} d\alpha d\beta \tag{53}$$

The cross terms again are zero, due to the symmetry of the $\beta$ integral around 0.

$$= -\frac{4}{\pi L^2} \sum_{n,d,d'} a_{n,d}a_{n,d'} C(\omega_d, \omega_{d'}\|L) S(\omega_d, \omega_{d'}\|L) + b_{n,d}b_{n,d'}S^2(\omega_d, \omega_{d'}\|L) \tag{54}$$

Where we've wrapped up the details into the following integrals of trigonometric functions over a gaussian measure:

$$C(a, b\|L) = \int_{-\infty}^{\infty} \cos(ax)\cos(bx)e^{-\frac{x^2}{L^2}} dx = \frac{L\sqrt{\pi}}{2}(e^{-\frac{L^2(a+b)^2}{4}} + e^{-\frac{L^2(a-b)^2}{4}}) \tag{55}$$

$$S(a, b\|L) = \int_{-\infty}^{\infty} \sin(ax)\sin(bx)e^{-\frac{x^2}{L^2}} dx = \frac{L\sqrt{\pi}}{2}(e^{-\frac{L^2(a-b)^2}{4}} - e^{-\frac{L^2(a+b)^2}{4}}) \tag{56}$$

So we get our final expression for the loss:

$$\mathcal{L}_1 = -\sum_{n,d,d'} a_{n,d}a_{n,d'} \underbrace{(e^{-\frac{L^2(\omega_d-\omega_{d'})^2}{2}} - e^{-\frac{L^2(\omega_d+\omega_{d'})^2}{2}})}_{\text{Sine Weighting}} + b_{n,d}b_{n,d'} \underbrace{(e^{-\frac{L^2(\omega_d-\omega_{d'})^2}{4}} - e^{-\frac{L^2(\omega_d+\omega_{d'})^2}{4}})^2}_{\text{Cosine Weighting}} \tag{57}$$

$$= -\sum_{d,d'} \left[ (e^{-\frac{L^2(\omega_d-\omega_{d'})^2}{2}} - e^{-\frac{L^2(\omega_d+\omega_{d'})^2}{2}}) \sum_n a_{n,d}a_{n,d'} + (e^{-\frac{L^2(\omega_d-\omega_{d'})^2}{4}} - e^{-\frac{L^2(\omega_d+\omega_{d'})^2}{4}})^2 \sum_n b_{n,d}b_{n,d'} \right] \tag{58}$$

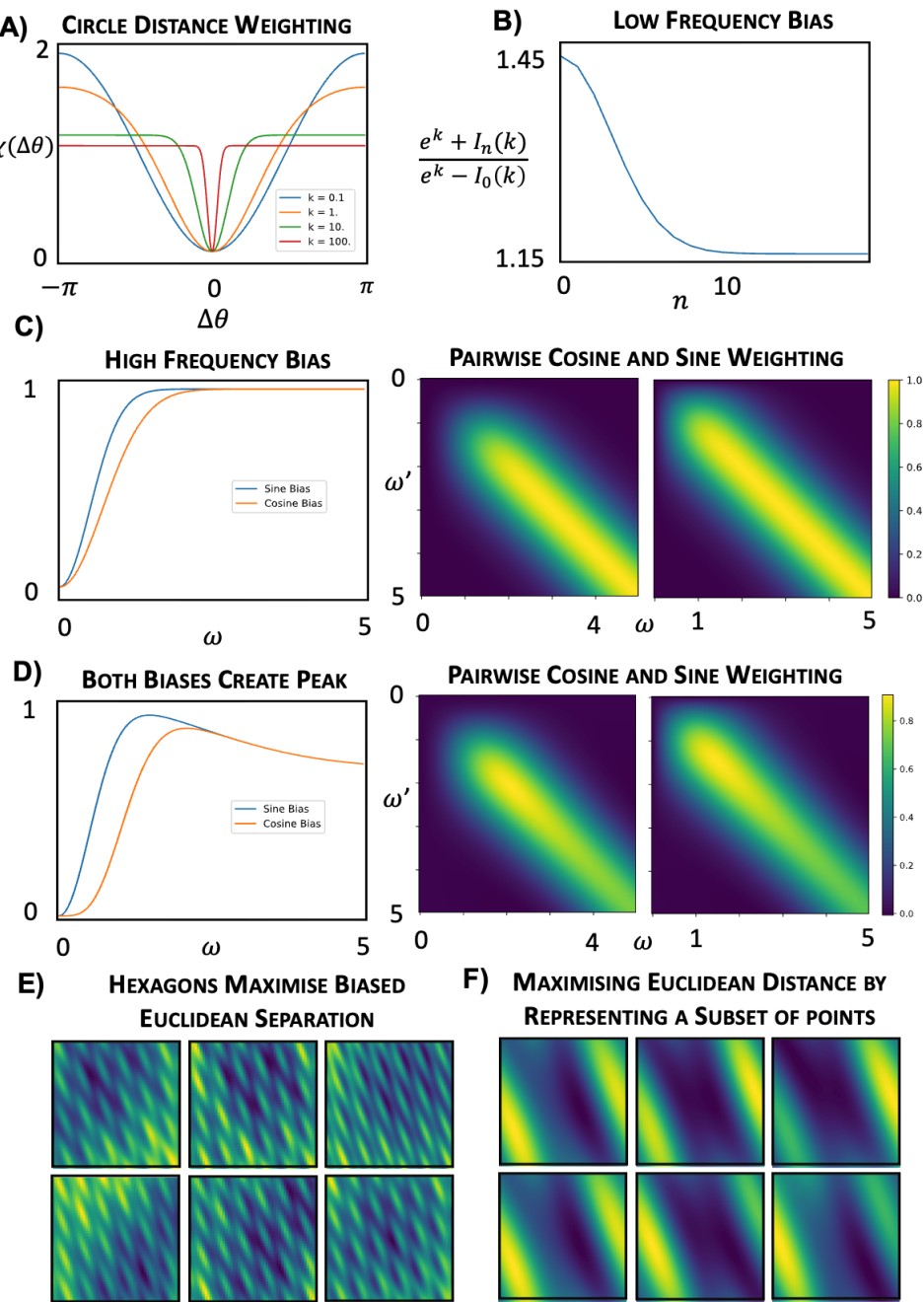

Figure 13: **A** The circular version of $\chi(\Delta\theta)$, equation 46, weights the importance of separating different pairs on the circle. downweighting nearby points, closer than $\frac{1}{k}$. **B** The Low Frequency Bias of equation 51 **C** High Frequency Bias of equation 57 and 59. Left: bias for isolated frequency (e.g. all frequencies when space is encoded uniformly). Right: cosine and sine bias for a pair of frequencies $\omega$ and $\omega'$. Diagonals of right = left. Right plots show both a high frequency bias and a smoothing over similar frequencies, where similar means within $\frac{1}{L}$. (Here $L = 1$) **D** High and Low Frequency Biases. Same as **C** for equation. 61. Notice peak around the inverse room size, $\frac{1}{L} = 1$. **E** Optimising a neural code on equation 60 with constraints produces one module of hexagons, as predicted. But, as shown in **F**, each neuron activates in the same way, hence this is a bad representation.

The downstream consequence of this loss is simple if a module encodes all areas of space evenly. Encoding evenly implies the phases of the neurons in the module are evenly distributed. The phase of a particular neuron is encoded in the coefficients of the sine and cosine of the base frequency, $a_{n1}$ and $b_{n1}$. Even distribution implies that $a_{n1}$ oscillates from $A_d$, where $A_d = \max_n a_{n1}$, to $-A_d$ and back again as you sweep the neuron index. As you perform this sweep the subsidiary coefficients, such as $a_{n2}$, perform the same oscillation, but multiple times. $a_{n2}$ does it twice, since, by the arguments that led us to grids, the peaks of the two waves must align (Figure 3B), and that implies the peak of the wave at twice the frequency moves at the same speed as you sweep the neuron number. This implies its phase oscillates twice as fast, and hence that $\sum_n a_{n1}a_{n2} = 0$. Arguments like this imply the frequencies actually decouple for a uniform encoding of space, making the loss,

$$\mathcal{L}_1 = -\sum_{d,n} \left[ (1 - e^{-2L^2\omega_d^2})a_{n,d}^2 + (1 - e^{-L^2\omega_d^2})^2 b_{n,d}^2 \right] \tag{59}$$

This is exactly a high frequency bias! (Fig 13C Left)

However, uniform encoding of the space is actually not optimal for this loss, as can be seen either from the form of equation 57, and verified numerically using the loss in the next section, Figure 13E-F. The weighting of the sine factor and the cosine factor differ, and in fact push toward sine, since it plateaus at lower frequencies. Combine that with a low frequency push, and you get an incentive to use only sines, which is allowed if you can encode only some areas of space, as the euclidean loss permits.

A final interesting feature of this loss is the way the coupling between frequencies gracefully extends the decoupling between frequencies to the space of continuous (rather than integer) frequencies. In our most simplified loss, equation 42, each frequency contributed according to the square of its amplitude, $\|\boldsymbol{a}_d\|^2 + \|\boldsymbol{b}_d\|^2$. How should this be extended to continuous frequencies? Certainly, if two frequencies were very far from one another we should retrieve the previous result, and their contributions would be expected to decouple, $\|\boldsymbol{a}_1\|^2 + \|\boldsymbol{b}_1\|^2 + \|\boldsymbol{a}_2\|^2 + \|\boldsymbol{b}_2\|^2$. But if $\boldsymbol{k}_2 = \boldsymbol{k}_1 + \delta\boldsymbol{k}$, for a small $\delta\boldsymbol{k}$ then the two frequencies are, for all intents and purposes, the same, so they should contribute an amplitude $\|\boldsymbol{a}_1 + \boldsymbol{a}_2\|^2 + \|\boldsymbol{b}_1 + \boldsymbol{b}_2\|^2 \neq \|\boldsymbol{a}_1\|^2 + \|\boldsymbol{b}_1\|^2 + \|\boldsymbol{a}_2\|^2 + \|\boldsymbol{b}_2\|^2$. Equation 57, Figure 13C right, performs exactly this bit of intuition. It tells us that the key lengthscale in frequency space is $\frac{1}{L}$. Two frequencies separated by more than this distance contribute in a decoupled way, as in equation 37. Conversely, at very small distances, much smaller than $\frac{1}{L}$ the contribution is exactly $\|\boldsymbol{a}_1 + \boldsymbol{a}_2\|^2 + \|\boldsymbol{b}_1 + \boldsymbol{b}_2\|^2$, as it should be. But additionally, it tells us the functional form of the behaviour between these two limits: a Gaussian decay.

### D.3 THE COMBINED EFFECT = HIGH AND LOW FREQUENCY BIAS

In this section we show that the simple euclidean loss with both $\chi$ and $p$ can be analysed, and it contains both the high and low frequency biases discussed previously in isolation. This is not very surprising, but we include it for completeness.

$$\mathcal{L}_1 = -\iint_{-\infty}^{\infty} \|\boldsymbol{g}(x) - \boldsymbol{g}(x')\|^2 \underbrace{(1 - e^{-\frac{(x-x')^2}{2l^2}})}_{\chi} \overbrace{\frac{1}{2\pi L^2} e^{-\frac{x^2+x'^2}{2L^2}}}^{p(x)p(x')} dx dx' \tag{60}$$

Following a very similar path to before we reach:

$$= -\frac{4}{\pi L^2} \sum_{n,d,d'} a_{nd}a_{nd'}C(\omega_d,\omega_{d'}\|L)\Delta S(\omega_d,\omega_{d'}\|L,\Lambda) + b_{nd}b_{nd'}\Delta S(\omega_d,\omega_{d'}\|L,\Lambda)S(\omega_d,\omega_{d'}\|L) \tag{61}$$

$$\Delta S(\omega_d,\omega_{d'}\|L,\Lambda) = S(\omega_d,\omega_{d'}\|L) - S(\omega_d,\omega_{d'}\|\Lambda) \tag{62}$$

$$\Lambda = \frac{lL}{\sqrt{l^2 + 2L^2}} \approx \frac{l}{\sqrt{2}} \qquad \text{since} \quad L > l \tag{63}$$

This contains the low and high frequency penalisation, at lengthscales $\frac{1}{L}$ and $\frac{1}{l}$, respectively. There are 4 terms, 2 are identical to equation 57, and hence perform the same role. The additional two

terms are positive, so should be minimised, and they can be minimised by making the frequencies smaller than $\sim \frac{1}{l}$, Figure 13D.

We verify this claim numerically by optimising a neural representation of 2D position, $g(x)$, to minimise $\mathcal{L}_1$ subject to actionable and biological constraints. It forms a module of hexagonal cells, Figure 13E, but the representation is shoddy, because the cells only encode a small area of space well, as can be seen by zooming on the response, Figure 13.

# E   ANALYSIS OF FULL LOSS: MULTIPLE MODULES OF GRIDS

In this Section we will perturbatively analyse the full loss function, which includes the neural lengthscale, $\sigma$. The consequences of including this lengthscale will become most obvious for the simplest example, the representation of an angle on a ring, $g(\theta)$, with a uniform weighting of the importance of separating different points. We'll use it to derive the push towards non-harmonically related modules.

Hence, we study the following loss:

$$\mathcal{L} = \frac{1}{4\pi^2} \iint_{-\pi}^{\pi} e^{-\frac{\|g(\theta)-g(\theta')\|^2}{2\sigma^2}} d\theta d\theta' \tag{64}$$

And examine how the effects of this loss differ from the euclidean verison studied in Appendix C.

This loss is difficult to analyse, but insight can be gained by making an approximation. We assume that the distance in neural space between the representations of two inputs depends only on the difference between those two inputs, i.e.:

$$\|g(\theta)-g(\theta')\|^2 = d(\theta-\theta') \tag{65}$$

Looking at the constrained form of the representation, we can see this is satisfied if frequencies are orthogonal:

$$g(\theta) = a_0 + \sum_{d=1}^{D} a_d \sin(n_d \theta) + b_d \cos(n_d \theta) \qquad a_0, \{a_d, b_d\}_{d=1}^{D} \in \mathbb{R}^N, n_d \in \mathbb{Z} \tag{66}$$

$$\begin{aligned} a_d \cdot a_{d'} &= \delta_{d,d'} A_d^2 \\ b_d \cdot b_{d'} &= \delta_{d,d'} A_d^2 \\ a_d \cdot b_{d'} &= 0 \end{aligned} \tag{67}$$

Then:

$$\|g(\theta)-g(\theta')\|^2 = \sum_{d=1}^{D} A_d^2 \sin^2\left(\frac{n_d}{2}(\theta-\theta')\right) \tag{68}$$

Now we will combine this convenient form with the following expansion:

$$e^{x\cos(y)} = I_0(x)\left[1 + 2\sum_{m=1}^{\infty} \frac{I_m(x)}{I_0(x)} \cos(my)\right] \tag{69}$$

Developing equation 64 using eqns. 69 and 68:

$$\mathcal{L} = \frac{1}{4\pi^2} \iint_{-\pi}^{\pi} e^{-\frac{\sum_d A_d^2 \sin^2(n_d \frac{\theta-\theta'}{2})}{2\sigma^2}} d\theta d\theta' \tag{70}$$

$$= \frac{1}{2\pi} \int_{-\pi}^{\pi} e^{-\frac{\sum_d A_d^2 \sin^2(n_d \alpha)}{2\sigma^2}} d\alpha \tag{71}$$

$$= \frac{1}{2\pi} \prod_d \int_{-\pi}^{\pi} e^{-\frac{A_d^2}{4\sigma^2}(1-\cos(2n_d \alpha))} d\alpha \tag{72}$$

$$= \frac{1}{2\pi} \underbrace{\prod_d \left[e^{-\frac{A_d^2}{4\sigma^2}} I_0\left(\frac{A_d^2}{4\sigma^2}\right)\right]}_{\text{Term A}} \underbrace{\int_{-\pi}^{\pi} \prod_d \left[1 + 2\sum_{m=1}^{\infty} \frac{I_m\left(\frac{A_d^2}{4\sigma^2}\right)}{I_0\left(\frac{A_d^2}{4\sigma^2}\right)} \cos(2n_d m\alpha)\right] d\alpha}_{\text{Term B}} \tag{73}$$

We will now look at each of these two terms separately and derive from each an intuitive conclusion. Term A will tell us that frequency amplitudes will tend to be capped at around the legnthscale $\sigma$. Term B will tell us how well different frequencies combine, and as such, will exert the push towards non-harmonically related frequency combinations.

### E.1 TERM A ENCOURAGES A DIVERSITY OF HIGH AMPLITUDE FREQUENCIES IN THE CODE

Fig 14A shows a plot of $I_0(\frac{A_d^2}{4\sigma^2})$. As can be seen there are a couple of regimes, if $A_d$, the amplitude of frequency $n_d$, is smaller than $\sigma$ then $I_0(\frac{A_d^2}{4\sigma^2}) \approx 1$. On the other hand, as $A_d$ grows, $I_0$ also grows. Asymptotically:

$$I_0(\frac{A_d^2}{4\sigma^2}) \sim \frac{e^{\frac{A_d^2}{4\sigma^2}}}{\sqrt{\frac{A_d^2}{4\sigma^2}}} \tag{74}$$

Correspondingly Term A also behaves differently in the two regimes, when $A_d$ is smaller than $\sigma$ the loss decreases exponentially with the amplitude of frequency $d$:

$$A_d < \sigma \qquad \text{Term A} = e^{-\frac{A_d^2}{4\sigma^2}} I_0(\frac{A_d^2}{4\sigma^2}) \sim e^{-\frac{A_d^2}{4\sigma^2}} \tag{75}$$

Alternatively:

$$A_d > \sigma \qquad \text{Term A} = e^{-\frac{A_d^2}{4\sigma^2}} I_0(\frac{A_d^2}{4\sigma^2}) \sim \frac{2\sigma}{A_d} \tag{76}$$

So, up to a threshold region defined by the lengthscale $\sigma$, there is an exponential improvement in the loss as you increase the amplitude. Beyond that the improvements drop to being polynomial. This makes a lot of sense, once you have used a frequency to separate a subset of points sufficiently well from one another you only get minimal gains for more increase in that frequency's amplitude, Figure 14B. Since there are many frequencies, and the norm constraint ensures that few can have an amplitude of order $\sigma$, it encourages the optimiser to put more power into the other frequencies. In other words, amplitudes are only usefully increased to a lengthscale $\sigma$, encouraging a diversity in high amplitude frequencies. This is moderately interesting, but will be a useful piece of information for the next derivation.

### E.2 TERM B ENCOURAGES THE CODE'S FREQUENCIES TO BE NON-HARMONICALLY RELATED

Term B is an integral of the product of $D$ infinite sums, which seems very counter-productive to examine. We are fortunate, however. First, the coefficients in the infinite sum decay very rapidly with $m$ when the ratio of $\frac{A_d}{\sigma}$ is $\mathcal{O}(1)$, Figure 14C. The previous section's arguments make it clear that, in any optimal code, $A_d$ will be smaller or of the same order as $\sigma$. As such, the inifite sum is rapidly truncated. Additionally, all of the coefficient ratios, $\left(\frac{I_m(\frac{A_d^2}{4\sigma^2})}{I_0(\frac{A_d^2}{4\sigma^2})}\right)$, are smaller than 1, as such we can understand the role of Term B by analysing just the first few terms in the following series expansion,

$$\text{Term B} = \prod_d \left[ 1 + 2\sum_{m=1}^{\infty} \frac{I_m(\frac{A_d^2}{4\sigma^2})}{I_0(\frac{A_d^2}{4\sigma^2})} \cos(2mn_d\alpha) \right] =$$

$$\underbrace{1}_{0^{\text{th}} \text{ Order}} + \underbrace{2\sum_{d,m} \frac{I_m(\frac{A_d^2}{4\sigma^2})}{I_0(\frac{A_d^2}{4\sigma^2})} \cos(2mn_d\alpha)}_{1^{\text{st}} \text{ Order}} + \underbrace{4\sum_{d,d',m,m'} \frac{I_m(\frac{A_d^2}{4\sigma^2})I_{m'}(\frac{A_{d'}^2}{4\sigma^2})}{I_0(\frac{A_d^2}{4\sigma^2})I_0(\frac{A_{d'}^2}{4\sigma^2})} \cos(2mn_d\alpha)\cos(2m'n_{d'}\alpha) + \dots}_{2^{\text{nd}}\text{Order}}$$
$$\tag{77}$$

where, due to the rapid decay of $\left(\frac{I_m(\frac{A_d^2}{4\sigma^2})}{I_0(\frac{A_d^2}{4\sigma^2})}\right)$ with $m$, each term is only non-zero for small $m$.

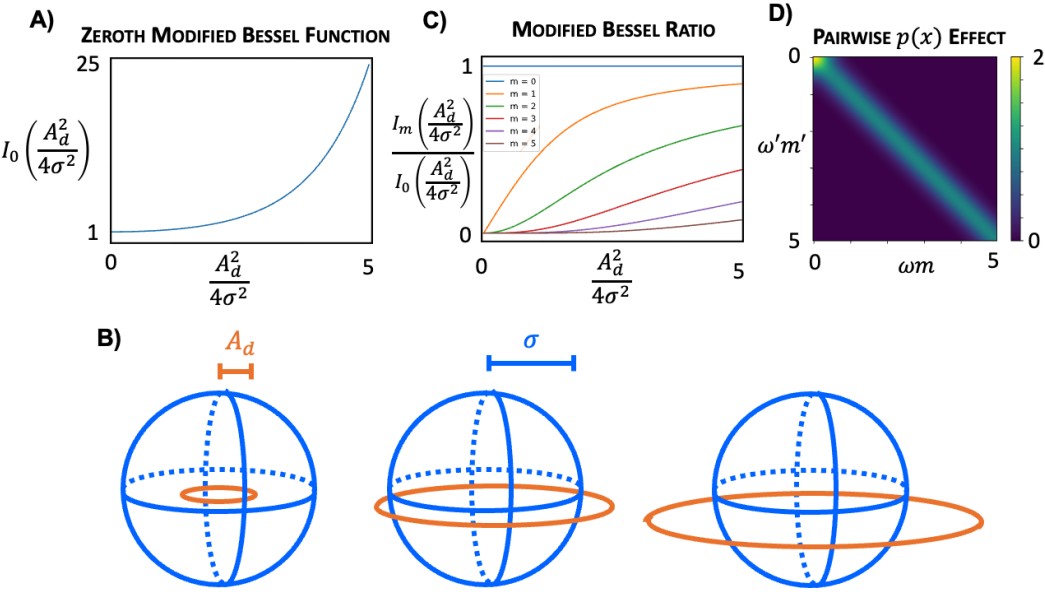

Figure 14: **A** Modified Bessel Function. **B** Amplitude Usefully Increases to a limit $\sigma$. When $A_d < \sigma$ then growing the amplitude pushes many points very usefully apart, hence the loss decreases quickly. However, once $A_d \sim \sigma$ many points are already sufficiently separated. The only gains from growing the amplitude more are in the small region that remain neighbouring, hence the loss starts decreasing much more slowly, and precious amplitude is much better invested in other smaller amplitude frequencies. **C** Ratios of Modified Bessel Functions. The terms appearing in the series expansion are, first, smaller than 1, so each additional term is the series, which is multiplied by an additional ratio smaller than 1, decays rapidly. And, second, these ratios decay rapidly with $m$, so the infinite sums within each term in the series expansion decay rapidly, containing roughly 3 or 4 non-zero terms for the largest amplitude frequencies in the code. **D** Pairwise Continuous Interactions. $p(x)$ causes frequencies to be penalised for both being smaller than $\sim \frac{1}{L}$, and, to a lesser extent, for being harmonically related, up to a lengthscale $\sim \frac{1}{L}$, equation 85. This same interaction applies for all pairs of frequencies, and all pairs of harmonics of frequencies, though the contribution is more downweighted the higher the harmonics in the pair.

The $0^{\text{th}}$ order term is a constant, and the $1^{\text{st}}$ order term is zero when you integrate over $\alpha$. As such, the leading order effect of this loss on the optimal choice of frequencies arrives via the $2^{\text{nd}}$ order term. The integral is easy,

$$\int_{-\pi}^{\pi} \cos(2mn_d\alpha)\cos(2mn_{d'}\alpha)d\alpha = \pi\delta(mn_d - m'n_{d'}) \tag{78}$$

We want to minimise the loss, and this contribution is positive, so any time the $2^{\text{nd}}$ order term is non-zero it is a penalty. As such, we want as few frequencies to satisfy the condition in the delta function, $mn_d = m'n_{d'}$. Recall from equation 77 that these terms are weighted by coefficients that decay rapidly as $m$ grows. As such, the condition that the loss is penalising, $mn_d = m'n_{d'}$, is really saying: penalise all pairs of frequencies in the code, $n_d$, $n_{d'}$, that can be related by some simple harmonic ratio, $n_d = \frac{m'}{m}n_{d'}$, where $m$ and $m'$ are small integers, roughly less than 5, Figure 14C.

This may seem obtuse, but is very interpretable! Simply harmonically related frequencies are exactly those that encode many inputs with the same response! Hence, they should be penalised, Fig 4C.

### E.3  A Harmonic Tussle - Non-Harmonics Leads to Multiple Modules

As described in Section 3.3, we now arrive at a dilemma. The arguments of Sections 3.1 and C suggest that the optimal representation is comprised of lattice modules, i.e. the neurons must build a code from a lattice of frequencies. A lattice is exactly the arrangement that ensures all the frequencies are related to one another by simple harmonic ratios. Yet, the arguments of the previous section

suggest that frequencies with simple harmonic relations are a bad choice - i.e. a frequency lattice would be the worst option! How can this be resolved?

The resolution will depend on the value of two parameters: the first, $\sigma$, is the neural lengthscale; the second, $N$, is the number of neurons, which, through the boundedness constraint, sets the scale of the amplitudes, $A_d$. We can already see this key ratio, $\frac{A_d}{\sigma}$, appearing in the preceding arguments. Its role is to choose the compromise point in this harmonic tussle, and in doing so it chooses the number of modules in the code.

A simple Taylor expansion illustrates that as $\sigma$ tends towards $\infty$ the full loss reverts to the euclidean approximation that was the focus of Sections C and D, in which the effect of the harmonic penalisation drops to zero.

$$e^{-\frac{\|\boldsymbol{g}(\theta)-\boldsymbol{g}(\theta')\|^2}{2\sigma^2}} = 1 - \|\boldsymbol{g}(\theta) - \boldsymbol{g}(\theta')\|^2 \frac{1}{2\sigma^2} + \mathcal{O}(\frac{1}{\sigma^4}) \tag{79}$$

Hence, as $\sigma \to \infty$ we recover our original optimal solution: all the neurons belong to one module, whose shape is determined by $\chi$ and $p$. This explains why, with few neurons (equivalent to $\sigma \to \infty$), the optimal solution is a single module of perfectly hexagonal grids, fig 4B.

But as $\sigma$ decreases the effect of the harmonic penalisation becomes larger and larger. Eventually, one module is no longer optimal. Yet, all the previous arguments that modularised codes are easily made non-negative, and hence valuable, still apply. The obvious solution is to have two modules. Within each module all the neurons are easily made non-negative, but between modules you can have very non-harmonic relations, trading off both requirements.

As such increasing the ratio $\frac{A_d}{\sigma}$, either by increasing the number of neurons or decreasing $\sigma$, creates an optimisation problem for which the high quality representations are modules of grids with more modules as the ratio gets bigger. We observed exactly this effect numerically, and it explains the presence of multiple modules of grids in the full version of our problem for a range of values of $\sigma$. Finally, we can extract from this analysis a guiding light: that the frequencies in different modules should be maximally non-harmonically related. This additional principle, beyond the simple low and high frequency biases of Section D, frames our normative arguments for the optimal arrangement of modules, that we explore Section 4.2 and Appendix K.

### E.4 Non-Harmonically Related Continuous Frequencies - A Smoothing Effect

We will address one final concern. When representing infinite variables the frequencies in the code become real numbers. For example, when representing a 1-dimensional continuous input, $x$, the code, $\boldsymbol{g}(x)$, has the form,

$$\boldsymbol{g}(x) = \boldsymbol{a}_0 + \sum_{d=1}^{D} \boldsymbol{a}_d \sin(\omega_d x) + \boldsymbol{b}_d \cos(\omega_d x) \qquad \omega_d \in \mathbb{R} \tag{80}$$

And it seems like the equivalent penalisation as in equation 78, $\delta(m\omega_d - m'\omega_{d'})$, is very easy to avoid. If $m\omega_d = m'\omega_{d'}$ and the code is being penalised, just increase $\omega_d$ by an infinitely small amount, $\delta\omega$, such that $m\omega_d + m\delta\omega \neq m'\omega_{d'}$, and suddenly this term does not contribute to the loss.

This, however, is not how the loss manifests. Instead frequencies are penalised for being close to one another, where close is defined by the size of the region the animal encodes, set by lengthscale $L$. This can be seen by analysing the following loss,

$$\mathcal{L} = \iint_{-\infty}^{\infty} e^{-\frac{\|\boldsymbol{g}(x)-\boldsymbol{g}(x')\|^2}{2\sigma^2}} p(x)p(x')dxdx' \tag{81}$$

$$= \frac{1}{2\pi L^2} \iint_{-\infty}^{\infty} e^{-\frac{\|\boldsymbol{g}(x)-\boldsymbol{g}(x')\|^2}{2\sigma^2}} e^{-\frac{x^2+x'^2}{2L^2}} dxdx' \tag{82}$$

Making the same assumption as in the previous Section, equation 67, and developing the loss identically,

$$\mathcal{L} = \frac{1}{2\sqrt{\pi}L} \underbrace{\prod_d \left[ e^{-\frac{A_d^2}{4\sigma^2}} I_0\left(\frac{A_d^2}{4\sigma^2}\right) \right]}_{\text{Term A}} \underbrace{\int_{-\pi}^{\pi} \prod_d \left[ 1 + 2\sum_{m=1}^{\infty} \frac{I_m\left(\frac{A_d^2}{4\sigma^2}\right)}{I_0\left(\frac{A_d^2}{4\sigma^2}\right)} \cos(2\omega_d m\alpha) \right] e^{-\frac{\alpha^2}{4L^2}} d\alpha}_{\text{Term B}} \quad (83)$$

We again analyse this through the series expansion in equation 77. The $0^{\text{th}}$ order term is again constant. The $1^{\text{st}}$ order term is proporational to,

$$\frac{1}{2\sqrt{\pi}L} \int_{-\infty}^{\infty} \cos(2m\omega_d\alpha) e^{-\frac{\alpha^2}{4L^2}} d\alpha = e^{-4L^2 m^2 \omega_d^2} \quad (84)$$

i.e., as in Section D.2, $p(x)$ implements a high frequency bias. In fact, the mathematical form of this penalty, a gaussian, is identical to that in diagonal terms of the previously derived sum, equation 57.

The $2^{\text{nd}}$ order term is where we find the answer to this Section's original concern,

$$\frac{1}{2\sqrt{\pi}L} \int_{-\infty}^{\infty} \cos(2m\omega_d\alpha) cos(2m'\omega_{d'}\alpha) e^{-\frac{\alpha^2}{4L^2}} d\alpha = \frac{1}{2\sqrt{\pi}L} C(2m\omega_d, 2m'\omega_{d'} \| 2L) \quad (85)$$

$$= \frac{1}{2}\left( e^{-4L^2(m\omega_d + m'\omega_{d'})^2} + e^{-4L^2(m\omega_d - m'\omega_{d'})^2} \right) \quad (86)$$

Which is similar to the cross terms in equation 57, but with the addition of harmonic interactions. This is the continuous equivalent to the penalty in equation 78. Frequencies and their low order harmonics are penalised for landing closer than $\sim \frac{1}{L}$ from one another, Fig 14D. In conclusion, the logic gracefully transfers to continuous frequencies.

## F    LINKS TO SENGUPTA ET AL. 2018

Sengupta et al. (2018) derive an optimal representation of structured variables, where their definition of structured exactly aligns with ours: the variable's transformations form a group (though the term they use is symmetric manifolds). Further, there are analytic similarities between the loss they study and ours. However, their optimal representations are place cells, and ours are multiple modules of grid cells. In this section we will disentangle these two threads, explaining this difference, which arises from our actionability principle, the neural lengthscale $\sigma$, and our use of infinite inputs.

Sengupta et al. (2018) study the optimisation of a loss derived from a dot product similarity matching objective, an objective that has prompted numerous interesting discoveries (Pehlevan & Chklovskii, 2019). Input vectors, $x$, stacked into the matrix $X$, are represented by vectors of neural activity, $y$, stacked into $Y$, that have to be non-negative and bounded. The loss measures the similarity between pairs of inputs, $x^T x'$, and pairs of outputs, $y^T y'$, and tries to optimise the neural representation, $y$, such that all input similarities above a threshold, $\alpha$, are faithfully reproduced in the similarity structure of the neural representation.

$$\min_{\substack{Y \geq 0 \\ \text{diag}(Y^T Y) \leq \beta \mathbf{1}}} -\text{Tr}((X^T X - \alpha \mathbf{E})Y^T Y) \quad (87)$$

where $E$ is a matrix of ones, and $\text{diag}(Y^T Y) \leq \beta \mathbf{1}$ enforces the neural activity's norm constraint.

Our loss is strikingly similar. Our inputs are variables, such as the angle, that play the role of $x$. Sengupta et al.'s input dot product similarity, $x^T x' - \alpha$, plays the role of $\chi(\theta, \theta')$, measuring similarity in input space. The neural activities, $g$, must, like $y$, be non-negative and bounded in L2 norm. And minimising the similarity matching loss is like minimising the simple euclidean form of the loss studied in Appendix D, with the integrals playing the role of the trace,

$$\mathcal{L}_1 = -\frac{1}{4\pi^2} \iint_{-\pi}^{\pi} \|g(\theta) - g(\theta')\|^2 \chi(\theta, \theta') d\theta d\theta', \quad (88)$$

The analogy then, is that minimising $\mathcal{L}_1$ is asking the most similar elements of the input to be represented most similarly, and dissimilar dissimilarly, as in the non-negative similarity matching objective; an orthogonal motivation for our loss!

Sengupta et al. (2018) use beautiful convex optimisation techniques (that cannot, unfortunately, be applied directly to our problem, Appendix G) to show that the optimal representation of the finite spaces they consider (circles and spheres) are populations of place cells. What is it about our losses that leads us to different conclusions?

Our problems differ in a few ways. First, since they use dot-product similarities, their loss is closest to our simplified losses, i.e. those without the presence of the neural lengthscale, $\sigma$. So, due to this difference, we would expect them to produce a single module of responses, as observed.

The second difference is the addition of actionability to our code which produces a harsh frequency content constraint, and is pivotal to our derivation (as can be seen from ablation studies, Appendix I). They, however, assume infinitely many neurons, at which point the frequency constraint becomes, especially when there is only one module, relatively vacuous. So we could still see their analysis as the infinite neuron version of ours.

Third, we use a different measure of input similarity, though we do not believe this would have particularly strong effects on our conclusions.

Finally, we consider infinite spaces, like position on a line, by introducing a probability distribution over inputs, $p(x)$.

As such, for representations of finite variables, like angle, the infinite neuron predictions of our simplified loss and Sengupta et al. should align. Thankfully, they do! As shown in Appendix M we predict place-cell like tuning for the representation of circular variables. This follows from our previous discussions, Appendix D.1: for finite spaces including a low frequency bias via the input similarity $\chi$ was argued to make the optimal representation a module of the lowest frequency lattices, place cells!

Hence, relative to Sengupta et al., by extending to the representation of infinite spaces with actionability we reach grids, rather than place cells. And by additionally including $\sigma$, the neural lengthscale, we produce multiple modules.

## G  OPTIMAL NON-NEGATIVE ACTIONABLE CODE - A NON-CONVEX PROBLEM

In this section we highlight that actionability is what makes our problem non-convex. Specifically, choosing the set of frequencies is difficult: for a fixed choice of frequencies the simplified loss (equation 7) and constraints are convex in the neural coefficients. We include it here to point towards tools that may be useful in proving grid's optimality, rather than heuristically justifying them with numerically support, as we have done.

Consider a representation $\boldsymbol{g}(\theta)$. For a fixed set of frequencies $\{n_d\}_{d=1}^D$, we'll aim to optimise the coefficients $\boldsymbol{a}_0, \{\boldsymbol{a}_d, \boldsymbol{b}_d\}_{d=1}^D$. The losses take simple forms, the functional objective to be minimised is $\min -\sum_d \|\boldsymbol{a}_d\|^2 + \|\boldsymbol{b}_d\|^2$. The boundedness constraint is similarly quadratic in the coefficients, equation 8. The only potential trouble is the non-negativity constraint. Sengupta et al. (2018) solve this problem by having no actionability and infinitely many neurons, we can take a different approach.

Our code is a Positive Trigonometric Polynomial, a function of the following form:

$$g_n(\theta) = \sum_{d=-D_{\max}}^{D_{\max}} r_{dn} e^{id\theta} \tag{89}$$

where $D_{\max} = \max_d n_d$, a type of polynomial studied in engineering and optimisation settings Dumitrescu (2007). Due to actionability our code is a sparse positive trigonometric polynomial, only $2D + 1$ of the $\boldsymbol{r_d}$ are non-zero.

While positivity is a hard constraint to enforce, Bochner's theorem gives us hope. It tells us that for $g_n(\theta)$ to be positive for all $\theta$, its fourier transform, $g_n(\omega)$ must be positive-definite. This means that the following matrix, made from the fourier coefficients, must be positive-definite:

$$
\boldsymbol{Q}_n = \begin{pmatrix} r_{n0} & r_{n1}^* & r_{n2}^* & \cdots & r_{nD_{\max}}^* \\ r_{n1} & r_{n0} & r_{n1}^* & & \vdots \\ r_{n2} & r_{n1} & r_{n0} & & \\ \vdots & & & \ddots & r_{n1}^* \\ r_{nD_{\max}} & \cdots & & r_{n1} & r_{n0} \end{pmatrix} \succ 0 \tag{90}
$$

The set of positive-definite matrices forms a convex set. Even the set of matrices $\boldsymbol{Q}_n$ where fixed diagonals are zero is a convex set. Further, this matrix displays all the right tradeoffs for positivity: $r_{n0}$ must be non-zero for the matrix to be positive-definite, i.e. there must be a constant offset to achieve non-negativity. Similarly, including harmonics makes achieving non-negativity easier than non-harmonics, just as in Figure 2A.

But even having made this link we cannot make progress, since we had to choose which frequencies had non-zero amplitude. Hence, the choice of frequency sparsity implied by actionability stops our problem from being convex. A plausible solution might be to solve the convex optimisation problem for a given set of frequencies, then to take gradients through that solution to update the frequencies, but we have not implemented this.

## H    LINKS TO SOME OTHER PATH INTEGRATION MODELS

Our theory is a theory of path-integrating representations. Actionability means you understand the rules of space (a representation should be unchanged after taking a step north then east then south then west), but it does not mean you understand where you are, or how fine-grained this understanding is (a representation that encodes all positions the same way satisfies the actionable rules for example). Our functional constraint rectifies this, as it requires all relevant locations to be represented differently, up to a certain spatial scale. Understanding rules, and representing locations differently is the same as path-integration - now a representation can be updated according to the rules, and the underlying spatial variables can be decoded.

### H.1    LINEAR PATH INTEGRATION MODELS

Linear path integration models of the entorhinal cortex, like ours, have, to the best of our knowledge, been suggested three times before (Issa & Zhang, 2012; Whittington et al., 2020; Gao et al., 2021). Whittington et al. and Gao et al. are both simulation based, optimising losses related to ours, and show the resulting optimisations look like grid cells, though both papers only obtain multiple modules of grids by manually enforcing block diagonal transformation matrices. In this section we will discuss the differences between these two losses and ours, to explain their results in our framework as much as possible.

**Gao et al.** optimise a representation of position and a set of linear neural transition matrices to minimise a series of losses. One loss enforces path integration, like our actionability constraint. Another bounds the firing rate of vectors, similar to our boundedness constraint. A final similarity is their version of the functional objective, which enforces the ability to linearly readout place-cell-like activity from the grid code.

However, additionally they enforce an 'isotropic scaling term'. This term measures how much the neural activity vector moves when you take a small step in direction $\psi$ in the input space, and encourages all the steps to be the same size independent of $\psi$. This can be understood through the strong force towards hexagonality it produces: if you know your representation is some type of grid then of all the two dimensional grid the one that is most symmetric is hexagons, with it's $C_6$ symmetry. Hence it will minimise this isotropic scaling.

We have shown that it is not necessary to include this term in the loss to produce hexagonal grids. However, there will be differences in behaviour between a code minimising this constraint and a

code that doesn't. The stronger push towards hexagonality that isotopy exerts will make deviations from perfect hexagonality, such as shearing Stensola et al. (2015) less optimal. A careful analysis of the behaviour of the two losses, both with functional, actionable, and biological constraints, but adding or removing isotropy, in different rooms could produce different predictions. Further, each module would be more robustly hexagonal with this term added, which might be necessary to make many many modules hexagonal.

Finally, the relation between our non-negative firing rates, and the concept of non-negativity in Gao et al. is slightly murky. Gao et al. use non-negative grid-to-place-cell weights, but also show this non-negativity is not required to produce grids. How this meshes with our findings that non-negativity is vital for grid formation remains unclear. This discrepancy could be caused by the isotropy constraint, but we remain unsure.

Theoretically, Gao et al. develop a rich group theory view for studying grid cells. We add to this work using Representation theory to make clear what constraints linear actionability implies (equation 5). Given this starting point we are able to explain intuitively and with analytic insight why this loss leads to grid cells, extend to multiple modules, and make neural predictions.

**Whittington et al.** train a recurrent neural network to path integrate, but also to separate the representation of different points in the input space by predicting different observations at different positions. Path integration is linear with action dependent matrices, exactly like our actionability, and the neural activities are bounded. In recent versions of the model, Whittington et al. (2021), non-negative firing rates are used, which gave hexagonal grid cells. This can be understood as a direct application of our framework, though the pressures on the representation are naturally more blurry since it is optimised with behavioural trajectories using a recurrent neural network. In particular, the goldilocks frequency bounds (section 3.2) can be thought of as coming from the finite map the agent explores (high frequency bias) and the discrete step-size in the environment (low frequency bias).

## H.2 SORSCHER, MEL, ET AL.'S NON-LINEAR PATH INTEGRATION MODEL

Sorscher et al. (2019) provide a nice theoretical analysis of how non-negativity and linear reconstruction of place cells can lead to hexagonal grids, providing valuable insight into previous observations in the field such as that of Dordek et al. (2016).

The key difference between our theoretical analysis and theirs, is that they rely on the covariance structure of place cell input to obtain a representation made from plane waves, whereas we show that actionability (path integration) alone demands a representation built from plane waves. The reliance on a particular place cell covariance structure makes a strict constraint on the form place cells take - whether this is compatible with observed place cells is a topic of much debate (Schaeffer et al., 2022; Sorscher et al., 2022). The subsequent arguments that lead to hexagonal grid cells bear some resemblance, but are different, as we outline here:

Sorscher et al. (2019) discuss how finite room effects (i.e. the animal only exploring a limited region of space) lead to a discretisation of Fourier space onto a lattice. Their place cell covariance structure argument then says allowed frequencies should lie on a ring since the spectrum is peaked in a small band of frequencies. Lastly, including a 3rd order regularisation on the code (such as non-negativity) induces a triplet interaction between frequencies on the ring, meaning they should add to zero - this is satisfied by a hexagonal code.

Our argument also leads to lattice frequencies, but this is due to actionability and non-negativity. Then, from the space of all lattices, a band-pass filter argument chooses hexagons: the high-pass arises because the animal only explores a limited region of space, and the low-pass because the animal only wants to encode space up to some useful resolution. This creates a preferred an annulus in frequency space, and leads to a preference for a hexagonal lattice, since it most densely packs frequencies into the annulus that is optimal for encoding space.

So the arguments have similar components - lattice frequencies, non-negativity, band-pass filters - but they put these pieces together in different ways.

These technical differences, among others, lead our two theories to make different predictions. For example, we provide a normative derivation of multiple modules, which we believe only arise in

Sorscher et al. (2019) due to a quantization effect. Finally, and crucially in our opinion, our predictions result from a normative framework for understanding generic representations.

## I    ABLATION STUDIES

We have argued that three principles - biological, functional, and actionable - are necessary to produce grid cells. In this section we show that removing any of these three and numerically optimising produces fundamentally different patterns, and we explain these results with the theory we've developed in this work.

We study the representation of a small torus, since for periodic spaces the three terms appear cleanly in the loss, so can be easily excised, Appendix B. Figure 15 shows that removing any term stops grid cells emerging.

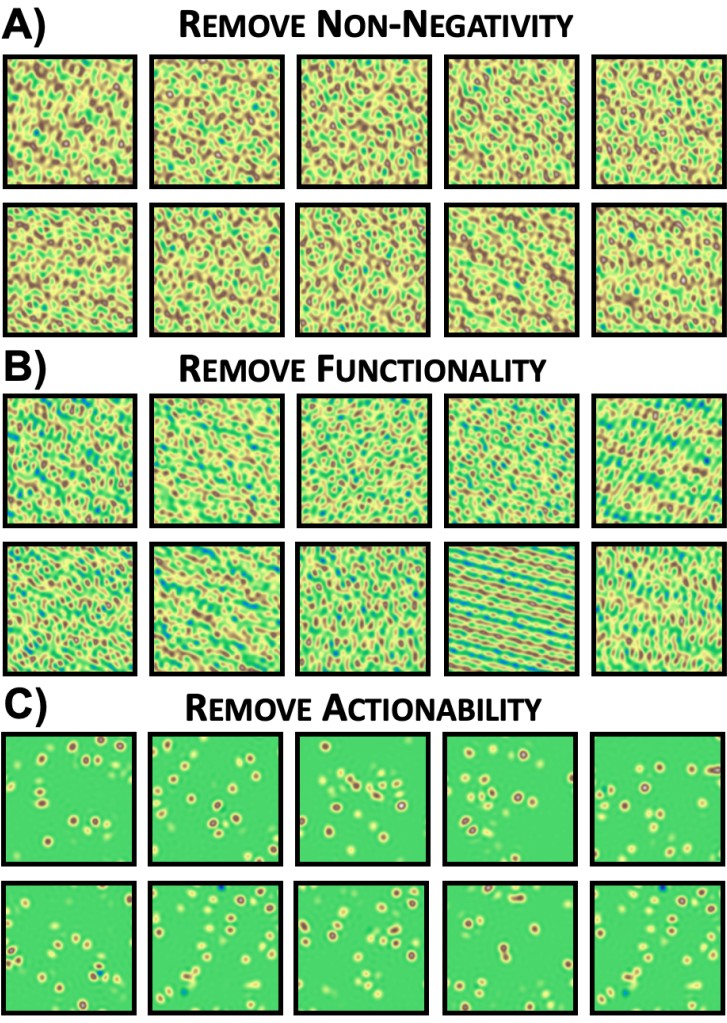

Figure 15: **A** With no non-negativity the code is simply built from $D$ non-harmonically related frequencies. **B** Without the functional term an optimal code can be made by increasing the constant vector sufficiently, and containing few frequencies, leaving a random pattern of sines and cosines. **C** Without actionability there is no push towards sparsity in fourier space. Bumps are still easy to positivise, without frequency sparsity there's no reason to arrange them into lattices. Hence, this response looks a bit like grid cells with the peak positions randomised.

**No Non-negativity** Without non-negativity the code wants to maximise the size of each of the limited number of frequencies in the code, ensuring that they are non-harmonically related. As such, it builds a wavey but fairly random looking code.

**No Functional** Without a functional term the loss simply wants the code to contain few frequencies and be non-negative, so produces random looking responses with a sparse fourier spectrum and a large constant offset to ensure non-negativity.

**No Actionability** Without actionability the code wants to be non-negative but separable. Bumps are easily positivised and encode the space well, but there is no reason to arrange them into lattices, and it would actually be detrimental to co-ordinate the placement across neurons so that any hypothetical lattice axes align. As such, it builds a random but bumpy looking pattern. This observation will be important for our discussion of the coding of 3-dimensional space, Appendix M.

## J  LATTICE SIZE:PEAK WIDTH RATIO SCALES WITH NUMBER OF NEURONS

Thanks to the actionability constraint, the number of neurons in a module controls the number of frequencies available to build the module's response pattern. In our scheme the frequencies within one module organise into a lattice, and the more frequencies in the module, the higher the top frequency, and the correspondingly sharper the grid peaks. This link allows us to extract the number of neurons from a single neuron's tuning curve in the module. In this section we add slightly more justification to this link.

We begin with a representation of 1D position, $g(x)$. When there are enough neurons the griddy response can be approximated as,

$$g_n(x) \approx \mathcal{N}(x\|0, \mu) * DC(\nu), \tag{91}$$

where $DC(\nu)$ is a Dirac comb with lengthscale $\nu$, $*$ is convolution, and $\mu$ is the grid peak width, Fig 16. The fourier transform of this object is simple, Fig 16,

$$\tilde{g}_n(\omega) \approx \mathcal{N}\left(\omega\|0, \frac{1}{\mu}\right) \times DC\left(\frac{1}{\nu}\right) \tag{92}$$

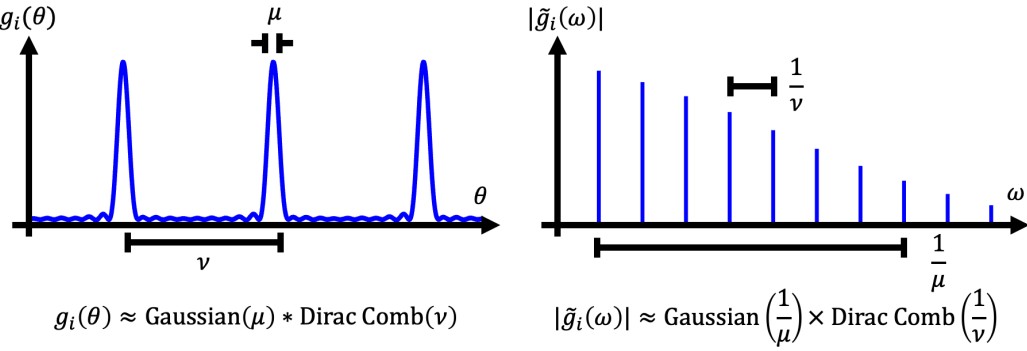

Figure 16: Approximate Large Neuron Responses. With many neurons the grid response is approximately a dirac comb convolved with a Gaussian, and hence the frequency response is approximately a Gaussian multiplied by a Dirac Comb. Actionability implies sparsity in fourier space, which we approximate as a finite number of frequencies having power above some threshold $\tau$, i.e. a sufficiently fast decay in frequency space, which corresponds to a sufficiently broad grid peak width.

We can now extract the tradeoff exactly. In this continuous approximation the equivalent of actionability's sparsity in frequency space is that the frequency spectra must decay sufficiently fast i.e. the number of frequencies with power above some small threshold must be less than half the number of neurons. This number is controlled by the ratio of $\mu$ and $\nu$. We'll set some arbitrary low scaled threshold, $\tau$, which defines the power a frequency must have, relative to the constant component, to be considered as a member of the code. The actual value of $\tau$ is inconsequential, as we just want

scaling rules. Then we'll derive the scaling of the frequency at which the amplitude of the gaussian drops below threshold, $\omega^*$:

$$\frac{\|\mathcal{N}(\omega^*\|0, \frac{1}{\mu})\|^2}{\|\mathcal{N}(0\|0, \frac{1}{\mu})\|^2} = e^{-\mu^2 \omega^{*2}} = \tau \tag{93}$$

$$\omega^* = \frac{\log(\frac{1}{2\tau})}{\mu} \tag{94}$$

Now we can ask how many frequencies there are in the code, by counting the number of peaks of the Dirac comb that occur before this threshold. Counting from 0, because in 1D negative frequencies are just scaled versions of positive frequencies:

$$D = \frac{\omega^*}{\frac{1}{\nu}} = \frac{\nu}{\mu} \log\left(\frac{1}{2\tau}\right) \tag{95}$$

Therefore, in 1D, when there are enough neurons, the number of frequencies, and therefore the number of the neurons scales with the lattice to peak width ratio:

$$N \propto \frac{\nu}{\mu} \tag{96}$$

A completely analogous argument can be made in 2D, Again the cutoff frequency is $\omega^* \propto \frac{1}{\mu}$. A different scaling emerges, however, because the number of frequencies in a dirac comb lower than some threshold scales with dimension. The frequency density of the dirac comb is proportional to $\frac{1}{\nu^2}$ peaks per unit area with the specifics depending on the shape of the frequency lattice, so we're asking how many peaks are there in a half circle of radius $\omega^*$. This is simple,

$$D \propto \frac{\pi \omega^{*2}}{2\frac{1}{\nu^2}} = \frac{\nu^2}{\mu^2} \frac{\pi}{2} \log\left(\frac{1}{2\tau}\right) \tag{97}$$

Hence we recover the stated scaling,

$$N \propto \left(\frac{\nu}{\mu}\right)^2 \tag{98}$$

## K  OPTIMAL RELATIVE ANGLE BETWEEN MODULES

Our analysis of the full loss, Appendix E, suggested that optimal representations should contain multiple modules, and each module should be chosen such that its frequencies are optimally non-harmonically related to the frequencies in other modules. This is a useful principle, but the full implications are hard to work through exactly. In this section we use rough approximations and coarse calculations to make estimates of one aspect of the relationships between modules: the optimal relative angle between lattice axes. Our scheme begins by calculating a smoothed frequency lattice density as a function of angle, $\rho(\psi)$. It then aligns two of these smoothed densities, representing two modules, at different angular offsets, and finds the angle which minimises the overlap. This angle is the predicted optimal angle. We then extend this to multiple modules by searching over all doubles or triples of alignment angles for the minimal overlap of three or four densities. We will talk through the details of this method, and arrive at the results that are quoted in section 4.2.

If you imagine standing at the origin in frequency space, corresponding to the constant component, and looking to infinity at an angle $\psi$ to the $k_x$ axis, the density function, $\rho(\psi)$, measures, as a function of $\psi$, the number of smoothed frequencies that you see, fig 17A. We assume there are many neurons in this module, and hence many frequencies in the frequency lattice. Then we make the same approximation as in eqns. 91 and 92, that the frequency response is a hexagonal grid multiplied by a decaying Gaussian. The lengthscale of this Guassian is the first parameter that will effect our results, and it scales linearly with the number of neurons in the module. Then we try and capture the repulsive effects of the lattice. Each lattice point repels frequencies over a nearby region, as calculated in Section E.4. This region is of size $\sim \frac{1}{2L}$, where $L$ is the exploration lengthscale of the animal. We therefore smooth the grid peaks in our density function by convolving the lattice with a Gaussian. We choose our units such that $L = 1$, then the width of this smoothing Gaussian is also order 1. The second important parameter is then the grid lattice lengthscale. The larger the

grid cell lengthscale the more overlapping the repulsive effects of the frequencies will be, and the smoother the repulsive density, $\rho(\psi)$. In maths,

$$\rho(\psi\|\mu,\nu) = \int_0^\infty \left[ \mathcal{N}\left(\boldsymbol{k}\Big|0,\frac{1}{\mu}\right) \times DC\left(\boldsymbol{k}\Big|\frac{1}{\nu}\right) \right] * \mathcal{N}(\boldsymbol{k}\|0,1) k_r dk_r \tag{99}$$

where $k_r$ is the radial part of the frequency. We calculate the obvious numerical approximation to this integral, Fig 17B, and in these calculations we ignore the effect of higher harmonics in equation E.4. This isn't as bad as is sounds, since higher harmonics contribute density at the same angle, $\psi$, part of the reason we chose to calculate this angular density.

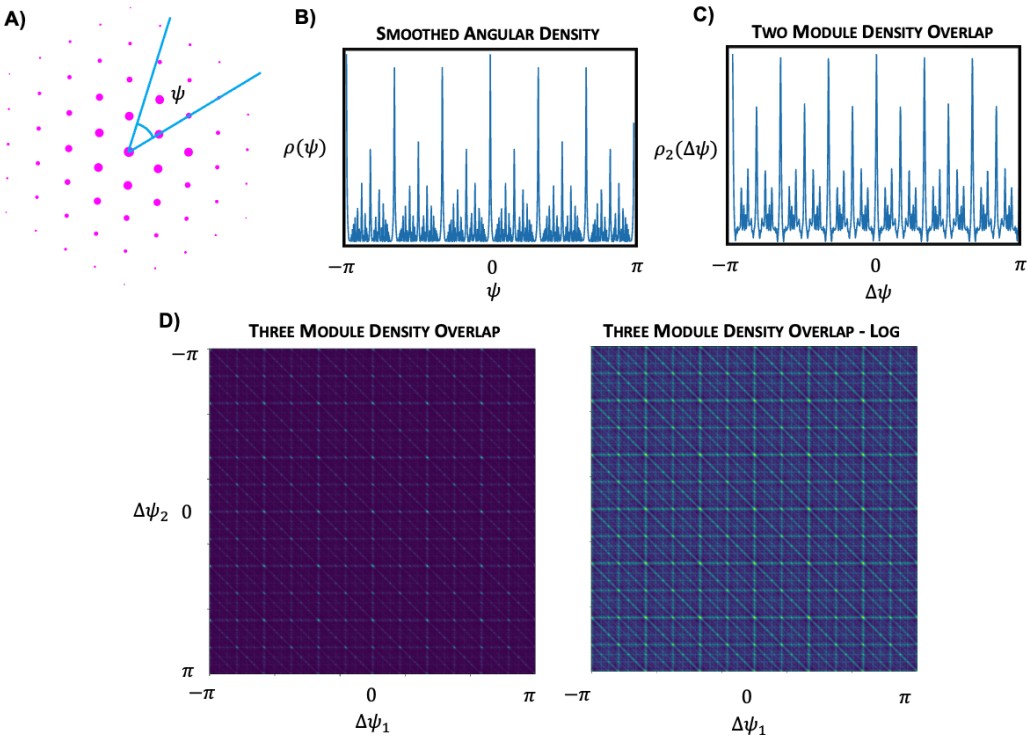

Figure 17: **A: Smoothed Angular Density** We stand at the origin and look out at angle $\psi$. We then count the density of smoothed grid frequency peaks along this line, where the smoothing occurs over lengthscale $\frac{1}{L}$. **B: Smoothed Density Plot. C: Overlap of Density Plots at Offsets $\Delta\psi$** This curve shows a clear minima at 4 degrees from perfect alignment of grid axes. **D: Three Module Overlaps** The overlap (left) and the log of the overlap (right) show similar patterns as a function of the two angular offsets between modules, and the overlap is minimise for stepwise offsets of around 4 degrees.

The next stage is finding the optimal angle between modules, starting with two identical modules. We do this by overlapping copies of $\rho(\psi\|\mu,\nu)$. For simplicity, we start with two identical lattices and calculate the overlap as a function of angular offset:

$$\tilde{\rho}_2(\Delta\psi\|\mu,\nu) = \int_{-\pi}^{\pi} \rho(\psi\|\mu,\nu)\rho(\psi+\Delta\psi\|\mu,\nu)d\psi \tag{100}$$

Then the predicted angular offset is,

$$\Delta\phi^* = \underset{\Delta\phi}{\arg\min}\, \tilde{\rho}_2(\Delta\psi\|\mu,\nu) \tag{101}$$

Figure 17C shows this is a small angular offset around 4 degrees, though it can vary between 3 and 8 degrees depending on $\mu$ and $\nu$. An identical procedure can be applied to multiple modules. For example, for the arrangement of three modules we define,

$$\tilde{\rho}_3(\Delta\psi, \Delta\psi' \| \mu, \nu) = \int_{-\pi}^{\pi} \rho(\psi\|\mu,\nu)\rho(\psi + \Delta\psi\|\mu,\nu)\rho(\psi + \Delta\psi + \Delta\psi'\|\mu,\nu)d\psi \tag{102}$$

and find the pair of offsets that minimise the overlap. For identical modules this also predicts small offsets between modules, of around 4 degrees between each module, Figure 17D, and similarly three identical modules should be oriented at multiples of 4 degrees from a reference lattice (i.e. offsets of 4, 8, and 12). As shown in Figure 5E, these small offsets appear to be present in data, validating our prediction.

There are many natural next steps for this kind of analysis. In future we will explore the effect of different lattice lengthscales and try to predict the optimal module arrangement. Another simple analysis you can do is explore how changing the grid lengthscale, $\nu$, effects the optimal relative angle. The larger the grid lattice lengthscale, the smaller its frequency lattice, and the more smooth the resulting repulsion density is. As a result, the optimal relative angle between large modules is larger than between small modules, a prediction we would like to test in data. It would be interesting to apply more careful numerical comparisons, for example matching the $\nu$ and $\mu$ to those measured in data, and comparing the real and optimal angular offsets.

## L ROOM GEOMETRY ANALYSIS

The functional objective depends on the points the animal wants to encode, which we assume are the points the animal visits. As such, the exploration pattern of the animal determines the optimal arrangement of grids, through the appearance of $p(x)$ in the functional objective. Different constraints to the exploration of the animal, such as different sized environments, will lead to different optimal representations. In this section we analyse the functional objective for a series of room shapes, and show they lead to the intricate frequency biases suggested in Section 4.3.

### L.1 1-DIMENSONAL BOX

To begin and to set the pattern for this section, we will study the representation of 1-dimensional position in a finite box. Ignoring the effects of $\chi$ for now, our loss is,

$$\mathcal{L} = \frac{1}{L^2} \iint_{-\frac{L}{2}}^{\frac{L}{2}} e^{-\frac{\|g(x)-g(x')\|^2}{2\sigma^2}} dxdx' \tag{103}$$

Making the symmetry assumption as in Section E our loss is,

$$\mathcal{L} = \frac{1}{L^2} \iint_{-\frac{L}{2}}^{\frac{L}{2}} e^{-\frac{\sum_d A_d^2 \sin^2(\frac{\omega_d(x-x')}{2})}{2\sigma^2}} dxdx' \tag{104}$$

Changing variables to $\alpha = x - x'$, $\beta = x + x'$, Figure 18A,

$$\mathcal{L} = \frac{1}{2L^2}\left[\int_0^L \int_{-L+\alpha}^{L-\alpha} e^{-\frac{\sum_d A_d^2 \sin^2(\frac{\omega_d\alpha}{2})}{2\sigma^2}} d\beta d\alpha + \int_{-L}^0 \int_{L+\alpha}^{-L-\alpha} e^{-\frac{\sum_d A_d^2 \sin^2(\frac{\omega_d\alpha}{2})}{2\sigma^2}} d\beta d\alpha\right] \tag{105}$$

$$= \frac{1}{L^2}\int_0^L e^{-\frac{\sum_d A_d^2 \sin^2(\frac{\omega_d\alpha}{2})}{2\sigma^2}}(L-\alpha)d\alpha \tag{106}$$

Performing the expansion as in Section E we reach:

$$\mathcal{L} = \frac{1}{L^2}\underbrace{\prod_d\left[e^{-\frac{A_d^2}{4\sigma^2}}I_0\left(\frac{A_d^2}{4\sigma^2}\right)\right]}_{\text{Term A}}\underbrace{\int_{-\pi}^{\pi}\prod_d\left[1 + 2\sum_{m=1}^{\infty}\frac{I_m\left(\frac{A_d^2}{4\sigma^2}\right)}{I_0\left(\frac{A_d^2}{4\sigma^2}\right)}\cos(\omega_d m\alpha)\right](L-\alpha)d\alpha}_{\text{Term B}} \tag{107}$$

Term A behaves the same as before, so we again study the series expension of term B, the analogue of equation 77. The $0^{\text{th}}$ order term is a constant with respect to the code, but the first order term provides the main frequency biasing effect, as in Section E.4.

$$\frac{1}{L^2} \int_0^L \cos(m\omega_d\alpha)(L - \alpha)d\alpha = \frac{1}{m^2 L^2 \omega_d^2}(1 - \cos(m\omega_d L)) \tag{108}$$

Hence the 1-dimensional box coarsely creates a high frequency bias ($> \frac{1}{L}$), but also encourages frequencies that fit whole numbers of wavelengths within the length of the box, Figure 18B.

Before we move on we will study the second order term. We won't repeat this for future finite rooms as it shows basically the same effect as in Section E.4: coarsly frequencies repel each other up to a lengthscale $\frac{1}{L}$. However, as for the first order term, there is additional structure, i.e. subsidiary optimal zeros in the loss, that could be of interest:

$$\frac{2}{L^2} \int_0^L \cos(m\omega_d\alpha) \cos(m'\omega_{d'}\alpha)(L - \alpha)d\alpha$$
$$= \frac{1 - \cos(L(m\omega_d + m'\omega_{d'}))}{(m\omega_d + m'\omega_{d'})^2 L^2} + \frac{1 - \cos(L(m\omega_d - m'\omega_{d'}))}{(m\omega_d - m'\omega_{d'})^2 L^2} \tag{109}$$

This is fundamentally a repulsion between frequencies closer than $\frac{1}{L}$ from one another, fig 18C. Additionally, the repulsion is zero when the difference between frequencies is exactly $\frac{2\pi}{L}$ times an integer.

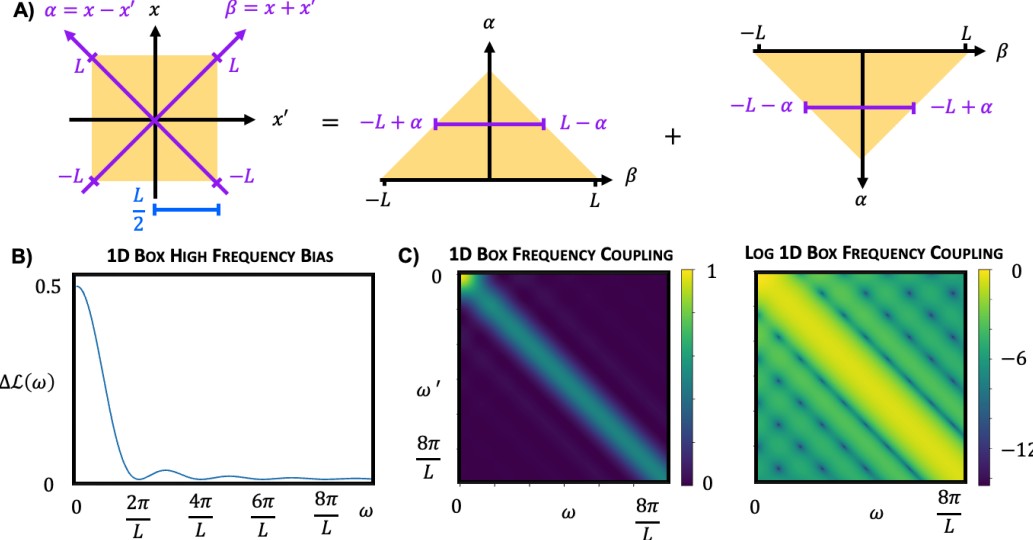

Figure 18: **A: Changing Integration Variables** The logic behind the integration change in 105. **B: 1D Box Frequency Bias** As in equation 108, shows both a high frequency bias and a preference for integer numbers of wavelengths in the box. **C: 1D Box Frequency Interaction** Plot of equation 109, shows both a high frequency bias, a repulsion between frequencies, and non-trivial optimal interaction differences.

## L.2 2-DIMENSIONAL BOX

A very similar analysis can be applied to the representation of a 2-dimensional box, and used to derive the frequency bias plotted in Figure 5F. We reach an exactly analogous point to equation 106,

$$\mathcal{L} = \frac{2}{L^4}\left[\int_0^L\int_0^L e^{-\frac{\sum_d A_d^2 \sin^2(\frac{k_{dx}x+k_{dy}y}{2})}{2\sigma^2}}(L-x)(L-y)dxdy+\right. \tag{110}$$

$$\left.\int_0^L\int_{-L}^0 e^{-\frac{\sum_d A_d^2 \sin^2(\frac{k_{dx}x+k_{dy}y}{2})}{2\sigma^2}}(L-x)(L+y)dxdy\right] \tag{111}$$

Again we'll study the first order term of the series expansion,

$$\Delta\mathcal{L}_1 \propto \frac{1-\cos(m_d k_{dx} L)}{m^2 k_{dx}^2}\frac{1-\cos(m_d k_{dy} L)}{m^2 k_{dy}^2} \tag{112}$$

This is exactly the bias plotted in Figure 5F. Changing to rectangles is easy, it is just a rescaling of one axis, which just inversely scales the corresponding frequency axis.

## L.3 CIRCULAR ROOM

The final analysis we do of this type is the hardest, a circular environment. It follows the same pattern, make the symmetry assumption, expand the series. Doing so we arrive at the following first order term:

$$\Delta\mathcal{L}_1 = \frac{1}{\pi^2 R^4}\iint_0^R\iint_0^{2\pi}\cos(m\boldsymbol{k}_d\cdot(\boldsymbol{x}-\boldsymbol{x}'))d\psi d\psi' rr' drdr' \tag{113}$$

where $(r,\psi)$ is a radial co-ordinate system, $\boldsymbol{x} = \begin{pmatrix} r\cos\psi \\ r\sin\psi \end{pmatrix}$. Using the double angle formula to spread the cosine, and rotating the 0 of each $\psi$ variable to align with $\boldsymbol{k}_d$, we get:

$$\Delta\mathcal{L}_1 = \frac{1}{\pi^2 R^4}\left[\left(\int_0^R\int_0^{2\pi}\cos(mr\|\boldsymbol{k}_d\|\cos\psi)d\psi rdr\right)^2+\left(\int_0^R\int_0^{2\pi}\sin(mr\|\boldsymbol{k}_d\|\cos\psi)d\psi rdr\right)^2\right] \tag{114}$$

The second integral is 0, because sine is odd and periodic. We can perform the first integral using the following Bessel function properties,

$$\int_0^{2\pi}\cos(a\cos(\psi))d\psi = 2\pi J_0(\|a\|) \tag{115}$$

Hence,

$$\Delta\mathcal{L}_1 = \frac{4}{R^4}\left(\int_0^R J_0(mr\|\boldsymbol{k}_d\|)rdr\right)^2 \tag{116}$$

Which can be solved using another Bessel function property,

$$\int_0^R rJ_0(ar)dr = \frac{RJ_1(aR)}{a} \tag{117}$$

Therefore,

$$\Delta\mathcal{L}_1 = \frac{4J_1(m\|\boldsymbol{k}_d\|R)^2}{R^2 m^2\|\boldsymbol{k}_d\|^2} \tag{118}$$

This is exactly the circular bias plotted in Figure 5G. Placing the base frequencies of grid module on the zeros of this bessel function would be optimal, i.e. calling a zero $\xi$, then the wavelengths of the grid modules should optimally align with,

$$\nu = \frac{2\pi R}{\xi_k} \tag{119}$$

This corresponds to grids with wavelengths that are the following multiples of the circle diameter: $0.82, 0.45, 0.31, 0.24, 0.19, \ldots$. Though, as with the other predictions, the major effect is the polynomial decay with frequency, so these effects will all be of larger importance for the larger grid modules. As such, we might expect only the first two grid modules to fit this pattern, the others should, in our framework, focus on choosing their lattice parameters to be optimally non-harmonic.

## M    REPRESENTATIONS OF CIRCLES, SPHERES, AND 3-DIMENSIONS

We believe our approach is relevant to the representation of many variables beyond 2-dimensional space. Most simply we can directly apply the tools we've developed to the representation of any variable whose transition structure is a group, Section A.1. This is actually true for both numerical and analytic approaches, but here we focus mainly on the numerical approach. We discuss the representation of an angle on a ring, a point on a sphere, and 3 dimensional space. Additional interesting variables might include a position in hyperbolic space (since tree-like category structures, such as the taxonomic hierarchy of species, are best described by hyperbolic spaces), or the full orientation of an object in 3-dimensional space.

For starters, we have frequently referenced the representation of an angle, $g(\theta)$. This is also something measured in brain, in the famous head direction circuit, perhaps the most beautiful circuit in the brain (Kim et al., 2017). There, you measure neurons that have unimodal tuning to heading direction. We match this finding, Figure 19A. This is understandable since, as discussed, on a finite space you still want a lattice module, but $\chi$, encourages you to be low frequency and there is no high frequency bias, since the space is finite and $p(\theta)$ is uniform. Thus, place cells are optimal. If we included more neurons it would likely form multiple modules of cells with higher frequency response properties.

We can also apply our work to the representation of a point on a sphere. Again, with the low frequency bias place cells are optimal, Figure 19B. Without it, or conceivably with enough neurons that multiple modules form, you can get many interesting patterns, Figure 19C-D. These could be relevant to the representation of object orientations, or in virtual reality experiments where animals must truly explore spherical spaces.

Finally, we can equally apply our framework to the representation of a point in three dimensional space. Grid cells have been measured while an animal explores in 3D and shown to have irregular firing fields in both bats (Ginosar et al., 2021), and mice (Grieves et al., 2021). Analogously to the griddy modules in one and two dimensions, our framework predicts griddy three dimensional responses that are similarly densely packing, which is verified in 19E. However, in Appendix I we performed ablation studies and found that removing the actionable constraint led to multimodal tuning curves without structure, and we find the same for 3D, Figure 19F, perhaps matching the multimodal but unstructured representation found in the brain (e.g. Figure 19G from Ginosar et al. (2021)). As such, it seems that in three dimensions the animals are building a non-negative and discriminative representation without the constraint to path integrate. Therefore, we suggest that the animals have sacrificed the ability to manipulate their internal representations in 3D and instead focus on efficiently encoding the space.

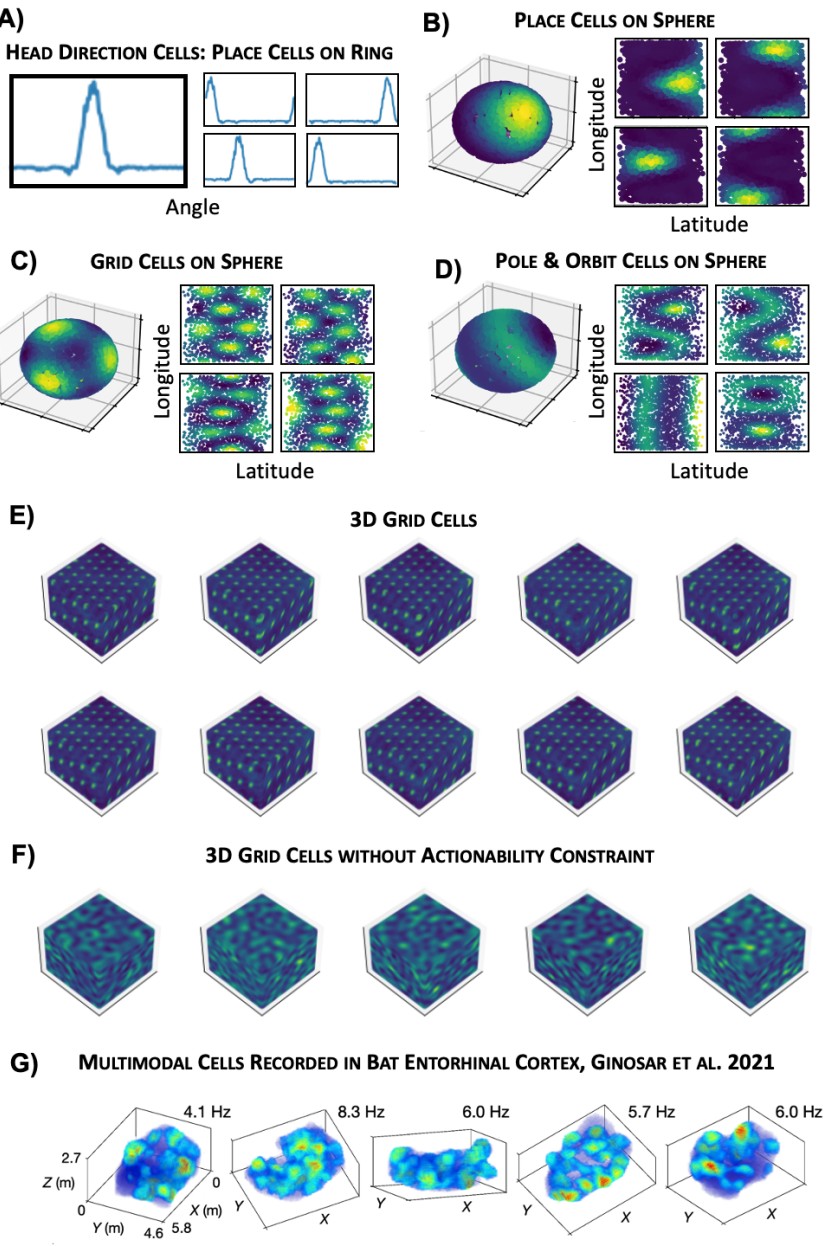

Figure 19: **A** Representation of an angle on the ring. With a low frequency bias the optimal one module response is a set of head-direction cells. **B** Representation of position on a sphere. With a low frequency response place cells are again optimal. **C-D** Allowing the optimiser to find other modules produces funky looking responses on the sphere, like grids (**C**) and 'pole & orbit' cells (**D**.) **E** A module of densely packing grids in 3D space. **F** The same cells without the actionability constraint, as we did for 2D in Figure 15. **G** Recordings of multimodal cells from bat entorhinal cortex from Ginosar et al. (2021).

