# OpenReview forum: "Actionable Neural Representations: Grid Cells from Minimal Constraints"
_ICLR.cc/2023/Conference — ICLR 2023 poster_

### Official Review · Reviewer_7jav · 2022-10-24

**Confidence:** 4
**Correctness:** 2
**Technical Novelty And Significance:** 3
**Empirical Novelty And Significance:** 2
**Recommendation:** 5

**Clarity, Quality, Novelty And Reproducibility:**

The clarity and quality would need some major improvement.  The novelty and significance were over-stated in the current version.
Overall, it feels that the work is a bit incremental.

**Strength And Weaknesses:**

Strengths:

The attempt to unify a somewhat messy literature is ambitious and should be applauded.
The proposed three key requirements for the grid patterns, if true, would be a useful finding.
The argument in Section 3.1 seems to be quite insightful.


Weaknesses:

** The idea of “actionable representation” and the use of representation theory and group theory to study the grid cell system is not a new one (e.g., see a recent line of research in Gao et al, ICLR, 2019; Gao et al., NeurIPS, 2021). The current paper did refer to Gao et al 2021 in the Introduction and had a detailed comparison to that paper in the SI. But it did not acknowledge that using group theory to study the grid cell was a key idea in that line of work.

** The math presented in the paper is not rigorous. Eq. (5) actually is a strong assumption. The paper seems to say that it is a necessary consequence of Eq. (3), thus misleading. Assuming matrix T consists of 1-by-1 and 2-by-2 blocks is a critical assumption that greatly simplified the problem.

** The writing needs to be improved. Various statements need to be toned down. The math in the SI needs to be better organized. Phrases such as “big result” should probably be removed.

** The comparison to the data is rather preliminary and unconvincing.

** Eq. (1) needs to be better motivated. The second and third terms together lead to a band-pass filter— this seems to be rather post-hoc. Previously, it was known that band-pass filtering can lead to grid patterns (Sorscher, Mel, et al, NeurIPS 2019). Some previous studies used the different-of-Gaussian place fields as inputs to the grid cell network, leading to band-pass filtering. This connection needs to be explained.  How the present theory is different from Sorscher, Mel, et al, NeurIPS 2019 also needs to be better explained.

** The study will benefit by treating neural noise more carefully.  It is unclear what assumption the model makes about the neural noise.

** how are Eq (3) and the positivity constraint compatible theoretically? The author seems to impose positivity after using Eq (3). If that is the case, Eq. (3) is no longer satisfied. I’d appreciate further clarification on this point.


---Other comments:
**It is unclear what the next paragraph really means:
 “To motivate our current rabbit hole further, this is exactly the result we hinted towards in Section 2. We discussed T(∆θ), and, in 2-dimensions, argued it was, up to a linear transform, the 2-dimensional rotation matrix. Further, we discussed how every extra two neurons allowed you to add another frequency, i.e. another rotation matrix. These rotation matrices are the irreps of the rotation group, and adding two neurons allows you to create a larger T (∆θ) by stacking rotation matrices on top of one another. Now including the invertible linear transform, S, we can state the 4-dimensional version of equation 4.”
I interpreted that the authors were making an important assumption here according to their math, but from the text,  it read like there is no assumption involved and everything just follows from the rule of deduction.

**The following might be a useful paper to discuss. While the authors assume that the grid cells are representing one space, there is work suggesting that grid cells may represent multiple spaces.
Spalla, Davide, et al. "Can grid cell ensembles represent multiple spaces?." Neural Computation 31.12 (2019): 2324-2347.


**Summary Of The Paper:**

This paper develops a theory to understand the grid cell responses by combining ingredients from existing studies. They articulate three properties that are needed for the grid responses to appear when representing space: actionable, functional, and biological.
This paper contains some quite interesting ideas. But at the same time,  it is also highly problematic in multiple ways.  These concerns greatly diminished my enthusiasm for this paper.

**Summary Of The Review:**

An ambitious attempt to unify grid coding theory with several major concerns. Various improvements could be made to strengthen the paper.

==========
The revision addressed some of concerns and improved the paper, although some of my conerns remain. In my view, the revision has turned this into a borderline paper. I've increased my score from 3 to 5.

---

> ### Author Response · Authors · 2022-11-14
> **Thanks for Review! Response (1/3)**
>
> >  The attempt to unify a somewhat messy literature is ambitious and should be applauded. The proposed three key requirements for the grid patterns, if true, would be a useful finding. The argument in Section 3.1 seems to be quite insightful.
>
> We thank the reviewer for the enthusiasm for our aims. We hope that our responses clarify comments and are convincing that we have made inroads into achieving these aims.
>
> We really appreciate the detailed reading and suggestions regarding our submission - your comments have helped us create a better manuscript. We hope we have successfully answered your concerns below, and look forward to discussing further.
>
> >  __Weakness 1:__ The idea of “actionable representation” and the use of representation theory and group theory to study the grid cell system is not a new one (e.g., see a recent line of research in Gao et al, ICLR, 2019; Gao et al., NeurIPS, 2021). The current paper did refer to Gao et al 2021 in the Introduction and had a detailed comparison to that paper in the SI. But it did not acknowledge that using group theory to study the grid cell was a key idea in that line of work.
>
> Apologies, we had not meant to downplay the links to the group theory used in the work of Gao et al., and this was also highlighted by another reviewer. We have rewritten the SI section to highlight the use of group theory in this work, and the differences between our two approaches. The key difference is that we use group and representations theory to provide analytic justification for why grid cells emerge, whereas Gao et al. provide numerical simulations only. We also explain multiple modules, introduce biological constraints, and make neural predictions.
>
> We added this paragraph to Appendix H.1:
>
> *Theoretically, Gao et al. develop a rich group theory view for studying grid cells. We add to this work using Representation theory to make clear what constraints linear actionability implies (equation (5)). Given this starting point we are able to explain intuitively and with analytic insight why this loss leads to grid cells, extend to multiple modules, and make neural predictions.*
>
> And this sentence to the literature review:
>
> *Our dive into Representation Theory builds on the group theory ideas in Gao et al. (2018, 2021)*
>
> > __Weakness 4:__ The comparison to the data is rather preliminary and unconvincing.
>
> We certainly agree with you that these results are preliminary. We would love to test these claims more quantitatively, but we did the best comparisons possible to data we found online. We are now in the process of getting data to test these claims.
>
> That said, we believe the presented data is favourable and worth sharing. For example, no-one has (to our knowledge) noticed or explained the consistently small angles between grid modules, nor given a normative explanation of why some grids squash and others maintain their lengthscale when the room changes.
>
> > __Weakness 6__: The study will benefit by treating neural noise more carefully. It is unclear what assumption the model makes about the neural noise.
>
> The reviewer is right: we do not explicitly consider neural noise and it would be interesting to do so. Implicitly the loss (equation (1)) could be interpreted as minimising the overlap between noisy neural firing where the noise is gaussian with a lengthscale of order sigma. Adding noise in a more systematic or biological way seems like an interesting direction for future work.
>
> > __Weakness 2:__ The math presented in the paper is not rigorous.
>
> Many thanks for the comment. We understand your frustration that the maths does not appear rigorous in the main text. This was a choice of ours to appeal to both experimental and theoretical neuroscientists. We kept the rigorous maths to the appendix (Appendix A: the step from actionability to sines and cosines, Appendix C: the analysis of the simple functional loss, Appendix D: the high and low frequency biases, Appendix E: the harmonic penalty, Appendix L: how the loss changes in different rooms) and feel it is best for it to stay there to the paper is accessible to a wider audience.

---

> > ### Author Response · Authors · 2022-12-02
> > **Second Response**
> >
> > Hi!
> >
> > As the end of the discussion period is fast approaching, do let us know if you have any further comments or concerns to which we can respond. We'd be interested to hear if our comments and changes to the manuscript helped answer your questions, especially concerning the use of representation theory! Is the justification for our logic now clearer?

---

> ### Author Response · Authors · 2022-11-14
> **Response (2/3)**
>
> > __Weakness 2 Cont.__: Eq. (5) actually is a strong assumption. The paper seems to say that it is a necessary consequence of Eq. (3), thus misleading. Assuming matrix T consists of 1-by-1 and 2-by-2 blocks is a critical assumption that greatly simplified the problem.
>
> We thank the reviewer for highlighting a confusing aspect of our work that was not explained well enough. The crucial point is that Eq. 5 & 6 follow exactly from Eq. 3 for compact topological groups (e.g. the transformation group for variables on a circle, 2D and 3D torus, or sphere). This is because the Peter-Weyl theorem guarantees the existence of a set of irreducible representations, and that any representation of the group is a direct product of the irreducible representations of the group. This is what we used to arrive at the matrix T, made from 1-by-1 and 2-by-2 blocks. Hence, it is this theorem, and a couple of lines of simple logic shown in Appendix A.3, that means Eq. (3) to Eq. (6) is a deduction, not an assumption.
>
> We believe the confusion might have arisen because (A) we wrote the appendix to be intuitive, rather than formal, and (B) we did not highlight enough that at all times we are considering compact groups (i.e. the 2D variable is periodic) - rather than a non-periodic 2D space. In section 2 of the main paper we had mistakenly said variable x lived in $\mathbb{R}^2$, but we had meant the variable lived in $\mathbb{T}^2$, i.e. on a torus. We apologise for this, and have amended the text accordingly.
>
> We have re-written the paragraph that leads up to equation 6 in order to highlight the key step: our (arbitrarily good) approximation of a region of flat 2D space with an equivalently sized region of periodic 2D space. Again, apologies for not doing so earlier. The text now reads:
>
> *This argument comes from an area of maths called Representation Theory (a different meaning of representation!) that places constraints on the matrices $\bf{T}$ for variables whose transformations form a mathematical object called a group. This includes many of interest, such as position on a circle, torus, or sphere. These constraints on matrices can be translated into constraints on an actionable neural code just like we did for $\bf{g}(\theta)$ (see Appendix A). When generalising the above example to 2D space (a torus), we must consider a few things: First, the space is two-dimensional, so compared to our previous equation 5, the frequencies, denoted $\bf{k}_d$, are now two dimensional. Second, to approximate a finite region of flat 2D space, we consider a similarly sized region of a torus. As the radius of the torus grows this approximation becomes arbitrarily good (see Appendix A.4 for discussion). Periodicity constrains the frequencies in equation 5 to be $\frac{n}{R}$ for integer $n$ and ring radius $R$. As the loop (torus in 2D) becomes very large these permitted frequencies become arbitrarily close, so we drop the integer constraint,*
>
> As a note, you are correct that if we had been considering a non-compact group such as non-periodic 2D space, with representations like this:
> $$\bf{T}(\bf{\Delta x}) = \begin{pmatrix}1, 0, \Delta x\\\\ 0, 1, \Delta y \\\\ 0, 0, 1\end{pmatrix} $$
> then we could not go from eq 3 to eq 5 or 6. We have edited Appendix A.2, and added an extensive discussion of this point in appendix A.4.
>
> > __Weakness 7:__ how are Eq (3) and the positivity constraint compatible theoretically? The author seems to impose positivity after using Eq (3). If that is the case, Eq. (3) is no longer satisfied. I’d appreciate further clarification on this point.
>
> We agree that for the non-compact example above then non-negativity and eq. 3 are not compatible. However, Eq (3) and non-negativity are compatible theoretically: e.g. $\bf{g}(\bf{x})$ = vector of 1s, $\bf{T}(\bf{\Delta x})$ = identity matrix. Equally, take the example representation of a 1D variable in figure 3A. It is positive (since the orange curve lies in the positive orthant) and satisfies the 1D version of equation (3), since there is a matrix which rotates the activity around the orange curve.
>
> > __Weakness 3:__ The writing needs to be improved. Various statements need to be toned down. The math in the SI needs to be better organized. Phrases such as “big result” should probably be removed.
>
> Apologies for this. We have endeavoured to improve our writing. For example we have changed “big result” to relate to the Peter-Weyl theorem, which is indeed a big result for the development of our ideas - all our results rest on it! We hope all our changes have helped with readability.

---

> > ### Comment · Reviewer_7jav · 2022-12-02
> > **thanks! and some follow up questions**
> >
> > I would like to thank the authors for their response to my comments. This response and revision improve several aspects of the work.
> >
> > (1) I am still unconvinced about the argument of 2-by-2 blocks, and the claim that going from Eq (3) to Eq (6) follows from the deduction. The authors seem to take something (it is that "something" I'm trying to figure out) for granted. Why the representation should be compact? And why should it be two-d? Why should it be flat? The grid cell system embeds the 2-d space into a high-dimensional representation. It is not at all clear why the embedding dimension should be 2D, or why the representation should consist of 2D subspaces as the authors are using. To me, these are all assumptions.  I'd be thankful if the authors could clarify these issues further.
> >
> > (2)  It is also unclear to me why the model proposed here really cast the problem as a path integration problem. It appears that the only thing needed from the path integration argument is Eq (6). After that point, everything becomes a coding argument. Thus the claim of this theory being a path integration theory seems to be a bit misleading to me.

---

> > > ### Author Response · Authors · 2022-12-03
> > > **Response to Follow Up Questions (1/2)**
> > >
> > > Thanks for the response! We look forward to hashing this out!
> > >
> > > > (1) I am still unconvinced about the argument of 2-by-2 blocks, and the claim that going from Eq (3) to Eq (6) follows from the deduction. The authors seem to take something (it is that "something" I'm trying to figure out) for granted. Why the representation should be compact? And why should it be two-d? Why should it be flat? The grid cell system embeds the 2-d space into a high-dimensional representation. It is not at all clear why the embedding dimension should be 2D, or why the representation should consist of 2D subspaces as the authors are using. To me, these are all assumptions. I'd be thankful if the authors could clarify these issues further.
> > >
> > > So, let’s first establish that the 2-by-2 blocks, and the fact that $g(x)$ is a 2D manifold, are separate.
> > >
> > > Our starting premise is that we want to understand how the brain represents periodic 2D position, i.e. the vector of neural firing rates $g(x)$. $x$ is a 2-dimensional compact variable, therefore $g(x)$ is necessarily 2-dimensional and compact. Every position on the manifold of all possible $g(x)$ is specified by a position in periodic 2D space, $x$, no assumption there! Let's call the set of all possible $g(x)$ the activity manifold.
> > >
> > > Further, there is no assumption that the activity manifold is flat, it can be, and is, very curvy! None of our representations, $g(x)$, can be understood as activity lying in a flat 2D plane, they are all curvy.
> > >
> > > So, we’re dealing with a curvy 2-dimensional activity manifold in a high dimensional neural space, and we’d like to choose this manifold so that it is the best, by our definition of best. Great. Now let’s return to the 2-by-2 blocks, the irreps.
> > >
> > > The fact that both the irreps and the activity manifold are 2-dimensional is a numerological accident! Generally they won’t have the same dimensionality. If you are considering the representation of an angle then $g(\theta)$ is a curvy 1-dimensional manifold (since the $\theta$ is 1-dimensional) but the irreps are still 2-by-2. A more extreme example is the representation of a position on a sphere; there again the position is 2-dimensional, hence $g(x)$ is again a curvy 2-dimensional manifold, but the irrep blocks can take any odd-numbered dimensionality (i.e. 3-by-3 blocks, or 9-by-9 blocks).
> > >
> > > Okay, so we’ve established that the dimensionality of the representation and the irreps are independent quantities that both happen to be 2 for 2D space. Now let’s go through one last section on what these 2-by-2 spaces are.
> > >
> > > The only assumption we make is actionability: we want there to exist matrices for all movements, $T(\Delta x)$, that consistently update the representation $T(\Delta x)g(x) = g(x + \Delta x)$. This is core to our programme, it makes the representation interesting, allowing it to perform clever things like zero-shot inference of next position given an action, as discussed in the main paper. To reiterate, that is our only additional constraint!
> > >
> > > It is then a deduction that uses the Peter-Weyl theorem and a couple of lines of logic to say that $g(x)$ has the form shown in equation (6). And it is this deduction that we invest effort to try to make intuitive (though it seems we have failed!).
> > >
> > > The game is the following: find a set of matrices $T(\Delta x)$ that when you multiply them together using matrix multiplication they obey the same rules that adding up 2D translations obey on a periodic 2D space. Things like $T(\Delta x_1) T(\Delta x_2) = T(\Delta x_1 + \Delta x_2)$, $T(-\Delta x) = T(\Delta x)^{-1}$, $T(0) = $Identity Matrix. It turns out any real matrix that satisfies all these rules (and is therefore a representation of the group) can, up to a linear transform, be made from 2-by-2 blocks each of which is rotation at some frequency; a consequence of the Peter-Weyl theorem. Our efforts to make this intuitive highlight the fact that rotation matrices satisfy these properties, and that stackings of rotation matrices also satisfy these properties. We do not have an intuitive answer for why these are the only possible matrices, but they are nonetheless!
> > >
> > > To briefly elaborate on one point: the only constraint that this implies on $g(x)$ is shown in equation 6. This does not make it flat. It is true and useful that if you apply an appropriate linear transform to $g(x)$ you can arrange things so that in each 2-dimensional subspace the activity loops round a circle, but importantly (A) that required a linear transform, without which this is not true, (B) even ignoring (A), the fact that you can project it onto 2D subspaces with easy transition rules does not mean the activity manifold is flat! The activity manifold is the sum of all these different components, hence can have complex curvy properties in the higher-dimensional neural space.
> > >
> > > Does that make any more sense? Please continue to ask questions, there’s some misunderstanding here that it would be great to hash out!

---

> > > ### Author Response · Authors · 2022-12-03
> > > **Response to Follow Up Questions (2/2)**
> > >
> > > > (2) It is also unclear to me why the model proposed here really cast the problem as a path integration problem. It appears that the only thing needed from the path integration argument is Eq (6). After that point, everything becomes a coding argument. Thus the claim of this theory being a path integration theory seems to be a bit misleading to me.
> > >
> > > Let’s define path-integration as the following: given a start position and a series of movements, you are able to say which position you end up in. Seems reasonable?
> > >
> > > This requires two things, that we have conveniently decomposed into actionability and functionality. You must be able to sum up the series of movements and work out their net effect (actionability). Then, once you’ve worked out where you’ve moved to in your representation, you have to be able to say which position you’ve ended up at (functionality). Actionability ensures you can add up movements and work out their consequences (by ensuring the representation transition rules mirror the variable’s transition rules). Functionality ensures you can tell which position you are at (by making the representation of all the points you might visit different from one another).
> > >
> > > So, by this argument, the whole paper is a path-integrating theory! We really are trying to get the core ideas necessary for a representation to path-integrate, and find out what that implies for a representation, that is all. Sorry if this was not made clear.
> > >
> > > One final comment: the fact that actionability leads us to a very simple criterion (equation (6)) is, in our view, something to be celebrated! To us it is concise, beautiful, and useful. That does not mean it is unimportant! In fact, it is a (perhaps the?) key piece of our work. Without actionability grids are not optimal (see ablation study, appendix I). Further, it is the novel theoretical development that will hopefully generalise. So while, yes, actionability simply gets us to eqn. (6), that it does belies its fundamental importance to our work!

---

> ### Author Response · Authors · 2022-11-14
> **Response (3/3)**
>
> > __Comment 1:__ It is unclear what the next paragraph really means: “To motivate our current rabbit hole further, this is exactly the result we hinted towards in Section 2. We discussed T(∆θ), and, in 2-dimensions, argued it was, up to a linear transform, the 2-dimensional rotation matrix. Further, we discussed how every extra two neurons allowed you to add another frequency, i.e. another rotation matrix. These rotation matrices are the irreps of the rotation group, and adding two neurons allows you to create a larger T (∆θ) by stacking rotation matrices on top of one another. Now including the invertible linear transform, S, we can state the 4-dimensional version of equation 4.” I interpreted that the authors were making an important assumption here according to their math, but from the text, it read like there is no assumption involved and everything just follows from the rule of deduction.
>
> Sorry for the lack of clarity. Hopefully the above responses mean that this paragraph makes more sense. In brief - it is not an important assumption, rather, it is trying to provide intuition for the Peter-Weyl theorem, and how that result relates to the intuition we provided in section 2 of the main paper. We hope our paper will be read by neuroscientists, and so aimed for maximal understandability. We have left this paragraph as is for now, hoping that our preceding discussion will make it intelligible. Please let us know if it is still opaque.
>
> > __Weakness 5:__ Eq. (1) needs to be better motivated.
>
> Many thanks for the comment. To understand a space an animal/agent must 1) have different representations for different locations otherwise it would confuse two different locations. 2) it cannot understand locations exactly as that would require infinite neurons. 3) There is no point trying to represent locations you never go to. Eq 1 is a loss that characterises these three things: The functional loss itself consists of three parts - 1) making sure each location is coded differently, 2) caring about this difference up to a certain resolution, and 3) only caring about locations actually visited.
>
> > The second and third terms together lead to a band-pass filter— this seems to be rather post-hoc. Previously, it was known that band-pass filtering can lead to grid patterns (Sorscher, Mel, et al, NeurIPS 2019). Some previous studies used the different-of-Gaussian place fields as inputs to the grid cell network, leading to band-pass filtering. This connection needs to be explained. How the present theory is different from Sorscher, Mel, et al, NeurIPS 2019 also needs to be better explained.
>
> As above, these terms are not post-hoc. We appreciate your comment on the  relationship to Sorscher, Mel et al 2019. However, their rationale for hexagons is very different to ours. Sorcsher, Mel et al. show that non-negativity leads to triplet interactions that produce hexagonal grids. This argument however is dependent on some specific (and likely implausible - Schaeffer et al 2022) aspects of the place cell code. Our argument is that non-negativity leads to a lattice of frequencies, and that a high pass (from p(x)) and low pass filter (from chi, i.e.  only resolving space up to a certain scale) selects a hexagonal lattice due to optimal packing. These are two very different arguments.
>
> Beyond these technical differences, our work captures key aspects of the grid code that are not present in Sorscher, Mel et al.’s description. For example we can explain in our framework why grid axes align. Further, we provide a normative derivation of multiple modules, which we believe only arise in Sorscher, Mel et al. due to quantization of the position (Figure 3). But most crucially, our predictions come as a result of our normative framework for understanding generic representations.
>
> We have added a section, Appendix H.2, that talks through these points.
>
> > __Comment 2:__ The following might be a useful paper to discuss. While the authors assume that the grid cells are representing one space, there is work suggesting that grid cells may represent multiple spaces. Spalla, Davide, et al. "Can grid cell ensembles represent multiple spaces?." Neural Computation 31.12 (2019): 2324-2347.
>
> Thank you for pointing us to this work, which we were not aware of previously. Our work does indeed assume that at any one time the grid cells represent one space, but at different times, depending on the shape of the environment, the grid cells may choose to map that to real space in different ways, e.g. by stretching and warping the grid. This underlies the analysis of how the different room shapes affect the optimal solution, and the suggestion that the grid cells change their coding to reflect the optimal solution in that room, a suggestion that is supported by data (Stensola et al. 2015). This could be implemented mechanistically by up or down-weighting the velocity input to a continuous attractor network in different environments.

---

> ### Author Response · Authors · 2022-12-05
> **Thanks for Revised Score!**
>
> Hi!
>
> Thanks for updating your evaluation of our work, we really appreciate your help improving this manuscript.
>
> We remain open to discussing confusing aspects of our work and answering any questions you have.

---

### Official Review · Reviewer_XxrH · 2022-10-25

**Confidence:** 3
**Correctness:** 3
**Technical Novelty And Significance:** 3
**Empirical Novelty And Significance:** 3
**Recommendation:** 8

**Clarity, Quality, Novelty And Reproducibility:**

About novelty, the group representation perspective has been previously investigated by Gao et al. (2021) and earlier Gao et al. (Learning grid cells as vector representation of self-position coupled with matrix representation of self-motion. ICLR 2019).

About actionable, any RNN that transforms the hidden vector based on the input action is an actionable representation, including the state space model in control, as well as latent space model-based RL, such as the dreamer model. The hidden vectors in most RNNs are actionable if you interpret the input as an action. As long as the transformation depends on the action, it does not need to be linear. In fact, the linear transformation or matrix group is not entirely biologically plausible because the recurrent connection weight matrix depends on the action.

The paper is clearly written, and the results and predictions are clearly explained.

The mathematical reasoning is good, but not very rigorous, at least in the main text.

Assuming Fourier plane waves is too specific and restrictive, with little biological plausibility. The continuous attractor network seems more biologically plausible.

About nonnegativity constraint, it is biologically true. But it is unclear if it is entirely necessary theoretically. Gao et al (2019, 2021) learned hexagon patterns without this constraint.

**Strength And Weaknesses:**

Strengths:

(1) The proposed loss function is new and reasonable.

(2) The explanation of the emergence of hexagon grid patterns is sound.

(3) The predictions seem to be supported by empirical evidence.

Weaknesses:

(1) The mathematical analysis in the main text is not very rigorous, although it is quite insightful and intuitive.

(2) The superposition of sine and cosine waves seems too restrictive, and may not model biological neurons.

(3) The proposed loss function, although reasonable, is not simple enough to serve as a first principle.

(4) The linear transformation model does not cover the continuous attractor network.




**Summary Of The Paper:**

This paper seeks to explain the properties of grid cells. Linear group representation consideration leads to the superposition of sine and cosine plane waves. A specially designed loss function is proposed to explain the emergence of hexagon grid patterns under the constraints of non-negative and bounded firing rate. The paper makes several predictions based on the proposed theory.

**Summary Of The Review:**

The paper provides an interesting explanation of the emergence of hexagon grid patterns of the grid cells. However, the loss function is not minimalistic and the superposition of Fourier plane waves is not generic enough.

---

> ### Author Response · Authors · 2022-11-14
> **Thanks for Review! Response (1/2)**
>
> > This paper seeks to explain the properties of grid cells. Linear group representation consideration leads to the superposition of sine and cosine plane waves. A specially designed loss function is proposed to explain the emergence of hexagon grid patterns under the constraints of non-negative and bounded firing rate. The paper makes several predictions based on the proposed theory.
>
> We thank you for your detailed and useful comments on our work. We have made changes accordingly, and hope that these changes, along with our response here, address your concerns. We look forward to continued discussion in the coming weeks.
>
> >  __Strengths:__
> > 1) The proposed loss function is new and reasonable.
> > 2) The explanation of the emergence of hexagon grid patterns is sound.
> > 3) The predictions seem to be supported by empirical evidence.
>
> Many thanks!
>
> >  __Weakness 1:__ The mathematical analysis in the main text is not very rigorous, although it is quite insightful and intuitive.
> > __And:__ The mathematical reasoning is good, but not very rigorous, at least in the main text.
>
> Many thanks for the comment. We understand your frustration that the maths does not appear rigorous in the main text. This was a choice of ours to appeal to both experimental and theoretical neuroscientists. We kept the rigorous maths to the appendix (Appendix A: the step from actionability to sines and cosines, Appendix C: the analysis of the simple functional loss, Appendix D: the high and low frequency biases, Appendix E: the harmonic penalty, Appendix L: how the loss changes in different rooms) and feel it is best for it to stay there to the paper is accessible to a wider audience.
>
> >  __Weakness 2:__ The superposition of sine and cosine waves seems too restrictive, and may not model biological neurons.
>
> This is fair. We agree that our approach is simple, and we see that as a positive since the approach is general to any group representation, and the simplicity still captures many aspects of the grid cell code and makes numerous neural predictions.
>
> In general, however, a superposition of sines and cosines is very unrestrictive and can account for any smooth function as per Fourier. Our case is more restrictive than that since we only allow a certain number of frequencies present (with more frequencies allowed with more neurons), rather than the potentially infinite frequencies as per Fourier. Thus our representation derived from group theory, is neither unrestrictive nor too restrictive - the number of neurons controls this restrictiveness.
>
> Overall, we agree that our theory is not a perfect model of biology, and there are cases, highlighted in the discussion, where our model will be wrong. However, the test of whether it is a useful model will come down to whether it allows new phenomena to be understood, or provides new interpretations. We are hopeful that this is the case, and have presented preliminary evidence for this. We are in the process of more quantitatively testing our theory’s predictions in data, though this will not be ready for many months.
>
> >  __Weakness 3:__ The proposed loss function, although reasonable, is not simple enough to serve as a first principle.
>
> Many thanks for the comment. We hope that it is clear from our theory and ablation studies that each component - actionability, functional, and biological - are each crucially important. Without any one of these, then we do not observe grid cells, but with all three we do, thus these three are minimal. Perhaps you are concerned that the functional loss is not simple enough? In this case we hope that we are able to convince you here: the functional loss itself consists of three parts - 1) making sure each location is coded differently, 2) caring about this difference up to a certain resolution, and 3) only caring about locations actually visited. This is the minimal set that an agent/animal has to understand: 1) it needs to have different representations for different locations otherwise it would confuse two different locations. 2) it cannot understand locations exactly as that would require infinite neurons. 3) There is no point trying to represent locations you never go to.
>
> We are unsure of how to answer this further though, as it is not clear why you think the proposed loss function is not simple enough. Nevertheless, we hope we have convinced you that it is in our above argument.

---

> > ### Author Response · Authors · 2022-12-02
> > **Second Response**
> >
> > Hi!
> >
> > As the end of the discussion period is fast approaching, do let us know if you have any further comments or concerns to which we can respond. We'd be interested to hear if our comments and changes to the manuscript helped answer your questions, especially concerning the minimality of our loss and the restrictiveness of sines and cosines!

---

> ### Author Response · Authors · 2022-11-14
> **Response (2/2)**
>
> > __Weakness 4:__ The linear transformation model does not cover the continuous attractor network.
> __And:__ Assuming Fourier plane waves is too specific and restrictive, with little biological plausibility. The continuous attractor network seems more biologically plausible.
> __And:__ About actionable, any RNN that transforms the hidden vector based on the input action is an actionable representation, including the state space model in control, as well as latent space model-based RL, such as the dreamer model. The hidden vectors in most RNNs are actionable if you interpret the input as an action. As long as the transformation depends on the action, it does not need to be linear. In fact, the linear transformation or matrix group is not entirely biologically plausible because the recurrent connection weight matrix depends on the action.
>
> This is a great point. We agree that it is likely that the brain uses a continuous attractor network and we agree that our theory does not cover the continuous attractor network. However, we note that our theory does not try to either. Instead we propose a normative theory of what spatial representations in the brain should look like, rather than considering the particular machinery that the brain may use. We are not trying to understand mechanism, we are trying to understand representational principles.
>
> We do think that it would be possible to approach the same problem from the constraints that a continuous attractor network would impose, and in particular when the activation function is non-negative. There are already some insights by Sorscher, Mel et al 2019 on this, but we suspect it may be possible to incorporate some of our analysis in too. However, we leave this for future work.
>
> It’s true that any RNN that appropriately transforms a hidden vector based on an input action has an actionable representation. The crucial thing that we do though is translate actionable into a theoretical framework, where it can be analysed with maths and we can learn something about the neural representations of an actionable representation.
>
> > __Comment 1:__ About novelty, the group representation perspective has been previously investigated by Gao et al. (2021) and earlier Gao et al. (Learning grid cells as vector representation of self-position coupled with matrix representation of self-motion. ICLR 2019).
>
> This is true, and though we tried, we should have done a better job of highlighting the similarities and differences of our work from Gao et al.’s.. We have added the following sentence to the literature review:
>
> *Our dive into Representation Theory builds on the group theory ideas in Gao et al. (2018, 2021)*
>
> And added this paragraph to Appendix H.1:
>
> *Theoretically, Gao et al. develop a rich group theory view for studying grid cells. We add to this work using Representation theory to make clear what constraints linear actionability implies (equation (5)). Given this starting point we are able to explain intuitively and with analytic insight why this loss leads to grid cells, extend to multiple modules, and make neural predictions.*
>
> > __Comment 2:__ About the nonnegativity constraint, it is biologically true. But it is unclear if it is entirely necessary theoretically. Gao et al (2019, 2021) learned hexagon patterns without this constraint.
>
> This is a good point. Gao et al. do indeed learn hexagons without non-negativity, but with an additional isotropy term in the loss. We are very interested in analysing how this emerges. Regardless, it seems you have to choose either non-negativity or isotropy. Considering that non-negativity is an intrinsic biological constraint, while isotropy is not, we believe non-negativity is sensible to include.

---

### Official Review · Reviewer_i5nn · 2022-10-25

**Confidence:** 3
**Correctness:** 3
**Technical Novelty And Significance:** 4
**Empirical Novelty And Significance:** Not applicable
**Recommendation:** 8

**Clarity, Quality, Novelty And Reproducibility:**

Among the EC-HPC models, the paper is quite novel in terms of technicality. The ideas are clearly communicated, and as long as the simulation codes are released it should be reproducible.

**Strength And Weaknesses:**

Strength:
- To-date, most computational models on grid cell emergence have focused on training a neural network to solve path-integration tasks and selecting grid cells based on their known neuroscientific properties. This creates a possibility, which also applies to other EC-HPC representations, that the neural representations are simply due to correlation with different behavioral variables (i.e. “you see what you’re looking for”). In this paper, the authors only imposed normative constraints rather than assumptions about known properties of grid cells, and still observed grid cells, which fills the gap in the literature on why EC-HPC representations exist in the first place.

Major weaknesses:
- Though it provides a good model for grid cell emergence, this framework is somewhat isolated in that it doesn’t consider the interactions between MEC (where grid cells are observed) and other regions in the EC-HPC circuit. For example, under this framework, do grid cells still help explain place cell remapping (Whittington et al., 2020)?
- The metric described in this paper (i.e. functional, biological, actionable) still does not account for whether the grid cells representations are useful for behavior, which in my opinion is what “functional” should mean. I think it would make the paper even stronger if you could show the connection between your model and path integration behavior, or some other behavioral task, eg. train model on path integration task and do lesion study.
- As raised in Schaeffer et al., 2022, grid cell emergence depends on hyperparameters and implementation choices. Is that also the case here? Eg. What happens when you increase the number of neurons? Or other hyperparameters listed in section B.3? It would be interesting to see whether the claim from Schaeffer et al. still holds when grid cells emerge from not path-integration but mathematical optimization.
- One can also see grid cells in non-spatial settings (Whittington et al., 2020), where a different set of constraints may apply (eg. the resolution of representation does not necessarily follow a Goldilocks of frequency). And grid cells have been shown to represent other behaviourally relevant variables besides location, such as time and distance [(Kraus et al., 2015)](https://www.cell.com/neuron/fulltext/S0896-6273(15)00820-X?_returnURL=https%3A%2F%2Flinkinghub.elsevier.com%2Fretrieve%2Fpii%2FS089662731500820X%3Fshowall%3Dtrue) . How can your model account for this? Or is it only applicable to spatial location?


Minor weaknesses:
- Please remake all hand drawn figures, including the ones in appendix, with computer softwares
- Page 5 first paragraph: “... and $\theta$ dependent parts (equation 6; Figure 3A)”: 1) Adding a hyphen, i.e. $\theta$-dependent, might be helpful for readers. 2) Do you mean equation 5?
- Page 5 first paragraph: “...This effect is limited by the firing rate bound: $||a_0||^2 - 2 L_0 = N$..."
Do you mean $1/2 L_0$ ?



**Summary Of The Paper:**

In this paper, the authors provided a solution of how to parametrically hard code grid cells with learnable parameters in order to be functional, actionable, and biological by minimizing the objective function. Under these three constraints, they showed that hexagonal grid cells are the optimal representation of locations in 1D angular and 2D euclidean space.

**Summary Of The Review:**

I’d recommend accepting this paper for ICLR, as it provides a normative explanation for why the multi-module representation of grid cells is the optimal representation of space. The paper is well-written and clearly communicated, but also please remake your hand-drawn figures.

---

> ### Author Response · Authors · 2022-11-14
> **Thanks for Review! Response (1/2)**
>
> > In this paper, the authors provided a solution of how to parametrically hard code grid cells with learnable parameters in order to be functional, actionable, and biological by minimizing the objective function. Under these three constraints, they showed that hexagonal grid cells are the optimal representation of locations in 1D angular and 2D euclidean space.
>
> We would like to thank you for carefully reading and commenting on our work. We appreciate your comments, which, as we detail below, we have tried to address, and which have helped us to improve the manuscript. We’d also like to point out that we applied our framework to spherical and 3D spaces too! (deep in Appendix M ;) )We look forward to discussing any remaining concerns further.
>
> > To-date, most computational models on grid cell emergence have focused on training a neural network to solve path-integration tasks and selecting grid cells based on their known neuroscientific properties. This creates a possibility, which also applies to other EC-HPC representations, that the neural representations are simply due to correlation with different behavioral variables (i.e. “you see what you’re looking for”). In this paper, the authors only imposed normative constraints rather than assumptions about known properties of grid cells, and still observed grid cells, which fills the gap in the literature on why EC-HPC representations exist in the first place.
>
> Many thanks!
>
> > Though it provides a good model for grid cell emergence, this framework is somewhat isolated in that it doesn’t consider the interactions between MEC (where grid cells are observed) and other regions in the EC-HPC circuit. For example, under this framework, do grid cells still help explain place cell remapping (Whittington et al., 2020)?
>
> This is a fair comment: our model is not a model of how the EC and HPC interact. Our work however does extract the EC component from Whittington et al. 2020’s Tolman-Eichenbaum machine (TEM) and explains why it produces grid cells - we understand the constraints of the grid code. However, should we use a conjunctive representation of the hippocampus (as in Whittington et al. 2020) between our grid cells and sensory observations then we would see a similar remapping phenomena to Whittington et al. 2020. In fact, any model with griddy entorhinal representations and a TEM-like EC-HPC conjunctive scheme would lead to the same remapping results.
>
>
> > The metric described in this paper (i.e. functional, biological, actionable) still does not account for whether the grid cells representations are useful for behavior, which in my opinion is what “functional” should mean. I think it would make the paper even stronger if you could show the connection between your model and path integration behavior, or some other behavioral task, eg. train model on path integration task and do lesion study.
>
> We agree that ‘functional’ and ‘actionable’ should bear relevance for behaviour, and we have designed them exactly for this purpose. Together these constraints mean the representation path-integrates, and path integration is critical for behaviour as without it you cannot plan etc.  In particular, 1) An actionable representation gets updated by an action consistent with how actions work in the underlying space (i.e. the representation obeys the same rules that space does - north+east+south+west takes both the actionable representation, and the real world, back to the same place). 2) Being functional means each location can be uniquely decoded up to some resolution, thus you can understand different parts of the world differently - a requirement for successful behaviour. These two components are equivalent to path-integration since path integration is updating a representation on the basis of an action to track real location. We have added the following paragraph in Appendix H (due to space constraints)and referred to it in the main text:
>
> _Our theory is a theory of path-integrating representations. Actionability means you understand the rules of space (a representation should be unchanged after taking a north then east then south then west), but it does not mean you understand where you are, or how fine-grained this understanding is (a constant representation satisfies the actionable rules for example). Our functional constraint rectifies this, as it requires all relevant locations to be represented differently, up to a certain spatial scale. Understanding rules, and representing locations differently is the same as path-integration - now a representation can be updated according to the rules, and it is different in different locations meaning the underlying spatial variables can be decoded._
>
>
> Furthermore, in Appendix I we show the effect of lesioning various constraints on the representations. With no functional term, the representation still follows rules but it does not successfully map the whole space; while with no actionable term it encodes space well but without rules.

---

> > ### Author Response · Authors · 2022-12-02
> > **Second Response**
> >
> > Hi!
> >
> > As the end of the discussion period is fast approaching, do let us know if you have any further comments or concerns to which we can respond. We'd be interested to hear if our comments and changes to the manuscript helped answer your questions, especially about the parameter dependence of our solutions!

---

> ### Author Response · Authors · 2022-11-14
> **Response (2/2)**
>
> > As raised in Schaeffer et al., 2022, grid cell emergence depends on hyperparameters and implementation choices. Is that also the case here? Eg. What happens when you increase the number of neurons? Or other hyperparameters listed in section B.3? It would be interesting to see whether the claim from Schaeffer et al. still holds when grid cells emerge from not path-integration but mathematical optimization.
>
> This is a very reasonable concern that we did not address adequately in our previous submission. We have now extensively updated appendices B.3 and B.4 to include a thorough discussion of the parameters, and our solution's robustness.
>
> We focus on 4 parameters that together define key parts of the problem: the two spatial lengthscales, l and L, the neural lengthscale, sigma, and the number of neurons, N. Since the position units are arbitrary we set L = 1, leaving us with three parameters to explore (The neural space units are not arbitrary due to the firing rate constraint).
>
> Our theory actually makes predictions on how these parameters change the representation. As discussed in appendix E.3, we expect the number of modules to depend on the ratio N:sigma. A small ratio means few modules, and a large ratio means many modules. The role of l is simpler: if it is large we should get hexagons, if it is sufficiently small it will have no effect and we should get arbitrary grids.
> We have now added simulations that verify these claims:
> 1) For fixed N and sigma, for a range of l one module of hexagons is optimal, figure 8. When l is too small, hexagons are no longer optimal.
> 2) For fixed N and l, decreasing sigma transition from a single module to multiple modules. The smaller sigma the more modules, figure 9.
> 3) You can vary the N, and find qualitatively similar behaviours at all population sizes tested.
>
> Our results do not produce hexagonal grid cells for all possible parameters, but they do for a reasonable range. Thus, we are open to the possibility that there are additional constraints the brain faces that we have not considered here. We have added the following sentence on this:
>
> *Future work could usefully explore whether more constraints are needed to robustly generate many hexagonal modules, or whether more neurons is enough as it was in figure 8.*
>
> > One can also see grid cells in non-spatial settings (Whittington et al., 2020), where a different set of constraints may apply (eg. the resolution of representation does not necessarily follow a Goldilocks of frequency).
>
> Many thanks for the comment. In our work the goldilocks frequency preference came from two lengthscales: the agent explores a limited range so you want frequencies high enough to oscillate in that range, and there is a limited resolution with which you want to encode the world that penalises high frequencies. We think the two of these forces are still at work in TEM. There is a limited map to encode, which enforces a low frequency cutoff, and all the points are separated by at least a lattice lengthscale (i.e. there is no continuity of points) and so there is a high-frequency cutoff. Frequencies that oscillate faster than the lattice lengthscale are not useful. As such, even in TEM, there is a push towards Goldilocks frequencies that may explain the emergence of hexagons. We have added this discussion to Appendix H, where we discuss links to Whittington et al. 2020 and other works.
>
> _In particular, the goldilocks frequency bounds (section 3.2) can be thought of as coming from the finite map the agent explores (high frequency bias) and the discrete step-size in the environment, which creates a low frequency bias_
>
> >  And grid cells have been shown to represent other behaviourally relevant variables besides location, such as time and distance (Kraus et al., 2015) . How can your model account for this? Or is it only applicable to spatial location?
>
> Many thanks for the comment. Our theory, like TEM, applies to situations where there is a common set of actions that act on some space, and so if that variable is structured like physical space then we can build analogous representations for such variables. Unlike TEM which only operates in discrete graph worlds, our theory applies to both discrete and continuous spaces. In Appendix M we applied our framework to 1D, 2D, 3D, and spheres, Appendix M. Both distance and time are 1D variables so a naive prediction would also be modules of grids in 1D, but we are actively working on applying our ideas more broadly in other work.
>
> > Please remake all hand drawn figures, including the ones in appendix, with computer softwares
>
> Done!
>
> > Page 5 first paragraph: “... and θ dependent parts (equation 6; Figure 3A)”: 1) Adding a hyphen, i.e. θ-dependent, might be helpful for readers. 2) Do you mean equation 5?
>
> Many thanks! Good spot. We have changed these accordingly.
>
>
> > Page 5 first paragraph: “...This effect is limited by the firing rate bound: ||a0||2−2L0=N..." Do you mean 1/2L0?
>
> Done! Many thanks!

---

### Author Response · Authors · 2022-11-14
**General Response to Reviewers**

We would like to thank all the reviewers for their time and comments. We appreciated their feedback and their comments have greatly improved the manuscript.

We have sent detailed responses to each of your comments, and have made numerous edits to the manuscript according to your comments and to address parts of the paper that were not stated clearly the first time around. In the submission we have highlighted the larger changes in a sort of plum colour. Below we summarise major changes we made to the manuscript, we look forward to discussing more in the coming weeks!

__Summary of larger changes to the Manuscript:__

1) In section 2 and Appendix A we have tried to outline our use of representation theory more formally. In particular, our reliance on approximating a small region of flat 2D space arbitrarily well with a small region of a very large 2D periodic space, and the pivotal role the Peter-Weyl theorem plays in our arguments.
2) In Appendix B (B.3 and B.4) we have added a much more extensive numerical exploration of the role of the key parameters that define our loss (l, L, sigma, N) on the optimal representations.
3) In the Related work section and Appendix H we have tried to highlight the relationship of our work to related ideas in Gao et al. (2018, 2021). Further, we have added a section highlighting the similarities and differences between our work and that of Sorscher, Mel et al. 2019.

---

### Author Response · Authors · 2022-11-18
**Last update to include very recent paper!**

We minorly revised our submission one final time to include the results of a very recent preprint (released this week!)

Otherwise the submission is unchanged.

We look forward to discussion with the reviewers!

---

### Author Response · Authors · 2022-12-02
**Anonymous Git repo and invitation for last questions!**

Hi All,

We'd like to share an anonymous github repo with all our code here:

https://anonymous.4open.science/r/ICLR_Actionable_Reps-67B5/readme.md

Further, as the end of the discussion period is fast approaching, we would appreciate hearing any feedback or lingering questions that you may have!

---

### Decision · Program_Chairs · 2023-01-20

**Decision:**

Accept: poster

**Justification For Why Not Higher Score:**

See weaknesses above.

**Justification For Why Not Lower Score:**

I am not an expert in the area, but the reviews agree that the paper generates insights and ideas that may be of interest to the neuroscience/cognitive science communities studying grid cells and path integration phenomenon. As such I am leaning to acceptance, but the paper can be bumped down.

**Metareview: Summary, Strengths And Weaknesses:**

Summary: Motivated by biological mechanisms like firing f grid cells and associated path integration for inferring pose from ego-centric observations, this paper is on the theme of understanding the optimal latent representation of 2D space. The paper proposes a learnable embedding trained by minimizing an objective function capturing equivariance, non-negativity and geometric requirements. It concludes with arguments that observed hexagonal representation of grid cells is the optimal representation of space.

Strengths:  The paper is clearly written, and the results and predictions are clearly explained. The reviewers found the conclusions to be insightful and unifying for this literature.

Weaknesses: Some concerns were raised in the reviews about (a) novelty as the the group representation perspective has been previously investigated by Gao et al. (2021); (b) lack of rigor in the mathematical analysis; (c) limited capacity of the proposed trigonometric model; and (d) technical presentation issues.

**Note From Pc:**

if the above contains the word "oral" or "spotlight" please see: "oral" presentation means -> notable-top-5% and "spotlight" means -> notable-top-25%. As stated in our emails, we are disassociating presentation type from AC recommendations